# Dichotomy of Early and Late Phase Implicit Biases Can Provably Induce Grokking

**Kaifeng Lyu**[*]
Princeton University
klyu@cs.princeton.edu

**Jikai Jin**[*]
Stanford University
jkjin@stanford.edu

**Zhiyuan Li**
Toyota Technological Institute at Chicago
zhiyuanli@ttic.edu

**Simon S. Du**
University of Washington
ssdu@cs.washington.edu

**Jason D. Lee**
Princeton University
jasonlee@princeton.edu

**Wei Hu**
University of Michigan
vvh@umich.edu

## Abstract

Recent work by Power et al. (2022) highlighted a surprising "grokking" phenomenon in learning arithmetic tasks: a neural net first "memorizes" the training set, resulting in perfect training accuracy but near-random test accuracy, and after training for sufficiently longer, it suddenly transitions to perfect test accuracy. This paper studies the grokking phenomenon in theoretical setups and shows that it can be induced by a dichotomy of early and late phase implicit biases. Specifically, when training homogeneous neural nets with large initialization and small weight decay on both classification and regression tasks, we prove that the training process gets trapped at a solution corresponding to a kernel predictor for a long time, and then a very sharp transition to min-norm/max-margin predictors occurs, leading to a dramatic change in test accuracy.

## 1 Introduction

The generalization behavior of modern over-parameterized neural nets has been puzzling: these nets have the capacity to overfit the training set, and yet they frequently exhibit a small gap between training and test performance when trained by popular gradient-based optimizers. A common view now is that the network architectures and training pipelines can automatically induce regularization effects to avoid or mitigate overfitting throughout the training trajectory.

Recently, Power et al. (2022) discovered an even more perplexing generalization phenomenon called *grokking*: when training a neural net to learn modular arithmetic operations, it first "memorizes" the training set with zero training error and near-random test error, and then training for much longer leads to a sharp transition from no generalization to perfect generalization. See Section 2 for our reproduction of this phenomenon. Beyond modular arithmetic, grokking has been reported in learning group operations (Chughtai et al., 2023), learning sparse parity (Barak et al., 2022; Bhattamishra et al., 2023), learning greatest common divisor (Charton, 2024), and image classification (Liu et al., 2023; Radhakrishnan et al., 2022).

Different viewpoints on the mechanism of grokking have been proposed, including the slingshot mechanism (cyclic phase transitions) (Thilak et al., 2022), random walk among minimizers (Millidge, 2022), slow formulation of good representations (Liu et al., 2022), the scale of initialization (Liu et al., 2023), and the simplicity of the generalizable solution (Nanda et al., 2023; Varma et al., 2023). However, existing studies failed to address two crucial aspects for gaining a comprehensive understanding of grokking:

1. No prior work has rigorously proved grokking in a neural network setting.

2. No prior work has provided a quantitative explanation as to why the transition from memorization to generalization is often sharp, instead of gradual.

---

[*]Equal Contribution

[1]Code is available at https://github.com/vfleaking/grokking-dichotomy

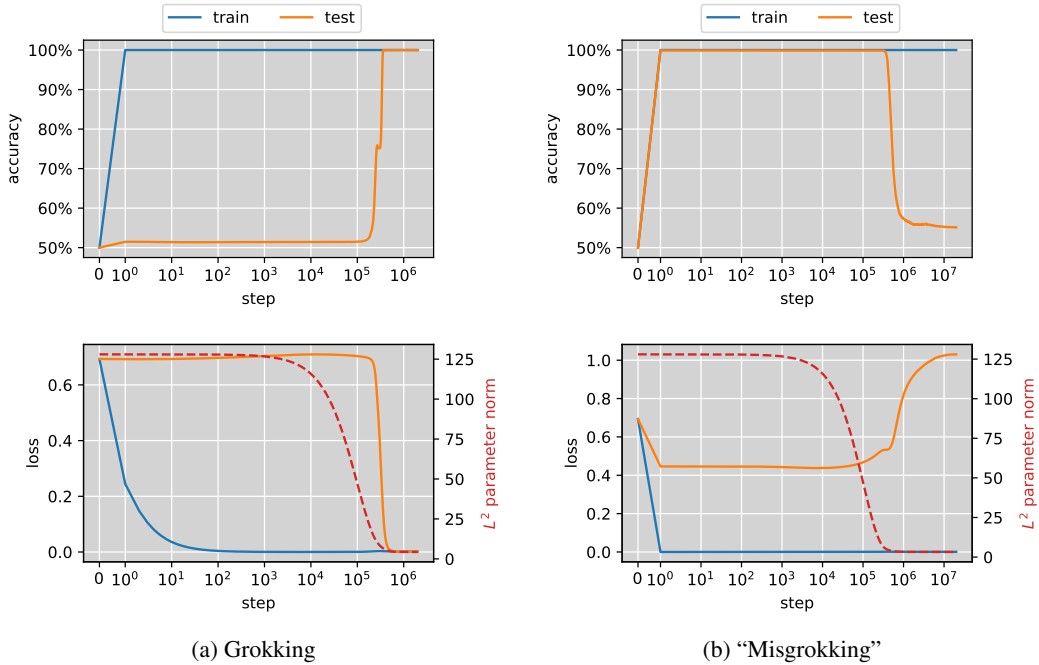

Figure 1: Training two-layer diagonal linear nets for linear classification can exhibit (a) grokking, for a dataset that can be linearly separated by a 3-sparse weight vector, or (b) "misgrokking", for a dataset that can be linearly separated by a large $L^2$-margin. See Section 3.2.3 for details.

**Our Contributions.** In this work, we address these limitations by identifying simple yet insightful theoretical setups where grokking with sharp transition can be rigorously proved and its mechanism can be intuitively understood. Our main intuition is that optimizers may implicitly induce different biases in early and late phases; grokking happens if the early phase bias implies an overfitting solution and late phase bias implies a generalizable solution.

More specifically, we focus on neural nets with large initialization and small weight decay. This is inspired by a recent work (Liu et al., 2023) showing that training with these two tricks can induce grokking on many tasks, even beyond learning modular arithmetic. Our theoretical analysis attributes this to the following *dichotomy* of early and late phase implicit biases: the large initialization induces a very strong early phase implicit bias towards kernel predictors, but over time, it decays and competes with a late phase implicit bias towards min-norm/max-margin predictors induced by the small weight decay, resulting in a transition that turns out to be provably sharp in between.

This implicit bias result holds for homogeneous neural nets, a broad class of neural nets that include commonly used MLPs and CNNs with homogeneous activation. We further exemplify this dichotomy in sparse linear classification with diagonal linear nets and low-rank matrix completion with over-parameterized models, where we provide theory and experiments showing that the kernel predictor does not generalize well but the min-norm/max-margin predictors do, hence grokking happens (see Figures 1a and 3).

Based on these theoretical insights, we are able to construct examples where the early phase implicit bias leads to a generalizable solution, and the late phase bias leads to overfitting. This gives rise to a new phenomenon which we call "*misgrokking*": a neural net first achieves perfect training and test accuracies, but training for a longer time leads to a sudden big drop in test accuracy. See Figure 1b.

Our proof of the kernel regime in the early phase is related to the Neural Tangent Kernel (NTK) regime studied in the literature (Jacot et al., 2018; Chizat et al., 2019). However, our result is qualitatively different from existing NTK analyses in that in our setting, the weight norm changes significantly (due to weight decay), while in the usual NTK regime, the weight changes only by a small amount, and so does its norm. Our proof relies on a careful analysis of the norm and direction of the weight, which enables us to obtain a tight bound on the time spent in the kernel regime. This novel result of a kernel regime in homogeneous neural nets under weight decay may be of independent interest. We further analyze the late phase using a gradient convergence argument, and it turns out that a slightly longer time suffices for convergence to the max-margin/min-norm predictor, thus exhibiting a sharp transition between the two phases.

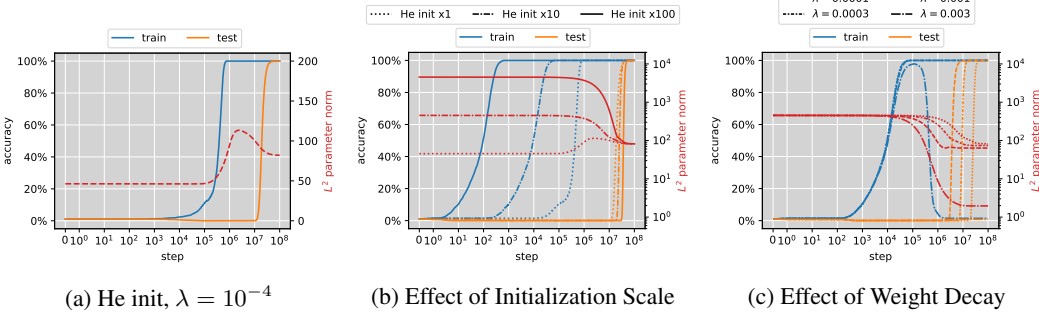

(a) He init, $\lambda = 10^{-4}$    (b) Effect of Initialization Scale    (c) Effect of Weight Decay

Figure 2: Training two-layer ReLU nets for modular addition exhibits grokking with He initialization and weight decay. Enlarging the initialization scale or reducing the weight decay delays the sharp transition in test accuracy.

Concretely, our contributions can be summarized as follows:

1. For training homogeneous neural nets with large initialization and small weight decay on classification and regression tasks, we prove a very sharp transition from optimal solutions of kernel SVM/regression to KKT solutions of global max-margin/min-norm problems associated with the neural net.

2. For classification and regression tasks respectively, we provide concrete examples with diagonal linear nets and overparameterized matrix completion, showing either grokking or misgrokking.

## 2 MOTIVATING EXPERIMENT: GROKKING IN MODULAR ADDITION

In this section, we provide experiments to show that the initialization scale and weight decay are important factors in the grokking phenomenon, in the sense that enlarging the initialization scale or reducing the weight decay delays the sharp transition in test accuracy.

**Grokking in Modular Addition.** Following many previous works (Nanda et al., 2023; Gromov, 2023), we focus on *modular addition*, which belongs to the modular arithmetic tasks where the grokking phenomenon was first observed (Power et al., 2022). The task is to learn the addition operation over $\mathbb{Z}_p$: randomly split $\{(a, b, c) : a + b \equiv c \pmod{p}\}$ into training and test sets, and train a neural net on the training set to predict $c$ given input pair $(a, b)$. The grokking phenomenon has been observed in learning this task under many training regimes, including training transformers and MLPs on cross-entropy and squared loss with various optimizers (full-batch GD, SGD, Adam, AdamW, etc.) (Power et al., 2022; Liu et al., 2022; Thilak et al., 2022; Gromov, 2023). For simplicity, we focus on a simple setting without involving too many confounding factors: training a two-layer ReLU net with full-batch GD. More specifically, we represent the input $(a, b) \in \mathbb{Z}_p \times \mathbb{Z}_p$ as the concatenation of the one-hot representations of $a$ and $b$, resulting in an input vector $\boldsymbol{x} \in \mathbb{R}^{2p}$, and then the neural net processes the input as follows:

$$f(\boldsymbol{\theta}; \boldsymbol{x}) = \boldsymbol{W}_2 \operatorname{ReLU}(\boldsymbol{W}_1 \boldsymbol{x} + \boldsymbol{b}_1), \tag{1}$$

where $\boldsymbol{\theta} = (\boldsymbol{W}_1, \boldsymbol{b}_1, \boldsymbol{W}_2)$ is the parameter vector consisting of $\boldsymbol{W}_1 \in \mathbb{R}^{h \times 2p}, \boldsymbol{b}_1 \in \mathbb{R}^h, \boldsymbol{W}_2 \in \mathbb{R}^{p \times h}$, $h$ is the width, and $\operatorname{ReLU}(\boldsymbol{v})$ stands for the ReLU activation that maps each entry $v_i$ of a vector $\boldsymbol{v}$ to $\max\{v_i, 0\}$. The neural net output $f(\boldsymbol{\theta}; \boldsymbol{x}) \in \mathbb{R}^p$ is treated as logits and is fed into the standard cross-entropy loss in training. We set the modulus $p$ as 97, the training data size as 40% of the total number of data ($p^2$), the width $h$ as 1024, the learning rate as 0.002 and weight decay as $10^{-4}$. Figure 2a successfully reproduces the grokking phenomenon: the training accuracy reaches 100% in $10^6$ steps while the test accuracy remains close to random guessing, but after training for one additional order of magnitude in the number of steps, the test accuracy suddenly jumps to 100%.

A natural question is how this grokking phenomenon depends on the training tricks in the pipeline. Liu et al. (2023) showed that, empirically, using a large initialization scale and a small but non-zero weight decay can induce grokking on various tasks even beyond modular arithmetic, including image classification on MNIST and sentiment classification on IMDB. Indeed, our experiments on modular addition can confirm the importance of initialization scale and weight decay.

**Effect of Initialization Scale.** To study the effect of initialization, we scale up the standard He initialization (He et al., 2015) by a factor of $\alpha > 0$, and then run the same training procedure. However, the model at large initialization produces a very large random number for each training

sample, which induces training instability. To remove this confounding factor, we set the weights $\boldsymbol{W}_2$ in the second layer to $\boldsymbol{0}$ so that the initial model output is always 0. Results in Figure 2b show that the sharp transition in test accuracy is delayed when $\alpha$ increases from 1 to 100.

**Effect of Weight Decay.** In Figure 2c, we run experiments with different weight decay values. We observe that increasing weight decay makes the transition happen earlier. However, when weight decay surpasses a small threshold, the regularization strength can become so strong that it even hinders the optimization of the training loss and causes the training accuracy to collapse eventually. This collapse due to weight decay is also consistent with the observation in Lewkowycz and Gur-Ari (2020). Conversely, as weight decay decreases, the grokking phenomenon becomes more significant because now it takes more time for the test accuracy to exhibit the sharp transition. But the weight decay has to be non-zero: we will discuss in Appendix B.1 that training without weight decay leads to perfect generalization in the end, but the transition in test accuracy is no longer sharp.

**Explanation?** Liu et al. (2023) built their intuition upon an empirical observation from Fort and Scherlis (2019): there is a narrow range of weight norm, called *Goldilocks zone*, where the generalization inside the zone is better than the outside. It is argued that the weight norm can take a longer time to enter this Goldilocks zone when training with a larger initial weight norm or smaller weight decay, hence causing the delayed generalization. However, Fort and Scherlis (2019); Liu et al. (2023) did not provide much explanation on why such a narrow Goldilocks zone could exist in the first place. When talking about MLPs and CNNs with ReLU activation, which are commonly used in practice, this becomes even more mysterious: the output functions $f(\boldsymbol{\theta}; \boldsymbol{x})$ of these neural nets are homogeneous to their parameters, i.e., $f(c\boldsymbol{\theta}; \boldsymbol{x}) = c^L f(\boldsymbol{\theta}; \boldsymbol{x})$ for all $c > 0$ (Lyu and Li, 2020; Ji and Telgarsky, 2020a; Kunin et al., 2023). As a result, it is impossible to explain the effect of norm just from the expressive power, since all classifiers that can be represented with a certain norm can also be represented by all other norms. This motivates us to dive deep into the training dynamics in grokking via rigorous theoretical analysis.

# 3 GROKKING WITH LARGE INITIALIZATION AND SMALL WEIGHT DECAY

In this section, we present our theory on homogeneous neural nets with large initialization and small weight decay, which attributes grokking to a dichotomy of early and late phase implicit biases.

## 3.1 THEORETICAL SETUP

We focus on training models for classification and regression. Let $\mathcal{D}_{\mathrm{X}}$ be the input distribution. For every input $\boldsymbol{x} \sim \mathcal{D}_{\mathrm{X}}$, let $y^*(\boldsymbol{x})$ be the classification/regression target of $\boldsymbol{x}$. We parameterize the model with $\boldsymbol{\theta}$ and use $f(\boldsymbol{\theta}; \boldsymbol{x})$ to denote the model output on input $\boldsymbol{x}$. We train the model by optimizing an empirical loss on a dataset $\{(\boldsymbol{x}_i, y_i)\}_{i=1}^n$ where the data points are drawn i.i.d. as $\boldsymbol{x}_i \sim \mathcal{D}_{\mathrm{X}}, y_i = y^*(\boldsymbol{x}_i)$. For each $i \in [n]$, we write $f_i(\boldsymbol{\theta}) := f(\boldsymbol{\theta}; \boldsymbol{x}_i)$ for short. We assume the model is *homogeneous* with respect to its parameters $\boldsymbol{\theta}$, a common assumption in analyzing neural networks (Lyu and Li, 2020; Ji and Telgarsky, 2020a; Telgarsky, 2023; Woodworth et al., 2020).

**Assumption 3.1** *For all $\boldsymbol{x}$, $f(\boldsymbol{\theta}; \boldsymbol{x})$ is $L$-homogeneous (i.e., $f_i(c\boldsymbol{\theta}) = c^L f_i(\boldsymbol{\theta})$ for all $c > 0$) and $\mathcal{C}^2$-smooth with respect to $\boldsymbol{\theta}$.*

The two-layer net in our motivating experiment and other MLPs and CNNs with ReLU activation are indeed $L$-homogeneous. However, we also need to assume the $\mathcal{C}^2$-smoothness to ease the definition and analysis of gradient flow, which excludes the use of any non-smooth activation. This can be potentially addressed by defining and analyzing gradient flow via Clarke's subdifferential (Clarke, 1975; 2001; Lyu and Li, 2020; Ji and Telgarsky, 2020a), but we choose to make this assumption for simplicity. Note that our concrete examples of grokking on sparse linear classification and matrix completion indeed satisfy this assumption.

As motivated in Section 2, we consider training with large initialization and small weight decay. Mathematically, we start training from $\alpha \bar{\boldsymbol{\theta}}_{\mathrm{init}}$, where $\bar{\boldsymbol{\theta}}_{\mathrm{init}} \in \mathbb{R}^d$ is a fixed vector and $\alpha$ is a large factor controlling the initialization scale. We also assume a small but non-zero weight decay $\lambda > 0$. To study the asymptotics more conveniently, we regard $\lambda$ as a function of $\alpha$ with order $\lambda(\alpha) = \Theta(\alpha^{-p})$ for some positive $p = \Theta(1)$.

To avoid loss explosion at initialization, we deterministically set or randomly sample $\bar{\boldsymbol{\theta}}_{\mathrm{init}}$ in a way that the initial output of the model is zero. This can be done by setting the last layer weights to zero

(same as our experiments in Section 2), using the symmetrized initialization in Chizat et al. (2019), or using the "difference trick" in Hu et al. (2020). This zero-output initialization is also used in many previous studies of implicit bias (Chizat et al., 2019; Woodworth et al., 2020; Moroshko et al., 2020) for the sake of mathematical simplicity. We note that our analysis should be easily extended to random initialization with small outputs, just as in Chizat et al. (2019); Arora et al. (2019b).

**Assumption 3.2** $f(\bar{\boldsymbol{\theta}}_{\text{init}}; \boldsymbol{x}) = 0$ *for all* $\boldsymbol{x}$.

We define the vanilla training loss as the average over the loss values of each individual sample:

$$\mathcal{L}(\boldsymbol{\theta}) = \frac{1}{n} \sum_{i=1}^{n} \ell(f_i(\boldsymbol{\theta}); y_i). \tag{2}$$

With weight decay $\lambda$, it becomes the following regularized training loss:

$$\mathcal{L}_\lambda(\boldsymbol{\theta}) = \mathcal{L}(\boldsymbol{\theta}) + \frac{\lambda}{2}\|\boldsymbol{\theta}\|_2^2. \tag{3}$$

For simplicity, in this paper, we consider minimizing the loss via gradient flow $\frac{\mathrm{d}\boldsymbol{\theta}}{\mathrm{d}t} = -\nabla\mathcal{L}_\lambda(\boldsymbol{\theta})$, which is the continuous counterpart of gradient descent and stochastic gradient descent when the learning rate goes to $0$. We use $\boldsymbol{\theta}(t; \boldsymbol{\theta}_0)$ to denote the parameter at time $t$ by running gradient flow from $\boldsymbol{\theta}_0$, and we write $\boldsymbol{\theta}(t)$ when the initialization $\boldsymbol{\theta}_0$ is clear from the context.

### 3.2 GROKKING IN CLASSIFICATION

In this subsection, we study the grokking phenomenon on classification tasks. For simplicity, we restrict to binary classification problems, where $y^*(\boldsymbol{x}) \in \{-1, +1\}$. While in practice it is standard to use the logistic loss $\ell(\hat{y}, y) = \log(1 + e^{-y\hat{y}})$ (a.k.a. binary cross-entropy loss), we follow the implicit bias literature to analyze the exponential loss $\ell(\hat{y}; y) = e^{-y\hat{y}}$ as a simple surrogate, which has the same tail behavior as the logistic loss when $y\hat{y} \to +\infty$ and thus usually leads to the same implicit bias (Soudry et al., 2018; Nacson et al., 2019b; Lyu and Li, 2020; Chizat and Bach, 2020).

By carefully analyzing the training dynamics, we rigorously prove that there is a sharp transition around $\frac{1}{\lambda} \log \alpha$. Before this point, gradient flow fits the training data perfectly while maintaining the parameter direction within a local region near the initial direction, which causes the classifier to behave like a kernel classifier based on Neural Tangent Kernel (NTK) (Jacot et al., 2018; Arora et al., 2019b;c). After the transition, however, the gradient flow escapes the local region and makes efforts to maximize the margin. Following the nomenclature in (Moroshko et al., 2020), we call the first regime the *kernel regime* and the second regime the *rich regime*. The key difference to the existing works analyzing the kernel and rich regimes (Moroshko et al., 2020; Telgarsky, 2023) is that the transition in our case is provably sharp, which is a crucial ingredient for the grokking phenomenon.

#### 3.2.1 KERNEL REGIME

First, we show that gradient flow gets stuck in the kernel regime over the initial period of $\frac{1-c}{\lambda} \log \alpha$, where the model behaves as if it were optimizing over a linearized model. More specifically, for all $\boldsymbol{x}$, we define $\nabla f(\bar{\boldsymbol{\theta}}_{\text{init}}; \boldsymbol{x})$ as the NTK feature of $\boldsymbol{x}$. As we are considering over-parameterized models, where the dimension of $\boldsymbol{\theta}$ is larger than the number of data $n$, it is natural to assume that the NTK features of the training data are linearly separable (Ji and Telgarsky, 2020b; Telgarsky, 2023):

**Assumption 3.3** *There exists* $\boldsymbol{h}$ *such that* $y_i \langle \nabla f_i(\bar{\boldsymbol{\theta}}_{\text{init}}), \boldsymbol{h} \rangle > 0$ *for all* $i \in [n]$.

Let $\boldsymbol{h}_{\text{ntk}}^*$ be the unit vector so that a linear classifier with weight $\boldsymbol{h}_{\text{ntk}}^*$ can attain the max $L^2$-margin on the NTK features of the training data, $\{(\nabla f_i(\bar{\boldsymbol{\theta}}_{\text{init}}), y_i)\}_{i=1}^{n}$, and $\gamma_{\text{ntk}}$ be the corresponding $L^2$-margin. That is, $\boldsymbol{h}_{\text{ntk}}^*$ is the unique unit vector that points to the direction of the unique optimal solution to the following constrained optimization problem:

$$\min \quad \frac{1}{2}\|\boldsymbol{h}\|_2^2 \quad \text{s.t.} \quad y_i \langle \nabla f_i(\bar{\boldsymbol{\theta}}_{\text{init}}), \boldsymbol{h} \rangle \geq 1, \quad \forall i \in [n], \tag{K1}$$

and $\gamma_{\text{ntk}} := \min_{i \in [n]} \{y_i \langle \nabla f_i(\bar{\boldsymbol{\theta}}_{\text{init}}), \boldsymbol{h}_{\text{ntk}}^* \rangle\}$.

The following theorem states that for any $c \in (0, 1)$, the solution found by gradient flow at time $\frac{1-c}{\lambda} \log \alpha$ represents the same classifier as the max $L^2$-margin linear classifier on the NTK features:

**Theorem 3.4** *For any all constants $c \in (0, 1)$, letting $T_c^-(\alpha) := \frac{1-c}{\lambda} \log \alpha$, it holds that*

$$\forall \boldsymbol{x} \in \mathbb{R}^d : \qquad \lim_{\alpha \to +\infty} \frac{1}{Z(\alpha)} f(\boldsymbol{\theta}(T_c^-(\alpha); \alpha \bar{\boldsymbol{\theta}}_{\text{init}}); \boldsymbol{x}) = \left\langle \nabla f(\bar{\boldsymbol{\theta}}_{\text{init}}; \boldsymbol{x}), \boldsymbol{h}_{\text{ntk}}^* \right\rangle,$$

*where $Z(\alpha) := \frac{1}{\gamma_{\text{ntk}}} \log \frac{\alpha^c}{\lambda}$ is a normalizing factor.*

**Key Proof Insight.** The standard NTK-based dynamical analysis requires $\boldsymbol{\theta}(t; \alpha \bar{\boldsymbol{\theta}}_{\text{init}}) \approx \alpha \bar{\boldsymbol{\theta}}_{\text{init}}$, but the weight decay in our case can significantly change the norm, hence violating this condition. The key ingredient of our proof is a much more careful dynamical analysis showing that $\boldsymbol{\theta}(t; \alpha \bar{\boldsymbol{\theta}}_{\text{init}}) \approx \alpha e^{-\lambda t} \bar{\boldsymbol{\theta}}_{\text{init}}$, i.e., the norm of $\boldsymbol{\theta}(t; \alpha \bar{\boldsymbol{\theta}}_{\text{init}})$ decays but the direction remains close to $\bar{\boldsymbol{\theta}}_{\text{init}}$. Then by $L$-homogeneity and Taylor expansion, we have

$$f(\boldsymbol{\theta}; \boldsymbol{x}) = \left(\alpha e^{-\lambda t}\right)^L f(\tfrac{e^{\lambda t}}{\alpha} \boldsymbol{\theta}; \boldsymbol{x}) \approx \left(\alpha e^{-\lambda t}\right)^L \left( f(\bar{\boldsymbol{\theta}}_{\text{init}}; \boldsymbol{x}) + \left\langle \nabla f(\bar{\boldsymbol{\theta}}_{\text{init}}; \boldsymbol{x}), \tfrac{e^{\lambda t}}{\alpha} \boldsymbol{\theta} - \bar{\boldsymbol{\theta}}_{\text{init}} \right\rangle \right)$$

$$= \alpha^L e^{-L\lambda t} \left\langle \nabla f(\bar{\boldsymbol{\theta}}_{\text{init}}; \boldsymbol{x}), \tfrac{e^{\lambda t}}{\alpha} \boldsymbol{\theta} - \bar{\boldsymbol{\theta}}_{\text{init}} \right\rangle.$$

The large scaling factor $\alpha^L e^{-L\lambda t}$ above enables fitting the dataset even when the change in direction $\frac{e^{\lambda t}}{\alpha} \boldsymbol{\theta} - \bar{\boldsymbol{\theta}}_{\text{init}}$ is very small. Indeed, we show that $\frac{e^{\lambda t}}{\alpha} \boldsymbol{\theta} - \bar{\boldsymbol{\theta}}_{\text{init}} \approx \frac{1}{\gamma_{\text{ntk}}} \alpha^L e^{-L\lambda t} \log \frac{\alpha^c}{\lambda} \boldsymbol{h}_{\text{ntk}}^*$ at time $\frac{1-c}{\lambda} \log \alpha$ by closely tracking the dynamics, hence completing the proof. See Appendix C.2 for details. When $\alpha^L e^{-L\lambda t}$ is no longer a large scaling factor, namely $t = \frac{1}{\lambda}(\log \alpha + \omega(1))$, this analysis breaks and the dynamics enter the rich regime.

### 3.2.2 RICH REGIME

Next, we show that continuing the gradient flow for a slightly long time to time $\frac{1+c}{\lambda} \log \alpha$, it is able to escape the kernel regime. More specifically, consider the following constrained optimization problem that aims at maximizing the margin of the predictor $f(\boldsymbol{\theta}; \boldsymbol{x})$ on the training data $\{(\boldsymbol{x}_i, y_i)\}_{i=1}^n$:

$$\min \quad \frac{1}{2} \|\boldsymbol{\theta}\|_2^2 \quad \text{s.t.} \quad y_i f_i(\boldsymbol{\theta}) \geq 1, \quad \forall i \in [n]. \tag{R1}$$

Then we have the following directional convergence result:

**Theorem 3.5** *For all constants $c > 0$ and for every sequence $\{\alpha_k\}_{k \geq 1}$ with $\alpha_k \to +\infty$, letting $T_c^+(\alpha) := \frac{1+c}{\lambda} \log \alpha$, there exists a time sequence $\{t_k\}_{k \geq 1}$ such that $\frac{1}{\lambda} \log \alpha_k \leq t_k \leq T_c^+(\alpha_k)$ and every limit point of $\left\{ \frac{\boldsymbol{\theta}(t_k; \alpha_k \bar{\boldsymbol{\theta}}_{\text{init}})}{\|\boldsymbol{\theta}(t_k; \alpha_k \bar{\boldsymbol{\theta}}_{\text{init}})\|_2} : k \geq 1 \right\}$ is along the direction of a KKT point of (R1).*

For the problem (R1), the KKT condition is a first-order necessary condition for global optimality, but it is not sufficient in general since $f_i(\boldsymbol{\theta})$ can be highly non-convex. Nonetheless, since gradient flow can easily get trapped at spurious minima with only first-order information, the KKT condition is widely adopted as a surrogate for global optimality in theoretical analysis (Lyu and Li, 2020; Wang et al., 2021; Kunin et al., 2023).

**Key Proof Insight.** The key is to use the loss convergence and norm decay bounds from the kernel regime as a starting point to obtain a small upper bound for the gradient $\nabla \mathcal{L}_\lambda(\boldsymbol{\theta})$. Then we connect the gradient upper bounds with KKT conditions. See Appendix C.3 for the proof.

### 3.2.3 EXAMPLE: LINEAR CLASSIFICATION WITH DIAGONAL LINEAR NETS

Now, we exemplify how our implicit bias results imply a sharp transition in test accuracy in a concrete setting: training diagonal linear nets for linear classification. Let $\mathcal{D}_X$ be a distribution over $\mathbb{R}^d$ and $y^*(\boldsymbol{x}) = \text{sign}(\langle \boldsymbol{w}^*, \boldsymbol{x} \rangle) \in \{\pm 1\}$ be the ground-truth target for binary classification, where $\boldsymbol{w}^* \in \mathbb{R}^d$ is an unknown vector. Following Moroshko et al. (2020), we consider training a so-called two-layer "diagonal" linear net to learn $y^*$. On input $\boldsymbol{x} \in \mathbb{R}^d$, the model outputs the following function $f(\boldsymbol{\theta}; \boldsymbol{x})$, where $\boldsymbol{\theta} := (\boldsymbol{u}, \boldsymbol{v}) \in \mathbb{R}^d \times \mathbb{R}^d \simeq \mathbb{R}^{2d}$ is the model parameter.

$$f(\boldsymbol{\theta}; \boldsymbol{x}) = \sum_{k=1}^d u_k^2 x_k - \sum_{k=1}^d v_k^2 x_k. \tag{4}$$

This model is considered a diagonal net because it can be seen as a sparsely-connected two-layer net with $2d$ hidden neurons, each of which only inputs and outputs a single scalar. More specifically, the input $\boldsymbol{x}$ is first expanded to an $2d$-dimensional vector $(\boldsymbol{x}, -\boldsymbol{x})$. Then the $k$-th hidden neuron

in the first half inputs $x_k$ and outputs $u_k x_k$, and the $k$-th hidden neuron in the second half inputs $-x_k$ and outputs $v_k x_k$. The output neuron in the second layer shares the same weights as the hidden neurons and takes a weighted sum over the hidden neuron outputs according to their weights: $f(\boldsymbol{\theta}; \boldsymbol{x}) = \sum_{k=1}^{d} u_k \cdot (u_k \cdot x_k) + \sum_{k=1}^{d} v_k \cdot (v_k \cdot (-x_k))$. Note that the model output is always linear with the input. For convenience, we can write $f(\boldsymbol{\theta}; \boldsymbol{x}) = \langle \boldsymbol{w}(\boldsymbol{\theta}), \boldsymbol{x} \rangle$ with effective weight $\boldsymbol{w}(\boldsymbol{\theta}) := \boldsymbol{u}^{\odot 2} - \boldsymbol{v}^{\odot 2}$.

In this setup, we can understand the early and late phase implicit biases very concretely. We start training from a large initialization: $\boldsymbol{\theta}(0) = \alpha \bar{\boldsymbol{\theta}}_{\text{init}}$, where $\alpha > 0$ controls the initialization scale and $\bar{\boldsymbol{\theta}}_{\text{init}} = (\mathbf{1}, \mathbf{1})$. That is, $u_k = v_k = \alpha$ for all $1 \leq k \leq d$ at $t = 0$. As noted in Moroshko et al. (2020), the kernel feature for each data point is $\nabla f(\bar{\boldsymbol{\theta}}_{\text{init}}; \boldsymbol{x}) = (2\boldsymbol{x}, -2\boldsymbol{x})$, so the max $L^2$-margin kernel classifier is the max $L^2$-margin linear classifier. This leads to the following corollary:

**Corollary 3.6 (Diagonal linear nets, kernel regime)** *Let $T_c^-(\alpha) := \frac{1-c}{\lambda} \log \alpha$. For all constants $c \in (0, 1)$, as $\alpha \to +\infty$, the normalized effective weight vector $\frac{\boldsymbol{w}(\boldsymbol{\theta}(T_c^-(\alpha); \alpha \bar{\boldsymbol{\theta}}_{\text{init}}))}{\|\boldsymbol{w}(\boldsymbol{\theta}(T_c^-(\alpha); \alpha \bar{\boldsymbol{\theta}}_{\text{init}}))\|_2}$ converges to the max $L^2$-margin direction, namely $\boldsymbol{w}_2^* := \arg\max_{\boldsymbol{w} \in \mathcal{S}^{d-1}} \{\min_{i \in [n]} y_i \langle \boldsymbol{w}, \boldsymbol{x}_i \rangle\}$.*

In contrast, if we train slightly longer from $T_c^-(\alpha)$ to $T_c^+(\alpha)$, we can obtain a KKT point of (R1).

**Corollary 3.7 (Diagonal linear nets, rich regime)** *Let $T_c^+(\alpha) := \frac{1+c}{\lambda} \log \alpha$. For all constants $c \in (0, 1)$, and for every sequence $\{\alpha_k\}_{k \geq 1}$ with $\alpha_k \to +\infty$, there exists a time sequence $\{t_k\}_{k \geq 1}$ such that $\frac{1}{\lambda} \log \alpha_k \leq t_k \leq T_c^+(\alpha_k)$ and every limit point of $\left\{ \frac{\boldsymbol{\theta}(t_k; \alpha_k \bar{\boldsymbol{\theta}}_{\text{init}})}{\|\boldsymbol{\theta}(t_k; \alpha_k \bar{\boldsymbol{\theta}}_{\text{init}})\|_2} : k \geq 1 \right\}$ is along the direction of a KKT point of the problem $\min \|\boldsymbol{u}\|_2^2 + \|\boldsymbol{v}\|_2^2$ s.t. $y_i \langle \boldsymbol{u}^{\odot 2} - \boldsymbol{v}^{\odot 2}, \boldsymbol{x}_i \rangle \geq 1$.*

As noted in these two works, in the diagonal linear case, optimizing (R1) to the global optimum is equivalent to finding the max $L^1$-margin linear classifier: $\min \|\boldsymbol{w}\|_1$ s.t. $y_i \langle \boldsymbol{w}, \boldsymbol{x}_i \rangle \geq 1, \forall i \in [n]$, or equivalently, $\boldsymbol{w}_1^* := \arg\max_{\boldsymbol{w} \in \mathcal{S}^{d-1}} \{\min_{i \in [n]} y_i \langle \boldsymbol{w}, \boldsymbol{x}_i \rangle\}$. Therefore, our theory suggests a sharp transition at time $\frac{1}{\lambda} \log \alpha$ from max $L^2$-margin linear classifier to max $L^1$-margin linear classifier.

While this corollary only shows KKT conditions rather than global optimality, we believe that one may be able to obtain the global optimality using insights from existing analysis of training diagonal linear nets without weight decay (Gunasekar et al., 2018b; Moroshko et al., 2020).

**Empirical Validation: Grokking.** As it is well known that maximizing the $L^1$-margin can better encourage sparsity than maximizing the $L^2$-margin, our theory predicts that the grokking phenomenon can be observed in training diagonal linear nets for $k$-sparse linear classification. To verify this, we specifically consider the following task: sample $n$ data points uniformly from $\{\pm 1\}^d$ for a very large $d$, and let the ground truth be a linear classifier with a $k$-sparse weight, where the non-zero coordinates are sampled uniformly from $\{\pm 1\}^k$. Applying standard generalization bounds based on Rademacher complexity can show that the max $L^2$-margin linear classifier needs $\mathcal{O}(kd)$ to generalize, while the max $L^1$-margin linear classifier only needs $\mathcal{O}(k^2 \log d)$. See Appendix E.2 for the proof. This suggests that the grokking phenomenon can be observed in this setting. In Figure 1a, we run experiments with $n = 256, d = 10^5, k = 3$, large initialization with initial parameter norm $128$, and small weight decay $\lambda = 0.001$. As expected, we observe the grokking phenomenon: in the early phase, the net fits the training set very quickly but fails to generalize; after a sharp transition in the late phase, the test accuracy becomes $100\%$.

**Empirical Validation: Misgrokking.** From the above example, we can see that grokking can be induced by the mismatch between the early-phase implicit bias and data and a good match between the late-phase implicit bias and data. However, this match and mismatch can largely depend on data. Conversely, if we consider the case where the labels in the linear classification problem are generated by a linear classifier with a large $L^2$-margin, the early phase implicit bias can make the neural net generalize easily, but the late phase implicit bias can destroy this good generalization since the ground-truth weight vector may not have a large $L^1$-margin. To justify this, we first sample a unit-norm vector $\boldsymbol{w}^*$, then sample a dataset from the distribution $(\boldsymbol{x}, y) \sim (\boldsymbol{z} + \frac{\gamma}{2} \text{sign}(\langle \boldsymbol{z}, \boldsymbol{w}^* \rangle) \boldsymbol{w}^*, \text{sign}(\langle \boldsymbol{z}, \boldsymbol{w}^* \rangle))$, where $\boldsymbol{z} \sim \mathcal{N}(\mathbf{0}, \boldsymbol{I}_d)$ (i.e., a Gaussian distribution that is separated by a margin in the middle). We take $\gamma = 25$ and sample $n = 32$ points. Indeed, we observe a phenomenon which we call "*misgrokking*": the neural net first fits the training set and achieves $100\%$ test accuracy, and then after training for sufficiently longer, the test accuracy drops to nearly $50\%$. See Figure 1b.

## 3.3 GROKKING IN REGRESSION

In this subsection, we study the grokking phenomenon on regression tasks with squared loss $\ell(\hat{y}, y) = (\hat{y} - y)^2$. Paralleling the classification setting, we analyze the behavior of gradient flow in both kernel and rich regimes, and show that there is a sharp transition around $\frac{1}{\lambda} \log \alpha$.

### 3.3.1 KERNEL REGIME

Similar to the classification setting, we first show that gradient flow gets stuck in the kernel regime over the initial period of $\frac{1-c}{\lambda} \log \alpha$ time. For all $\boldsymbol{x}$, we define $\nabla f(\bar{\boldsymbol{\theta}}_{\mathrm{init}}; \boldsymbol{x})$ as the NTK feature of $\boldsymbol{x}$. For over-parameterized models, where the dimension of $\boldsymbol{\theta}$ is larger than the number of data $n$, we make the following natural assumption that is also widely used in the literature (Du et al., 2019b; Chizat et al., 2019; Arora et al., 2019b):

**Assumption 3.8** *The NTK features of training samples* $\{\nabla f_i(\bar{\boldsymbol{\theta}}_{\mathrm{init}}; \boldsymbol{x})\}_{i=1}^n$ *are linearly independent.*

Now we let $\boldsymbol{h}_{\mathrm{ntk}}^*$ be the vector with minimum norm such that the linear predictor $\boldsymbol{g} \mapsto \langle \boldsymbol{g}, \boldsymbol{h} \rangle$ perfectly fits $\{(\nabla f_i(\bar{\boldsymbol{\theta}}_{\mathrm{init}}), y_i)\}_{i=1}^n$. That is, $\boldsymbol{h}_{\mathrm{ntk}}^*$ is the solution to the following constrained optimization problem:

$$\min \quad \frac{1}{2}\|\boldsymbol{h}\|_2^2 \quad \text{s.t.} \quad \left\langle \nabla f_i(\bar{\boldsymbol{\theta}}_{\mathrm{init}}), \boldsymbol{h} \right\rangle = y_i, \quad \forall i \in [n]. \tag{K2}$$

Then we have the following result that is analogous to Theorem 3.4. See Appendix C.2 for the proof.

**Theorem 3.9** *For all constants* $c \in (0, 1)$, *letting* $T_{\mathrm{c}}^-(\alpha) := \frac{1-c}{\lambda} \log \alpha$, *it holds that*

$$\forall \boldsymbol{x} \in \mathbb{R}^d: \qquad \lim_{\alpha \to +\infty} f(\boldsymbol{\theta}(T_{\mathrm{c}}^-(\alpha); \alpha\bar{\boldsymbol{\theta}}_{\mathrm{init}}); \boldsymbol{x}) = \left\langle \nabla f(\bar{\boldsymbol{\theta}}_{\mathrm{init}}; \boldsymbol{x}), \boldsymbol{h}_{\mathrm{ntk}}^* \right\rangle.$$

### 3.3.2 RICH REGIME

Similar to the classification setting, we then show that gradient flow is able to escape the kernel regime at time $\frac{1+c}{\lambda} \log \alpha$. Specifically, consider the following constrained optimization problem that searches for the parameter with the minimum norm that can perfectly fit the training data:

$$\min \quad \frac{1}{2}\|\boldsymbol{\theta}\|_2^2 \quad \text{s.t.} \quad f_i(\boldsymbol{\theta}) = y_i, \quad \forall i \in [n]. \tag{R2}$$

Then we have the following convergence result. See Appendix D.2 for the proof.

**Theorem 3.10** *For any constant* $c > 0$, *letting* $T_{\mathrm{c}}^+(\alpha) := \frac{1+c}{\lambda} \log \alpha$, *for any sequence of* $\{\alpha_k\}_{k \geq 1}$ *with* $\alpha_k \to +\infty$, *there exists a time sequence* $\{t_k\}_{k \geq 1}$ *satisfying* $\frac{1}{\lambda} \log \alpha_k \leq t_k \leq T_{\mathrm{c}}^+(\alpha_k)$, *such that* $\|\boldsymbol{\theta}(t_k; \alpha_k\bar{\boldsymbol{\theta}}_{\mathrm{init}})\|_2$ *are uniformly bounded and that every limit point of* $\{\boldsymbol{\theta}(t_k; \alpha_k\bar{\boldsymbol{\theta}}_{\mathrm{init}}) : k \geq 1\}$ *is a KKT point of* (R2).

### 3.3.3 EXAMPLE: OVERPARAMETERIZED MATRIX COMPLETION

As an example, we consider the matrix completion problem of recovering a low-rank symmetric matrix based on partial observations of its entries. Specifically, we let $\boldsymbol{X}^* \in \mathbb{R}^{d \times d}$ be the ground-truth symmetric matrix with rank $r \ll d$ and we observe a (random) set of entries indexed by $\Omega \subseteq [d] \times [d]$. In this setting, the input distribution $\mathcal{D}_X$ is the uniform distribution on the finite set $[d] \times [d]$. For every input $\boldsymbol{x} = (i, j) \sim \mathcal{D}_X$, $y^*(\boldsymbol{x}) = \langle \boldsymbol{P}_{\boldsymbol{x}}, \boldsymbol{X}^* \rangle$ where $\boldsymbol{P}_x = \boldsymbol{e}_i \boldsymbol{e}_j^\top$.

Following previous works (Arora et al., 2019a; Razin and Cohen, 2020; Woodworth et al., 2020; Li et al., 2021), we solve the matrix completion problem using the matrix factorization approach, *i.e.*, we parameterize a matrix as $\boldsymbol{W} = \boldsymbol{U}\boldsymbol{U}^\top - \boldsymbol{V}\boldsymbol{V}^\top$, where $\boldsymbol{U}, \boldsymbol{V} \in \mathbb{R}^{d \times d}$, and optimize $\boldsymbol{U}, \boldsymbol{V}$ so that $\boldsymbol{W}$ matches with the ground truth on observed entries. This can be viewed a neural net that is parameterized by $\boldsymbol{\theta} = (\mathrm{vec}(\boldsymbol{U}), \mathrm{vec}(\boldsymbol{V})) \in \mathbb{R}^{2d^2}$ and outputs $f(\boldsymbol{\theta}; \boldsymbol{x}) = \langle \boldsymbol{P}_{\boldsymbol{x}}, \boldsymbol{W}(\boldsymbol{\theta}) \rangle$ given $\boldsymbol{x} = (i, j)$, where $\boldsymbol{W}(\boldsymbol{\theta}) := \boldsymbol{U}\boldsymbol{U}^\top - \boldsymbol{V}\boldsymbol{V}^\top$. It is easy to see that $f(\boldsymbol{\theta}; \boldsymbol{x})$ is 2-homogeneous, satisfying Assumption 3.1. Given a dataset $\{(\boldsymbol{x}_i, y_i)\}_{i=1}^n$, we consider using gradient flow to minimize the $\ell_2$-regularized square loss defined in (3) which can be equivalently written as

$$\mathcal{L}(\boldsymbol{\theta}) = \frac{1}{n} \sum_{i=1}^n (f(\boldsymbol{\theta}; \boldsymbol{x}_i) - y_i)^2 + \frac{\lambda}{2} \left( \|\boldsymbol{U}\|_F^2 + \|\boldsymbol{V}\|_F^2 \right). \tag{5}$$

In the kernel regime, Theorem 3.9 implies the following result for identity initialization.

**Corollary 3.11 (Matrix completion, kernel regime)** *Let $\bar{\theta}_{\text{init}} := (\text{vec}(\boldsymbol{I}), \text{vec}(\boldsymbol{I}))$. For all constants $c \in (0,1)$, letting $T_c^-(\alpha) = \frac{1-c}{\lambda} \log \alpha$, it holds for all $(i,j) \in [d] \times [d]$ that*

$$\lim_{\alpha \to +\infty} \left[ \boldsymbol{W}(\boldsymbol{\theta}(T_c^-(\alpha); \alpha\bar{\theta}_{\text{init}})) \right]_{ij} = \begin{cases} \boldsymbol{X}_{ij}^* & \text{if } (i,j) \in \Omega \text{ or } (j,i) \in \Omega, \\ 0 & \text{otherwise.} \end{cases}$$

In other words, in the kernel regime, while GD is able to fit the observed entries, it always fills the unobserved entries with $0$, leading to a significant failure in recovering the matrix.

In the rich regime, Theorem 3.10 implies that the solution transitions sharply near $\frac{1}{\lambda} \log \alpha$ to a KKT point of the norm minimization problem. If this KKT point is globally optimal, then it is known that $\boldsymbol{W}(\boldsymbol{\theta})$ is the minimum nuclear norm solution to the matrix completion problem: $\min \|\boldsymbol{W}\|_*$ s.t. $\boldsymbol{W} = \boldsymbol{W}^\top$, $\boldsymbol{W}_{ij} = \boldsymbol{X}_{ij}^*, \forall (i,j) \in \Omega$ (Ding et al., 2022, Theorem 3.2). It is known that the minimum nuclear norm solution can recover the ground truth with very small errors when $\tilde{\mathcal{O}}(d \log^2 d)$ entries are observed (Theorem F.5).

Although it is not proved that this KKT point is globally optimal, the following theorem provides positive evidence by showing that gradient flow eventually converges global minima and recovers the ground truth, if the time is not limited around $\frac{1}{\lambda} \log \alpha$.

**Theorem 3.12 (Matrix completion, rich regime)** *Suppose that $\text{rank}(\boldsymbol{X}^*) = r = \mathcal{O}(1)$ and $\boldsymbol{X}^* = \boldsymbol{V}_{\boldsymbol{X}^*} \boldsymbol{\Sigma}_{\boldsymbol{X}^*} \boldsymbol{V}_{\boldsymbol{X}^*}^\top$ is a SVD of $\boldsymbol{X}^*$, where each row of $\boldsymbol{V}_{\boldsymbol{X}^*}$ has $\ell_\infty$-norm bounded by $\sqrt{\frac{\mu}{d}}$. If the number of observed entries satisfies $N \gtrsim \mu^4 d \log^2 d$, then for any $\sigma > 0$, the gradient flow trajectory $(\boldsymbol{\theta}(t))_{t \geq 0}$ for (5) starting from random initialization $(\boldsymbol{U}(0))_{ij}, (\boldsymbol{V}(0))_{ij} \overset{i.i.d}{\sim} \mathcal{N}(\alpha \mathbb{1}\{i = j\}, \sigma^2)$ converges to a global minimizer $\boldsymbol{\theta}_\infty$ of $\mathcal{L}_\lambda$. Moreover, $\|\boldsymbol{W}(\boldsymbol{\theta}_\infty) - \boldsymbol{X}^*\|_F \lesssim \sqrt{\lambda \|\boldsymbol{X}^*\|_*} \mu^2 \log d$ with probability $\geq 1 - d^{-3}$.*

**Remark 3.13** *The small random perturbation to the identity initialization $(\alpha\boldsymbol{I}, \alpha\boldsymbol{I})$ in the above theorem is needed to guarantee that gradient flow does not get stuck at saddle points and thus converges to global minimizers almost surely. By the continuity of the gradient flow trajectory w.r.t. initialization, the conclusion of Corollary 3.11 still holds if $\sigma = \sigma(\alpha)$ is sufficiently small.*

**Empirical Validation.** We verify our theory on a simple algorithmic task: matrix completion for the multiplication table. Here, the multiplication table refers to a matrix $\boldsymbol{X}^*$ where the $(i,j)$-entry is $ij/d^2$. It is easy to see that this matrix is of rank-1 since $\boldsymbol{X}^* = \boldsymbol{u}\boldsymbol{u}^\top$ for $\boldsymbol{u} = (0, \frac{1}{d}, \frac{2}{d}, \ldots, \frac{d-1}{d})$. This means the late phase bias can complete the matrix with few observed entries; however, the early phase bias fills every unobserved entry with $0$, leading to a large test loss. This dichotomy of implicit biases implies grokking. To check this, we set $d = 97$ and randomly choose $5\%$ of the entries as the training set. We run GD with initialization scale $\alpha = 10$, learning rate $\eta = 0.1$, weight decay $10^{-4}$. Figure 3 shows that grokking indeed happens: GD takes $\sim 10^1$ steps to minimize the training loss, but the sharp transition in test loss occurs only after $\sim 10^6$ steps.

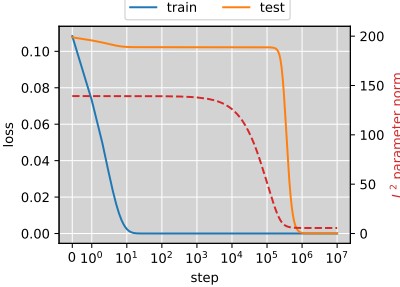

Figure 3: Matrix completion for the multiplication table exhibits grokking.

## 4 Conclusion

In this paper, we show that the grokking phenomenon provably occurs in several setups. Our results suggest that the sharp transition in test accuracy may stem from a dichotomy of implicit biases between the early and late training phases. While the early phase bias guides the optimizer to overfitting solutions, it is quickly corrected when the late phase bias takes effect.

Some limitations of our work are as follows. First, our work only studies the training dynamics with large initialization and small weight decay, but these may not be the only source of the dichotomy of the implicit biases. We further discuss in Appendix B that the late phase implicit bias can also be induced by implicit margin maximization and sharpness reduction, though the transition may not be as sharp as the weight decay case. Also, our work focuses on understanding the cause of grokking but does not study how to make neural nets generalize without so much delay in time. We leave it to future work to explore the other sources of grokking and practical methods to eliminate grokking.

ACKNOWLEDGMENTS

Kaifeng Lyu is partly supported by NSF and ONR. Simon S. Du is supported by supported by NSF IIS 2110170, NSF DMS 2134106, NSF CCF 2212261, NSF IIS 2143493, NSF CCF 2019844, NSF IIS 2229881. Wei Hu is supported by Google Research Scholar Program.

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

## A  ADDITIONAL RELATED WORKS

**Identifying Mechanisms that Cause Grokking.**    Some works attempted to identify mechanisms that cause grokking through experiments. Davies et al. (2022) hypothesized that both grokking and double descent are the result of the existence of two patterns: one pattern is faster to learn but generalizes poorly, and the other pattern is slower to learn but generalizes well. However, it is unclear how these notions can be rigorously defined for general training dynamics. Similar to the work of Liu et al. (2023) that we discussed in Section 2, Varma et al. (2023) hypothesized that grokking happens when the generalizable solutions have a smaller parameter norm than the overfitting solutions, but the former takes a longer time to learn. They also demonstrated that violating a part of this condition indeed changes the generalization behavior from grokking to "ungrokking" and "semi-grokking", two interesting phenomena they named in the paper. For training with adaptive gradient methods, Thilak et al. (2022) identified the Slingshot mechanism, which refers to a cyclic behavior of the last-layer parameter norm that empirically causes grokking. Notsawo Jr et al. (2023) also discussed a similar relationship between the oscillations in training loss and grokking. For the task of learning sparse parity, Merrill et al. (2023) empirically observed that the number of active neurons significantly decrease as the test accuracy starts to improve. While many explanations of grokking may make sense intuitively, none of these works provides rigorous justification their claims with mathematical analysis for neural net training, while our work is grounded by theoretical analyses of implicit biases in the kernel and rich regimes.

**Progress Measures of Grokking.**    Efforts have been made to find progress measures that are improving smoothly and are predictive of the time to perfect generalization. Nanda et al. (2023); Chughtai et al. (2023); Gromov (2023) focused on the tasks of learning modular arithmetic or more general group operations, and attempted to reverse engineer the weights of neural nets after grokking. It was found that the weights can exhibit special structures to make the neural net internally compute trigonometric functions or other representations of group elements. These works also made efforts to define progress measures for grokking based on this understanding of the final weights. In a related study, Morwani et al. (2024) derived analytical formulas for the weights of two-layer neural nets, assuming the margin is ultimately maximized. Hu et al. (2023) computed a variety of metrics throughout training and fit a Hidden Markov Model (HMM) over them to study the phase transition in grokking. A common issue with these works is that why progress measures themselves can make progress is still not well understood and requires new insights into the training dynamics, which is the focus of our work.

**Dynamical Analysis of Grokking.**    A few recent works were devoted to theoretically analyze the training dynamics in the grokking phenomenon. Liu et al. (2022) attributed the delayed generalization in grokking to the slow learning speed of the input embeddings. They especially analyzed the dynamics of the input embeddings in a related optimization problem, and showed that the eigenvalues of the coefficient matrix in the ODE can be used to predict the convergence. However, it is unclear how this relates to the training dynamics of the original problem and why the neural net overfits the dataset before the predicted convergence time. Žunkovič and Ilievski (2022) analyzed a simple linear classification problem, where the loss is the squared loss with explicit $\ell_1$ and $\ell_2$ regularization. Assuming specific input distributions, they proved sharp transition in test accuracy and derived an estimate of the grokking time, which depends on the initialization and regularization strength. Levi et al. (2024) leveraged random matrix theory to analyze grokking in a standard linear regression setting, where the test accuracy can provably exhibit sharp transition if the accuracy is defined as the fraction of points whose regression loss is smaller than a small threshold. All these works are limited to linear models, while our work is able to analyze deep homogeneous neural nets.

**Concurrent Works.**    Concurrent to our work, Kumar et al. (2024) hypothesized that grokking is due to a transition from lazy to rich regimes, and observed that manipulating the scale parameter for the output and the task-model alignment between NTK and ground-truth labels can control the time to escape the lazy regime and the test accuracy when staying in the lazy regime. This observation is consistent with our work, but we provide rigorous analyses for both lazy and rich regimes with quantitative bounds. Xu et al. (2024) proved that training two-layer neural nets on XOR cluster data with noisy labels can exhibit grokking. Both their work and our work go beyond linear models, but their analysis does not show whether the transition in test accuracy is sharp or not, while our work provides quantitative bounds for the sharp transition in grokking.

**Implicit Bias.** A line of works seek to characterize the implicit bias of optimization algorithms that drive them to generalizable solutions. Several forms of implicit bias have been considered, including equivalence to NTK (Du et al., 2019b;a; Allen-Zhu et al., 2019; Zou et al., 2020; Chizat et al., 2019; Arora et al., 2019b; Ji and Telgarsky, 2020b; Cao and Gu, 2019), margin maximization (Soudry et al., 2018; Nacson et al., 2019a; Lyu and Li, 2020; Ji and Telgarsky, 2020a), parameter norm minimization (Gunasekar et al., 2017; 2018a; Arora et al., 2019a) and sharpness minimization (Blanc et al., 2020; Damian et al., 2021; HaoChen et al., 2021; Li et al., 2022; Lyu et al., 2022; Gu et al., 2023). In this work, we characterize the early phase implicit bias based on NTK and late phase implicit bias based on margin maximization.

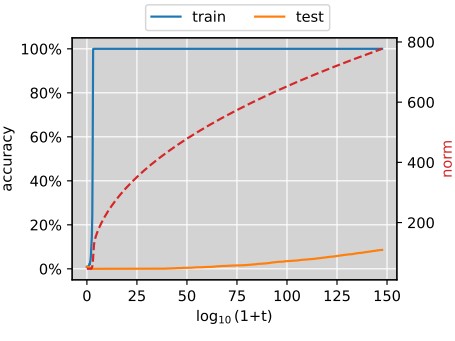 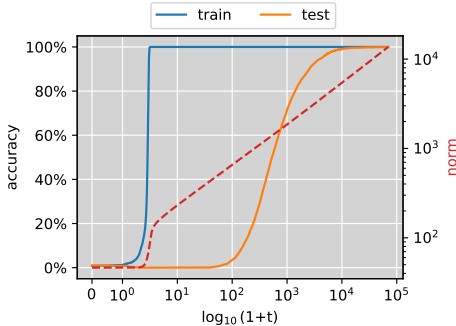

(a) time in log scale (shorter horizon)    (b) time in double log scale (longer horizon)

Figure 4: Training two-layer ReLU nets for modular addition exhibits grokking without weight decay, but the transition is no longer sharp: it can take exponentially longer than the time needed for reaching 100% training accuracy.

## B    DISCUSSION: OTHER IMPLICIT BIASES CAN ALSO INDUCE GROKKING

The main paper focuses on the grokking phenomenon induced by large initialization and small weight decay, where the transition in the implicit bias is very sharp: $\frac{1-c}{\lambda} \log \alpha$ leads to the kernel predictor, but increasing it slightly to $\frac{1+c}{\lambda} \log \alpha$ leads to a KKT solution of min-norm/max-margin problems associated with the neural net. In this section, we discuss two other possible sources of grokking: implicit margin maximization and sharpness reduction. These two sources can also cause the neural net to generalize eventually, but the transition in test accuracy is not as sharp as the case of using large initialization and small weight decay.

### B.1    IMPLICIT MARGIN MAXIMIZATION WITHOUT WEIGHT DECAY

One may wonder if grokking still exists if we remove the small weight decay in our setting. In Liu et al. (2023), it has been argued that adding a non-zero weight decay is crucial for grokking since the weight norm has to lie in a certain range (Goldilocks zone) to generalize.

We note that for classification tasks, Lyu and Li (2020); Ji and Telgarsky (2020a) have shown that gradient descent/flow on homogeneous neural nets converges to KKT points of the margin maximization problem (R1), even without weight decay. So it is still possible that some implicit bias traps the solution in early phase, such as the implicit bias to NTK Allen-Zhu et al. (2019); **?**); Ji and Telgarsky (2020b); Cao and Gu (2019). Then after training for sufficiently long time, the margin maximization bias arises and leads to grokking.

We empirically show that training two-layer neural nets without weight decay indeed leads to grokking, but the transition is no longer sharp. More specifically, we use the same setting as in Section 2 but with weight decay $\lambda = 0$. We focus on analyzing the training dynamics of full-batch gradient descent with learning rate so small that the trajectory is close to the gradient flow, which ensures that gradient noise/training instability is not a potential source of implicit regularization.

However, directly training with a constant learning rate leads to very slow loss convergence in the late phase. Inspired by Lyu and Li (2020); Nacson et al. (2019b), we set the learning rate as $\frac{\eta}{\mathcal{L}(\boldsymbol{\theta}(t))}$, where $\eta$ is a constant. The main intuition behind this is that the Hessian of $\mathcal{L}(\boldsymbol{\theta})$ can be shown to be bounded by $\mathcal{O}(\mathcal{L}(\boldsymbol{\theta}) \cdot \operatorname{poly}(\|\boldsymbol{\theta}\|_2))$, and gradient descent stays close to gradient flow when the learning rate is much lower than the reciprocal of the smoothness. Figure 4 plots training and test curves, where $x$-axis shows the corresponding continuous time calculated by summing the learning rates up to this point. It is shown that a very quick convergence in training accuracy occurs in the early phase, but the test accuracy improves extremely slowly in the late phase: the test accuracy slowly transitions from $\sim 0\%$ to $100\%$ as $t$ increases from $10^{10^2}$ to $10^{10^4}$. In contrast, Figure 2 shows that training with weight decay leads to a sharp transition when the number of steps is $\sim 10^7$.

We note that this much longer time for grokking is not surprising, since the convergence to max-margin solutions is already as slow as $\mathcal{O}(\frac{\log\log t}{\log t})$ for linear logistic regression (Soudry et al., 2018).

## B.2 SHARPNESS REDUCTION WITH LABEL NOISE

Grokking can also be triggered by the sharpness-reduction implicit bias of optimization algorithm, *e.g.*, label noise SGD (Blanc et al., 2020; Damian et al., 2021; Li et al., 2022) and 1-SAM (Wen et al., 2023). The high-level idea here is that the minimizers of the training loss connect as a Riemannian manifold due to the overparametrization in the model. Training usually contains two phases for algorithms which implicitly minimize the sharpness when the learning rate is small, where in the first phase the algorithm minimizes the training loss and finds a minimizer with poor generalization, while in the second phase the algorithm implicitly minimizes the sharpness along the manifold of the minimizers, and the model groks when the sharpness is sufficiently small. The second phase is typically longer than the first phase by magnitude and this causes grokking.

As a concrete example, Li et al. (2022) shows that grokking happens when training two-layer diagonal linear networks (4) by label noise SGD (6) when the learning rate is sufficiently small. We consider the same initialization as in Section 3.2.3, where $\boldsymbol{\theta}(0) = \alpha(\mathbf{1}, \mathbf{1})$, where $\alpha > 0$ controls the initialization scale. Different from Section 3.2.3, we consider a $\ell_2$ loss, and the label is generated by a sparse linear ground-truth, that is, $\boldsymbol{y}^*(\boldsymbol{x}) = \boldsymbol{x}^\top \boldsymbol{w}^*$, where $\boldsymbol{w}^*$ is $\kappa$-sparse. We also assume data distribution is the isotropic Gaussian distribution $\mathcal{N}(0, I)$.

$$\boldsymbol{\theta}(t+1) = \boldsymbol{\theta}(t) - \eta \nabla_{\boldsymbol{\theta}}(f(\theta; \boldsymbol{x}_{i_t}) - y_{i_t} + \xi_t)^2, \text{where } \xi_t \overset{i.i.d.}{\sim} \mathcal{N}(0, 1) \tag{6}$$

**Theorem B.1** *There exists $C_1 > 0, C_2 \in (0, 1)$, such that for any $n, d$ satisfying $C_1 \kappa \ln d \leq n \leq d(1 - C_2)$, the following holds with $1 - e^{-\Omega(n)}$ probability with respect to randomness of training data:*

1. *The gradient flow starting of $\mathcal{L}$ from $\boldsymbol{\theta}(0)$ has a limit, $\boldsymbol{\theta}_{GF}$, which achieves zero training loss. The expected population loss $\boldsymbol{\theta}_{GF}$ is at least $\Omega(\|\boldsymbol{w}^*\|_2^2)$. (Theorem 6.7)*

2. *For any $\alpha \in (1, 2)$ and $T > 0$, $\boldsymbol{\theta}(T/\eta^\alpha)$ converges to $\boldsymbol{\theta}_{GF}$ in distribution as $\eta \to 0$. (Implication of Theorem B.9)*

3. *For any $T > 0$, $\boldsymbol{\theta}(T/\eta^2)$ converges to some $\bar{\boldsymbol{\theta}}(T)$ in distribution as $\eta \to 0$ and $\bar{\boldsymbol{\theta}}(T)$ has zero training loss. Moreover, the population loss of $\bar{\boldsymbol{\theta}}(T)$ converges to 0 as $T \to \infty$. (Theorem 6.1)*

The above theorem shows that a sharp transition of the implicit biases can be seen in the log time scale: when $\log t = \log T + \alpha \log \frac{1}{\eta}$ for any $\alpha \in (1, 2)$, it converges to $\boldsymbol{\theta}_{GF}$; but increasing the time slightly in the log scale to $\log t = \log T + 2 \log \frac{1}{\eta}$ leads to $\bar{\boldsymbol{\theta}}(T)$, which minimizes the population loss to 0. However, this transition is less sharp than our main results, where we show that increasing the time slightly in the normal scale, from $\frac{1-c}{\lambda} \log \alpha$ to $\frac{1+c}{\lambda} \log \alpha$, changes the implicit bias.

## C  PROOFS FOR CLASSIFICATION WITH L2 REGULARIZATION

### C.1  PRELIMINARIES

Let $F_{\mathrm{LL}} : [1, +\infty) \to [0, +\infty), x \mapsto x \log x$ be the linear-times-log function, and $F_{\mathrm{LL}}^{-1} : [0, +\infty) \to [1, +\infty)$ be its inverse function.

**Lemma C.1** *For all $y > 0$, $F_{\mathrm{LL}}^{-1}(y) \geq \frac{y}{\log(y+1)}$.*

**Proof:**  Since $F_{\mathrm{LL}}$ is an increasing function, it suffices to show $F_{\mathrm{LL}}\left(\frac{y}{\log(y+1)}\right) \leq y$, and this can be indeed proved by

$$F_{\mathrm{LL}}\left(\frac{y}{\log(y+1)}\right) \leq \frac{y}{\log(y+1)} \log \frac{y}{\log(y+1)} = y \cdot \frac{\log \frac{y}{\log(y+1)}}{\log(y+1)} \leq y,$$

where the last inequality holds because $\frac{y}{\log(y+1)} \leq y + 1$. $\qquad \square$

**Lemma C.2** *For all $y > 0$ and $\beta \in (0, 1)$,*

$$F_{\mathrm{LL}}^{-1}(\beta y) - 1 \geq \beta(F_{\mathrm{LL}}^{-1}(y) - 1).$$

**Proof:**  By concavity of $F_{\mathrm{LL}}^{-1}$, $F_{\mathrm{LL}}^{-1}(\beta y) \geq \beta F_{\mathrm{LL}}^{-1}(y) + (1 - \beta)F_{\mathrm{LL}}^{-1}(0)$. Rearranging the terms gives the desired result. $\qquad \square$

**Lemma C.3** *The following holds for $r_1, \ldots, r_n > 0$ and $\mathcal{L} = \frac{1}{n} \sum_{i=1}^{n} r_i$:*

$$\mathcal{L} \log \frac{1}{n\mathcal{L}} \leq \frac{1}{n} \sum_{i=1}^{n} r_i \log \frac{1}{r_i} \leq \mathcal{L} \log \frac{1}{\mathcal{L}}.$$

**Proof:**  For proving the first inequality, note that $\log \frac{1}{r_i} \geq \log \frac{1}{n\mathcal{L}}$, so

$$\frac{1}{n} \sum_{i=1}^{n} r_i \log \frac{1}{r_i} \geq \frac{1}{n} \sum_{i=1}^{n} r_i \log \frac{1}{n\mathcal{L}} = \mathcal{L} \log \frac{1}{n\mathcal{L}}.$$

For the other inequality, by Jensen's inequality, we have

$$\frac{1}{n} \sum_{i=1}^{n} r_i \log \frac{1}{r_i} = -\frac{1}{n} \sum_{i=1}^{n} F_{\mathrm{LL}}(r_i) \leq -F_{\mathrm{LL}}(\mathcal{L}) = \mathcal{L} \log \frac{1}{\mathcal{L}},$$

which completes the proof. $\qquad \square$

### C.2  KERNEL REGIME

Now we present the proof of Theorem 3.4 for the kernel regime. We follow the notations in Section 3.2.1 and define $\boldsymbol{h}_{\mathrm{ntk}}^*$ as the unique unit vector along the direction of the optimal solution to (K1) and $\gamma_{\mathrm{ntk}} := \min_{i \in [n]}\{y_i \langle \nabla f_i(\bar{\boldsymbol{\theta}}_{\mathrm{init}}), \boldsymbol{h}_{\mathrm{ntk}}^* \rangle\}$. The kernel regime is a training regime where the parameter does not move very far from the initialization. Mathematically, let $\epsilon_{\max} > 0$ be a small constant so that

$$\max_{i \in [n]}\{\|\nabla f_i(\boldsymbol{\theta}) - \nabla f_i(\bar{\boldsymbol{\theta}}_{\mathrm{init}})\|_2 : i \in [n]\} < \frac{\gamma_{\mathrm{ntk}}}{2}$$

holds for all $\|\boldsymbol{\theta} - \bar{\boldsymbol{\theta}}_{\mathrm{init}}\|_2 < \epsilon_{\max}$, and let $T_{\max} := \inf\left\{t \geq 0 : \left\|\frac{e^{\lambda t}}{\alpha}\boldsymbol{\theta}(t) - \bar{\boldsymbol{\theta}}_{\mathrm{init}}\right\|_2 > \epsilon_{\max}\right\}$ be the time that $\boldsymbol{\theta}(t)$ moves far enough to change the NTK features significantly.

A key difference to existing analyses of the kernel regime is that our setting has weight decay, so the norm can change a lot even when the parameter direction has not changed much yet. Therefore, we normalize the parameter by multiplying a factor of $\frac{e^{\lambda t}}{\alpha}$ and compare it to the initial parameter direction. We consider $\boldsymbol{\theta}(t)$ as moving too far only if $\|\frac{e^{\lambda t}}{\alpha}\boldsymbol{\theta}(t) - \bar{\boldsymbol{\theta}}_{\mathrm{init}}\|_2$ is too large, which is different from existing analyses in that they usually consider the absolute moving distance $\|\boldsymbol{\theta}(t) - \boldsymbol{\theta}(0)\|_2$.

In the following, we first derive an upper bound for the loss convergence rate, and then use it to lower bound the training time in the kernel regime $T_{\max} \geq \frac{1}{\lambda}(\log \alpha - \frac{1}{L} \log \log A + \Omega(1))$. Finally, we establish the implicit bias towards the kernel predictor with a more careful analysis of the trajectory.

More notations are needed for the proof. Let $q_i(\boldsymbol{\theta}) := y_i f_i(\boldsymbol{\theta})$ be the output margin of the neural net at the $i$-th training sample. Let $r_i(\boldsymbol{\theta}) = \ell(f_i(\boldsymbol{\theta}), y_i) = \exp(-q_i(\boldsymbol{\theta}))$ be the loss incurred at the $i$-th training sample. Let $A := \frac{\alpha^{2(L-1)}}{\lambda}$, which is a factor that will appear very often in our proof. Let $G_{\mathrm{ntk}} := \sup\{\|\nabla f_i(\boldsymbol{\theta})\|_2 : \|\boldsymbol{\theta} - \bar{\boldsymbol{\theta}}_{\mathrm{init}}\|_2 < \epsilon_{\max}, i \in [n]\}$ be the maximum norm of NTK features. The following lemma gives an upper bound for the training loss over time.

**Lemma C.4** *For all* $0 \leq t \leq T_{\max}$,

$$\frac{\mathrm{d}\mathcal{L}}{\mathrm{d}t} \leq -\frac{\gamma_{\mathrm{ntk}}^2}{4} \alpha^{2(L-1)} e^{-2(L-1)\lambda t} \mathcal{L}^2 + \lambda L \cdot \mathcal{L} \log \frac{1}{\mathcal{L}}, \tag{7}$$

$$\mathcal{L}(\boldsymbol{\theta}(t)) \leq \max \left\{ \frac{1}{1 + \frac{\gamma_{\mathrm{ntk}}^2}{16(L-1)} A(1 - e^{-2(L-1)\lambda t})}, \frac{e^{2(L-1)\lambda t}}{F_{\mathrm{LL}}^{-1}(\frac{\gamma_{\mathrm{ntk}}^2}{8L} A) - 1} \right\}. \tag{8}$$

**Proof:** By the update rule of $\boldsymbol{\theta}$, the following holds for all $t > 0$:

$$\frac{\mathrm{d}\mathcal{L}}{\mathrm{d}t} = \left\langle \nabla \mathcal{L}(\boldsymbol{\theta}), \frac{\mathrm{d}\boldsymbol{\theta}}{\mathrm{d}t} \right\rangle = \underbrace{- \left\| \frac{1}{n} \sum_{i=1}^n r_i \nabla q_i(\boldsymbol{\theta}) \right\|_2^2}_{=:V_1} + \underbrace{\frac{\lambda L}{n} \sum_{i=1}^n r_i q_i}_{=:V_2}.$$

For $V_1$, we have

$$-V_1 = \left\| \frac{1}{n} \sum_{i=1}^n r_i \nabla q_i(\boldsymbol{\theta}) \right\|_2^2 \geq \left( \frac{1}{n} \sum_{i=1}^n r_i \langle \nabla q_i(\boldsymbol{\theta}), \boldsymbol{h}_{\mathrm{ntk}}^* \rangle \right)^2. \tag{9}$$

Note that for all $t \leq T_{\max}$,

$$\frac{1}{n} \sum_{i=1}^n r_i \langle \nabla q_i(\boldsymbol{\theta}), \boldsymbol{h}_{\mathrm{ntk}}^* \rangle \geq \frac{1}{n} \sum_{i=1}^n r_i \left( \langle \nabla q_i(\alpha e^{-\lambda t} \bar{\boldsymbol{\theta}}_{\mathrm{init}}), \boldsymbol{h}_{\mathrm{ntk}}^* \rangle - \|\nabla f_i(\boldsymbol{\theta}) - \nabla f_i(\alpha e^{-\lambda t} \bar{\boldsymbol{\theta}}_{\mathrm{init}})\|_2 \right)$$

$$\geq \frac{1}{n} \sum_{i=1}^n r_i \cdot \alpha^{L-1} e^{-(L-1)\lambda t} \left( \gamma_{\mathrm{ntk}} - \frac{1}{2}\gamma_{\mathrm{ntk}} \right)$$

$$= \frac{\gamma_{\mathrm{ntk}}}{2} \alpha^{L-1} e^{-(L-1)\lambda t} \mathcal{L},$$

where the second inequality is due to $\langle \nabla q_i(\bar{\boldsymbol{\theta}}_{\mathrm{init}}), \boldsymbol{h}_{\mathrm{ntk}}^* \rangle \geq \gamma_{\mathrm{ntk}}$, $\left\| \nabla f_i(\frac{e^{\lambda t}}{\alpha} \boldsymbol{\theta}(t)) - \nabla f_i(\bar{\boldsymbol{\theta}}_{\mathrm{init}}) \right\|_2 \leq \frac{\gamma_{\mathrm{ntk}}}{2}$ and the $(L-1)$-homogeneity of $\nabla f_i$. Together with (9), we obtain the following upper bound for $V_1$:

$$V_1 \leq -\frac{\gamma_{\mathrm{ntk}}^2}{4} \alpha^{2(L-1)} e^{-2(L-1)\lambda t} \mathcal{L}^2. \tag{10}$$

For $V_2$, by Lemma C.3 we have $\frac{1}{n} \sum_{i=1}^n r_i \log \frac{1}{r_i} \leq \mathcal{L} \log \frac{1}{\mathcal{L}}$. So we have the following upper bound for $V_2$:

$$V_2 \leq \lambda L \cdot \mathcal{L} \log \frac{1}{\mathcal{L}}. \tag{11}$$

Combining this with (10) together proves (7).

For proving (8), we first divide by $\mathcal{L}^2$ on both sides of (7) to get

$$\frac{\mathrm{d}}{\mathrm{d}t} \frac{1}{\mathcal{L}} \geq \frac{\gamma_{\mathrm{ntk}}^2}{4} \alpha^{2(L-1)} e^{-2(L-1)\lambda t} - \lambda L \cdot \frac{1}{\mathcal{L}} \log \frac{1}{\mathcal{L}}. \tag{12}$$

Let $\mathcal{E} := \{t \in [0, T_{\max}] : F_{\mathrm{LL}}(\frac{1}{\mathcal{L}}) \geq \frac{\gamma_{\mathrm{ntk}}^2}{8L} A e^{-2(L-1)\lambda t}\}$. For all $t \in [0, T_{\max}]$, if $t \in \mathcal{E}$, then by Lemma C.2,

$$\mathcal{L}(\boldsymbol{\theta}(t)) \leq \frac{1}{F_{\mathrm{LL}}^{-1}\left(\frac{\gamma_{\mathrm{ntk}}^2}{8L} A e^{-2(L-1)\lambda t}\right)} \leq \frac{1}{1 + \left(F_{\mathrm{LL}}^{-1}(\frac{\gamma_{\mathrm{ntk}}^2}{8L} A) - 1\right) e^{-2(L-1)\lambda t}}$$

$$\leq \frac{1}{\left(F_{\mathrm{LL}}^{-1}(\frac{\gamma_{\mathrm{ntk}}^2}{8L} A) - 1\right) e^{-2(L-1)\lambda t}} \leq \frac{e^{2(L-1)\lambda t}}{F_{\mathrm{LL}}^{-1}(\frac{\gamma_{\mathrm{ntk}}^2}{8L} A) - 1}.$$

Otherwise, let $t'$ be the largest number in $\mathcal{E} \cup \{0\}$ that is smaller than $t$. Then

$$\frac{1}{\mathcal{L}(\boldsymbol{\theta}(t))} = \frac{1}{\mathcal{L}(\boldsymbol{\theta}(t'))} + \int_{t'}^t \frac{\mathrm{d}}{\mathrm{d}\tau} \frac{1}{\mathcal{L}} \, \mathrm{d}\tau$$

$$\geq \frac{1}{\mathcal{L}(\boldsymbol{\theta}(t'))} + \int_{t'}^t \frac{\gamma_{\mathrm{ntk}}^2}{8} \alpha^{2(L-1)} e^{-2(L-1)\lambda\tau} \, \mathrm{d}\tau$$

$$= \frac{1}{\mathcal{L}(\boldsymbol{\theta}(t'))} + \frac{\gamma_{\mathrm{ntk}}^2}{16(L-1)} A(e^{-2(L-1)\lambda t'} - e^{-2(L-1)\lambda t}).$$

If $t' \in \mathcal{E}$, then $\frac{1}{\mathcal{L}(\boldsymbol{\theta}(t))} \geq F_{\mathrm{LL}}^{-1}\left(\frac{\gamma_{\mathrm{ntk}}^2}{8L} A e^{-2(L-1)\lambda t'}\right) \geq F_{\mathrm{LL}}^{-1}\left(\frac{\gamma_{\mathrm{ntk}}^2}{8L} A e^{-2(L-1)\lambda t}\right)$, which contradicts to $t \notin \mathcal{E}$. So it must hold that $t' = 0$, and thus $\frac{1}{\mathcal{L}(\boldsymbol{\theta}(t))} \geq 1 + \frac{\gamma_{\mathrm{ntk}}^2}{16(L-1)} A(1 - e^{-2(L-1)\lambda t})$. Combining all these together proves (8). $\square$

**Lemma C.5** *There exists a constant $\Delta T$ such that $T_{\max} \geq \frac{1}{\lambda}\left(\log \alpha - \frac{1}{L} \log\log A + \Delta T\right)$ when $\alpha$ is sufficiently large. Furthermore, for all $t \leq \frac{1}{\lambda}\left(\log \alpha - \frac{1}{L} \log\log A + \Delta T\right)$,*

$$\left\| \frac{1}{\alpha} e^{\lambda t} \boldsymbol{\theta}(t) - \bar{\boldsymbol{\theta}}_{\mathrm{init}} \right\|_2 = \mathcal{O}(\alpha^{-L} \log A) \cdot e^{L\lambda t}.$$

**Proof:** By definition of $T_{\max}$, to prove the first claim, it suffices to show that there exists a constant $\Delta T$ such that $\left\| \frac{1}{\alpha} e^{\lambda t} \boldsymbol{\theta}(t) - \bar{\boldsymbol{\theta}}_{\mathrm{init}} \right\|_2 \leq \epsilon_{\max}$ for all $t \leq \min\{\frac{1}{\lambda}\left(\log \alpha - \frac{1}{L} \log\log A + \Delta T\right), T_{\max}\}$. When $t \leq T_{\max}$, we have the following gradient upper bound:

$$\|\nabla \mathcal{L}(\boldsymbol{\theta}(t))\|_2 = \left\| \frac{1}{n} \sum_{i=1}^n r_i \nabla q_i(\boldsymbol{\theta}(t)) \right\|_2 \leq \frac{1}{n} \sum_{i=1}^n r_i \|\nabla q_i(\boldsymbol{\theta}(t))\|_2$$

$$\leq \frac{1}{n} \sum_{i=1}^n r_i \alpha^{L-1} e^{-(L-1)\lambda t} \|\nabla q_i(\tfrac{1}{\alpha} e^{\lambda t} \boldsymbol{\theta}(t))\|_2$$

$$\leq G_{\mathrm{ntk}} \alpha^{L-1} e^{-(L-1)\lambda t} \mathcal{L}.$$

By update rule, $e^{\lambda t} \boldsymbol{\theta}(t) - \alpha \bar{\boldsymbol{\theta}}_{\mathrm{init}} = \int_0^t e^{\lambda \tau} \nabla \mathcal{L}(\boldsymbol{\theta}(\tau)) \mathrm{d}\tau$. Applying the gradient upper bounds gives

$$\frac{1}{\alpha} \left\| e^{\lambda t} \boldsymbol{\theta}(t) - \alpha \bar{\boldsymbol{\theta}}_{\mathrm{init}} \right\|_2 \leq \frac{1}{\alpha} \int_0^t \left\| e^{\lambda \tau} \nabla \mathcal{L}(\boldsymbol{\theta}(\tau)) \right\|_2 \mathrm{d}\tau$$

$$\leq \frac{1}{\alpha} \int_0^t e^{\lambda \tau} G_{\mathrm{ntk}} \alpha^{L-1} e^{-(L-1)\lambda \tau} \mathcal{L}(\boldsymbol{\theta}(\tau)) \, \mathrm{d}\tau$$

$$= G_{\mathrm{ntk}} \alpha^{L-2} \int_0^t e^{-(L-2)\lambda \tau} \mathcal{L}(\boldsymbol{\theta}(\tau)) \, \mathrm{d}\tau.$$

Substituting $\mathcal{L}$ in $\int_0^t e^{-(L-2)\lambda \tau} \mathcal{L}(\boldsymbol{\theta}(\tau)) \, \mathrm{d}\tau$ with the loss upper bound in Lemma C.4, we obtain the following for all $t \leq T_{\max}$,

$$\int_0^t e^{-(L-2)\lambda \tau} \mathcal{L}(\boldsymbol{\theta}(\tau)) \, \mathrm{d}\tau \leq \int_0^t \max \left\{ \frac{e^{-(L-2)\lambda \tau}}{1 + \frac{\gamma_{\mathrm{ntk}}^2}{16(L-1)} A(1 - e^{-2(L-1)\lambda \tau})}, \frac{e^{-(L-2)\lambda \tau} \cdot e^{2(L-1)\lambda \tau}}{F_{\mathrm{LL}}^{-1}(\frac{\gamma_{\mathrm{ntk}}^2}{8L} A) - 1} \right\} \mathrm{d}\tau.$$

So we have $\int_0^t e^{-(L-2)\lambda\tau}\mathcal{L}(\boldsymbol{\theta}(\tau))\,\mathrm{d}\tau \leq \max\{I_1, I_2\}$, where

$$I_1 := \int_0^t \frac{\mathrm{d}\tau}{1 + \frac{\gamma_{\mathrm{ntk}}^2}{16(L-1)}A(1 - e^{-2(L-1)\lambda\tau})}, \qquad I_2 := \int_0^t \frac{e^{L\lambda\tau}}{F_{\mathrm{LL}}^{-1}(\frac{\gamma_{\mathrm{ntk}}^2}{8L}A) - 1}\,\mathrm{d}\tau.$$

We can exactly compute the integrals for both $I_1$ and $I_2$. For $I_1$, it holds for all $t \leq \min\{T_{\max}, \mathcal{O}(\frac{1}{\lambda}\log\alpha)\}$ that

$$
\begin{aligned}
I_1 &= \frac{1}{2(L-1)\lambda\left(1 + \frac{\gamma_{\mathrm{ntk}}^2}{16(L-1)}A\right)}\log\left(\left(1 + \frac{\gamma_{\mathrm{ntk}}^2}{16(L-1)}A\right)e^{2(L-1)\lambda t} - \frac{\gamma_{\mathrm{ntk}}^2}{16(L-1)}A\right)\\
&\leq \frac{1}{2(L-1)\lambda\left(1 + \frac{\gamma_{\mathrm{ntk}}^2}{16(L-1)}A\right)}\log\left(\left(1 + \frac{\gamma_{\mathrm{ntk}}^2}{16(L-1)}A\right)\mathrm{poly}(\alpha) - \frac{\gamma_{\mathrm{ntk}}^2}{16(L-1)}A\right)\\
&= \mathcal{O}\left(\frac{\log A}{\lambda A}\right).
\end{aligned}
$$

For $I_2$, it holds for all $t \leq T_{\max}$ that

$$I_2 = \frac{e^{L\lambda t} - 1}{L\lambda \cdot \left(F_{\mathrm{LL}}^{-1}(\frac{\gamma_{\mathrm{ntk}}^2}{8L}A) - 1\right)} = \mathcal{O}\left(\frac{\log A}{\lambda A}\right) \cdot e^{L\lambda t}.$$

Putting all these together, we have

$$
\begin{aligned}
\frac{1}{\alpha}\left\|e^{\lambda t}\boldsymbol{\theta}(t) - \alpha\bar{\boldsymbol{\theta}}_{\mathrm{init}}\right\|_2 &= \mathcal{O}(\alpha^{L-2}) \cdot \left(\mathcal{O}\left(\frac{\log A}{\lambda A}\right) + \mathcal{O}\left(\frac{\log A}{\lambda A}\right) \cdot e^{L\lambda t}\right)\\
&= \mathcal{O}\left(\frac{\alpha^{L-2}\log A}{\lambda A}\right) \cdot e^{L\lambda t} = \mathcal{O}(\alpha^{-L}\log A) \cdot e^{L\lambda t}.
\end{aligned}
\tag{13}
$$

Therefore, there exists a constant $C > 0$ such that when $\alpha$ is sufficiently large, $\left\|\frac{1}{\alpha}e^{\lambda t}\boldsymbol{\theta}(t) - \bar{\boldsymbol{\theta}}_{\mathrm{init}}\right\|_2 \leq C\alpha^{-L}\log A \cdot e^{L\lambda t}$ for all $t \leq \min\{T_{\max}, \mathcal{O}(\frac{1}{\lambda}\log\alpha)\}$. Setting $\Delta T := \frac{1}{L}\log\frac{\epsilon_{\max}}{C}$, we obtain the following bound for all $t \leq \min\{T_{\max}, \frac{1}{\lambda}(\log\alpha - \frac{1}{L}\log\log A + \Delta T)\}$:

$$\left\|\frac{1}{\alpha}e^{\lambda t}\boldsymbol{\theta}(t) - \bar{\boldsymbol{\theta}}_{\mathrm{init}}\right\|_2 \leq C\alpha^{-L}\log A \cdot \exp\left(L\log\alpha - \log\log A + \log\frac{\epsilon_{\max}}{C}\right) = \epsilon_{\max},$$

which implies $T_{\max} \geq \frac{1}{\lambda}(\log\alpha - \frac{1}{L}\log\log A + \Delta T)$. Combining this with (13) proves the second claim. $\square$

Now we turn to analyze the implicit bias. Let $\boldsymbol{\delta}(t; \alpha\bar{\boldsymbol{\theta}}_{\mathrm{init}}) := \left(\frac{1}{\alpha}e^{\lambda t}\boldsymbol{\theta}(t; \alpha\bar{\boldsymbol{\theta}}_{\mathrm{init}}) - \bar{\boldsymbol{\theta}}_{\mathrm{init}}\right)$ and $\boldsymbol{h}(t; \alpha\bar{\boldsymbol{\theta}}_{\mathrm{init}}) := \alpha^L e^{-L\lambda t}\boldsymbol{\delta}(t; \alpha\bar{\boldsymbol{\theta}}_{\mathrm{init}})$. The following lemma shows that the output function can be approximated by a linear function.

**Lemma C.6** For all $t \leq \frac{1}{\lambda}\left(\log\alpha - \frac{1}{L}\log\log A + \Delta T\right)$,

$$q_i(\tfrac{1}{\alpha}e^{\lambda t}\boldsymbol{\theta}) = \langle\nabla q_i(\bar{\boldsymbol{\theta}}_{\mathrm{init}}), \boldsymbol{\delta}\rangle + \mathcal{O}(\alpha^{-L}\log A) \cdot e^{L\lambda t}\|\boldsymbol{\delta}\|_2, \tag{14}$$

$$q_i(\boldsymbol{\theta}) = \langle\nabla q_i(\bar{\boldsymbol{\theta}}_{\mathrm{init}}), \boldsymbol{h}\rangle + \mathcal{O}(\alpha^{-L}\log A) \cdot e^{L\lambda t}\|\boldsymbol{h}\|_2. \tag{15}$$

**Proof:** By Lemma C.5 and Taylor expansion, we have

$$
\begin{aligned}
q_i(\tfrac{1}{\alpha}e^{\lambda t}\boldsymbol{\theta}) &= q_i(\bar{\boldsymbol{\theta}}_{\mathrm{init}}) + \langle\nabla q_i(\bar{\boldsymbol{\theta}}_{\mathrm{init}}), \boldsymbol{\delta}\rangle + \mathcal{O}(\|\boldsymbol{\delta}\|_2^2)\\
&= q_i(\bar{\boldsymbol{\theta}}_{\mathrm{init}}) + \langle\nabla q_i(\bar{\boldsymbol{\theta}}_{\mathrm{init}}), \boldsymbol{\delta}\rangle + \mathcal{O}(\alpha^{-L}\log A) \cdot e^{L\lambda t}\|\boldsymbol{\delta}\|_2\\
&= \langle\nabla q_i(\bar{\boldsymbol{\theta}}_{\mathrm{init}}), \boldsymbol{\delta}\rangle + \mathcal{O}(\alpha^{-L}\log A) \cdot e^{L\lambda t}\|\boldsymbol{\delta}\|_2,
\end{aligned}
$$

which proves (14). Combining this with the $L$-homogeneity of $f_i$ proves (15). $\square$

In the following two lemmas, we derive lower bounds for the loss convergence that will be used in the implicit bias analysis later.

**Lemma C.7** *For all $0 \leq t \leq T_{\max}$,*

$$\frac{\mathrm{d}\mathcal{L}}{\mathrm{d}t} \geq -G_{\mathrm{ntk}}^2 \alpha^{2(L-1)} e^{-2(L-1)\lambda t} \mathcal{L}^2 + \lambda L \cdot \mathcal{L} \log \frac{1}{n\mathcal{L}}, \tag{16}$$

$$\mathcal{L}(\boldsymbol{\theta}(t)) \geq \min\left\{ \frac{n}{F_{\mathrm{LL}}^{-1}\left(\frac{2}{nL} G_{\mathrm{ntk}}^2 A e^{-2(L-1)\lambda t}\right)}, \frac{1}{n} \exp(1 - 2(L-1)\lambda) \right\}. \tag{17}$$

**Proof:** The proof is similar to that of [Lemma C.4](#), but now we are proving lower bounds. By the update rule of $\boldsymbol{\theta}$, the following holds for all $t > 0$:

$$\frac{\mathrm{d}\mathcal{L}}{\mathrm{d}t} = \left\langle \nabla \mathcal{L}(\boldsymbol{\theta}), \frac{\mathrm{d}\boldsymbol{\theta}}{\mathrm{d}t} \right\rangle = \underbrace{-\left\| \frac{1}{n} \sum_{i=1}^{n} r_i \nabla q_i(\boldsymbol{\theta}) \right\|_2^2}_{=:V_1} + \underbrace{\frac{\lambda L}{n} \sum_{i=1}^{n} r_i \log \frac{1}{r_i}}_{=:V_2}.$$

For $V_1$, we can lower bound it as follows:

$$V_1 \geq -\left\| \frac{1}{n} \sum_{i=1}^{n} r_i \nabla q_i(\boldsymbol{\theta}) \right\|_2^2$$

$$\geq -\alpha^{2(L-1)} e^{-2(L-1)\lambda t} \left\| \frac{1}{n} \sum_{i=1}^{n} r_i \nabla q_i(\tfrac{1}{\alpha} e^{\lambda t} \boldsymbol{\theta}) \right\|_2^2$$

$$\geq -\alpha^{2(L-1)} e^{-2(L-1)\lambda t} G_{\mathrm{ntk}}^2 \mathcal{L}^2,$$

where the last inequality holds because $\|\nabla q_i(\tfrac{1}{\alpha} e^{\lambda t} \boldsymbol{\theta})\|_2 \leq \|\nabla q_i(\bar{\boldsymbol{\theta}}_{\mathrm{init}})\|_2 + \|\nabla q_i(\tfrac{1}{\alpha} e^{\lambda t} \boldsymbol{\theta}) - \nabla q_i(\bar{\boldsymbol{\theta}}_{\mathrm{init}})\|_2 \leq G_{\mathrm{ntk}}$.

For $V_2$, by [Lemma C.3](#) we have $\frac{1}{n} \sum_{i=1}^{n} r_i \log \frac{1}{r_i} \geq \mathcal{L} \log \frac{1}{n\mathcal{L}}$. So $V_2 \geq \lambda L \cdot \mathcal{L} \log \frac{1}{n\mathcal{L}}$. Combining the inequalities for $V_1$ and $V_2$ together proves (16).

For proving (8), we first divide by $\mathcal{L}^2$ on both sides of (16) to get

$$\frac{\mathrm{d}}{\mathrm{d}t} \frac{1}{\mathcal{L}} \leq G_{\mathrm{ntk}}^2 \alpha^{2(L-1)} e^{-2(L-1)\lambda t} - \lambda L \cdot \frac{1}{\mathcal{L}} \log \frac{1}{n\mathcal{L}}. \tag{18}$$

Let $\mathcal{E} := \{t \in [0, T_{\max}] : F_{\mathrm{LL}}(\frac{1}{n\mathcal{L}}) \leq \frac{2}{nL} G_{\mathrm{ntk}}^2 A e^{-2(L-1)\lambda t}\}$. It is easy to see that $0 \in \mathcal{E}$. For all $t \in [0, T_{\max}]$, if $t \in \mathcal{E}$, then by [Lemma C.2](#),

$$\mathcal{L}(\boldsymbol{\theta}(t)) \geq \frac{n}{F_{\mathrm{LL}}^{-1}\left(\frac{2}{nL} G_{\mathrm{ntk}}^2 A e^{-2(L-1)\lambda t}\right)}.$$

If no $t$ is at the boundary of $\mathcal{E}$, then we are done. Otherwise, let $t > 0$ be one on the boundary of $\mathcal{E}$. Then at time $t$,

$$\frac{\mathrm{d}}{\mathrm{d}t} \frac{1}{n\mathcal{L}} = -\frac{\lambda L}{2} \cdot \frac{1}{n\mathcal{L}} \log \frac{1}{n\mathcal{L}}.$$

So as long as $1 + \log \frac{1}{n\mathcal{L}(\boldsymbol{\theta}(t))} > 2(L-1)\lambda$,

$$\frac{\mathrm{d}}{\mathrm{d}t} F_{\mathrm{LL}}(\frac{1}{n\mathcal{L}}) = (1 + \log \frac{1}{n\mathcal{L}}) \frac{\mathrm{d}}{\mathrm{d}t} \frac{1}{n\mathcal{L}}$$

$$= -(1 + \log \frac{1}{n\mathcal{L}}) \cdot \frac{\lambda L}{2} \cdot \frac{1}{n\mathcal{L}} \log \frac{1}{n\mathcal{L}}$$

$$= -(1 + \log \frac{1}{n\mathcal{L}}) \cdot \frac{2}{nL} G_{\mathrm{ntk}}^2 A e^{-2(L-1)\lambda t} < \left(\frac{2}{nL} G_{\mathrm{ntk}}^2 A e^{-2(L-1)\lambda t}\right)',$$

which means $t$ is not actually on the boundary. So $1 + \log \frac{1}{n\mathcal{L}(\boldsymbol{\theta}(t))} \leq 2(L-1)\lambda$. Repeating this argument with inequalities and taking some integrals can show that every point outside $\mathcal{E}$ satisfies $1 + \log \frac{1}{n\mathcal{L}(\boldsymbol{\theta}(t))} \leq 2(L-1)\lambda$. $\qquad\square$

**Lemma C.8** *For any constant $c \in (0, 1)$, letting $T_c^-(\alpha) := \frac{1-c}{\lambda} \log \alpha$, it holds that*

$$\frac{\mathrm{d}\|\boldsymbol{h}\|_2}{\mathrm{d}t} = \alpha^{2(L-1)} e^{-2(L-1)\lambda t} \left\langle \frac{1}{n} \sum_{i=1}^n r_i(\boldsymbol{\theta}) \nabla q_i(\tfrac{1}{\alpha} e^{\lambda t} \boldsymbol{\theta}), \frac{\boldsymbol{h}}{\|\boldsymbol{h}\|_2} \right\rangle - L\lambda \|\boldsymbol{h}\|_2, \qquad (19)$$

$$\frac{\mathrm{d}}{\mathrm{d}t} \log \frac{1}{\mathcal{L}} \geq \left( \gamma_{\mathrm{ntk}} + \mathcal{O}(\alpha^{-L} \log A) \cdot e^{L\lambda t} \right) \frac{\mathrm{d}\|\boldsymbol{h}\|_2}{\mathrm{d}t}. \qquad (20)$$

**Proof:** By update rule,

$$\begin{aligned}
\frac{\mathrm{d}\boldsymbol{h}}{\mathrm{d}t} &= \alpha^L e^{-L\lambda t} \frac{\mathrm{d}\boldsymbol{\delta}}{\mathrm{d}t} - L\lambda \alpha^L e^{-L\lambda t} \boldsymbol{\delta} \\
&= \alpha^L e^{-L\lambda t} \left( \frac{e^{\lambda t}}{\alpha} \frac{1}{n} \sum_{i=1}^n r_i(\boldsymbol{\theta}(t)) \nabla q_i(\boldsymbol{\theta}(t)) \right) - L\lambda \boldsymbol{h} \\
&= \alpha^{2(L-1)} e^{-2(L-1)\lambda t} \frac{1}{n} \sum_{i=1}^n r_i(\boldsymbol{\theta}(t)) \nabla q_i(\tfrac{1}{\alpha} e^{\lambda t} \boldsymbol{\theta}(t)) - L\lambda \boldsymbol{h}.
\end{aligned}$$

Then we can deduce (19) from $\frac{\mathrm{d}}{\mathrm{d}t} \|\boldsymbol{h}\|_2 = \left\langle \frac{\mathrm{d}\boldsymbol{h}}{\mathrm{d}t}, \frac{\boldsymbol{h}}{\|\boldsymbol{h}\|_2} \right\rangle$.

For proving (20), first, we apply the chain rule to get

$$\frac{\mathrm{d}}{\mathrm{d}t} \log \frac{1}{\mathcal{L}} = \frac{1}{\mathcal{L}(\boldsymbol{\theta})} \langle -\nabla \mathcal{L}(\boldsymbol{\theta}) - \lambda \boldsymbol{\theta}, -\nabla \mathcal{L}(\boldsymbol{\theta}) \rangle = \underbrace{\frac{1}{\mathcal{L}(\boldsymbol{\theta})} \|\nabla \mathcal{L}(\boldsymbol{\theta})\|_2^2}_{=:V_1} - \underbrace{\frac{\lambda}{\mathcal{L}(\boldsymbol{\theta})} \langle \boldsymbol{\theta}, -\nabla \mathcal{L}(\boldsymbol{\theta}) \rangle}_{=:V_2}.$$

For $V_1$, note that

$$\|\nabla \mathcal{L}(\boldsymbol{\theta})\|_2 = \left\| \frac{1}{n} \sum_{i=1}^n r_i(\boldsymbol{\theta}) \nabla q_i(\boldsymbol{\theta}) \right\|_2 = \alpha^{L-1} e^{-(L-1)\lambda t} \underbrace{\left\| \frac{1}{n} \sum_{i=1}^n r_i(\boldsymbol{\theta}) \nabla q_i(\tfrac{1}{\alpha} e^{\lambda t} \boldsymbol{\theta}) \right\|_2}_{=:V_3}.$$

By Lemma C.5, $\|\boldsymbol{\delta}(t)\|_2 = \mathcal{O}(\alpha^{-L} \log A) \cdot e^{L\lambda t}$, so $\langle \nabla q_i(\tfrac{1}{\alpha} e^{\lambda t} \boldsymbol{\theta}), \boldsymbol{h}_{\mathrm{ntk}}^* \rangle = \langle \nabla q_i(\bar{\boldsymbol{\theta}}_{\mathrm{init}}), \boldsymbol{h}_{\mathrm{ntk}}^* \rangle + \mathcal{O}(\alpha^{-L} \log A) \cdot e^{L\lambda t}$. One way to give a lower bound for $V_3$ is to consider the projection of $\frac{1}{n} \sum_{i=1}^n r_i(\boldsymbol{\theta}) \nabla q_i(\tfrac{1}{\alpha} e^{\lambda t} \boldsymbol{\theta})$ along $\boldsymbol{h}_{\mathrm{ntk}}^*$:

$$\begin{aligned}
V_3 &\geq \left\langle \frac{1}{n} \sum_{i=1}^n r_i(\boldsymbol{\theta}) \nabla q_i(\tfrac{1}{\alpha} e^{\lambda t} \boldsymbol{\theta}), \boldsymbol{h}_{\mathrm{ntk}}^* \right\rangle \\
&= \frac{1}{n} \sum_{i=1}^n r_i(\boldsymbol{\theta}) \left( \langle \nabla q_i(\bar{\boldsymbol{\theta}}_{\mathrm{init}}), \boldsymbol{h}_{\mathrm{ntk}}^* \rangle + \mathcal{O}(\alpha^{-L} \log A) \cdot e^{L\lambda t} \right) \\
&\geq \mathcal{L}(\boldsymbol{\theta}) \cdot \left( \gamma_{\mathrm{ntk}} + \mathcal{O}(\alpha^{-L} \log A) \cdot e^{L\lambda t} \right) > 0.
\end{aligned}$$

Another way to give a lower bound for $V_3$ is to consider the projection of $\frac{1}{n} \sum_{i=1}^n r_i(\boldsymbol{\theta}) \nabla q_i(\tfrac{1}{\alpha} e^{\lambda t} \boldsymbol{\theta})$ along $\boldsymbol{h}$:

$$V_3 \geq \left\langle \frac{1}{n} \sum_{i=1}^n r_i(\boldsymbol{\theta}) \nabla q_i(\tfrac{1}{\alpha} e^{\lambda t} \boldsymbol{\theta}), \frac{\boldsymbol{h}}{\|\boldsymbol{h}\|_2} \right\rangle.$$

Putting these two lower bounds together gives:

$$\begin{aligned}
V_1 &= \alpha^{2(L-1)} e^{-2(L-1)\lambda t} \frac{V_3^2}{\mathcal{L}(\boldsymbol{\theta})} \\
&\geq \left( \gamma_{\mathrm{ntk}} + \mathcal{O}(\alpha^{-L} \log A) \cdot e^{L\lambda t} \right) \alpha^{2(L-1)} e^{-2(L-1)\lambda t} \left\langle \frac{1}{n} \sum_{i=1}^n r_i(\boldsymbol{\theta}) \nabla q_i(\tfrac{1}{\alpha} e^{\lambda t} \boldsymbol{\theta}), \frac{\boldsymbol{h}}{\|\boldsymbol{h}\|_2} \right\rangle.
\end{aligned}$$

For $V_2$, by Lemma C.6, we have

$$
\begin{aligned}
V_2 &= \frac{\lambda}{\mathcal{L}(\boldsymbol{\theta})} \cdot \frac{1}{n} \sum_{i=1}^{n} r_i(\boldsymbol{\theta}) \langle \boldsymbol{\theta}, \nabla q_i(\boldsymbol{\theta}) \rangle \\
&= L\lambda \cdot \frac{1}{n\mathcal{L}(\boldsymbol{\theta})} \sum_{i=1}^{n} r_i(\boldsymbol{\theta}) q_i(\boldsymbol{\theta}) \\
&= L\lambda \cdot \frac{1}{n\mathcal{L}(\boldsymbol{\theta})} \sum_{i=1}^{n} r_i(\boldsymbol{\theta}) \left( \langle \nabla q_i(\bar{\boldsymbol{\theta}}_{\text{init}}), \boldsymbol{h} \rangle + \mathcal{O}(\alpha^{-L} \log A) \cdot e^{L\lambda t} \|\boldsymbol{h}\|_2 \right) \\
&\leq L\lambda \|\boldsymbol{h}\|_2 (\gamma_{\text{ntk}} + \mathcal{O}(\alpha^{-L} \log A) \cdot e^{L\lambda t}).
\end{aligned}
$$

Putting the bounds for $V_1$ and $V_2$ together proves (20). $\qquad\square$

Now we are ready to prove the directional convergence of $\boldsymbol{h}(T_{\text{c}}^{-}(\alpha); \alpha\bar{\boldsymbol{\theta}}_{\text{init}})$ to $\boldsymbol{h}_{\text{ntk}}^{*}$.

**Theorem C.9** *For any constant $c \in (0, 1)$, letting $T_{\text{c}}^{-}(\alpha) := \frac{1-c}{\lambda} \log \alpha$, it holds that*

$$
\lim_{\alpha \to +\infty} \frac{\boldsymbol{h}(T_{\text{c}}^{-}(\alpha); \alpha\bar{\boldsymbol{\theta}}_{\text{init}})}{\frac{1}{\gamma_{\text{ntk}}} \log \frac{\alpha^c}{\lambda}} = \boldsymbol{h}_{\text{ntk}}^{*}.
$$

**Proof:** Integrating (20) in Lemma C.8 over $t \in [0, T_{\text{c}}^{-}(\alpha)]$ gives

$$
\log \frac{1}{\mathcal{L}(\boldsymbol{\theta}(T_{\text{c}}^{-}(\alpha)))} \geq \left( \gamma_{\text{ntk}} + \mathcal{O}(\alpha^{-cL} \log A) \right) \|\boldsymbol{h}(T_{\text{c}}^{-}(\alpha))\|_2.
$$

By (8) in Lemma C.4 and (17) in Lemma C.7, we have

$$
\begin{aligned}
\log \frac{1}{\mathcal{L}(\boldsymbol{\theta}(T_{\text{c}}^{-}(\alpha)))} &= \log(F_{\text{LL}}^{-1}(\Theta(A)) - 1) - 2(L-1)\lambda T_{\text{c}}^{-}(\alpha) \\
&= \log A - 2(L-1)(1-c)\log \alpha + O(\log \log A) \\
&= \log \frac{\alpha^c}{\lambda} + O(\log \log A) \to +\infty.
\end{aligned}
$$

By Lemma C.6, $\log \frac{1}{\mathcal{L}(\boldsymbol{\theta}(T_{\text{c}}^{-}(\alpha)))} \leq O(\|\boldsymbol{h}(T_{\text{c}}^{-}(\alpha))\|_2)$, so we also have $\|\boldsymbol{h}(T_{\text{c}}^{-}(\alpha))\|_2 \to +\infty$.

Then

$$
\begin{aligned}
\gamma_{\text{ntk}} &\geq \frac{\min_{i\in[n]} \langle \nabla f_i(\bar{\boldsymbol{\theta}}_{\text{init}}), \boldsymbol{h} \rangle}{\|\boldsymbol{h}\|_2} \geq \frac{\min_{i\in[n]}\{q_i(\boldsymbol{\theta})\}}{\|\boldsymbol{h}\|_2} + \mathcal{O}(\alpha^{-L} \log A) \cdot e^{L\lambda T_{\text{c}}^{-}(\alpha)} \\
&\geq \frac{\log \left( \frac{1}{n} \sum_{i=1}^{n} \exp(-q_i(\boldsymbol{\theta})) \right)}{\|\boldsymbol{h}\|_2} + \mathcal{O}(\alpha^{-cL} \log A) \\
&\geq \frac{\log \frac{1}{\mathcal{L}(\boldsymbol{\theta})}}{\|\boldsymbol{h}\|_2} + \mathcal{O}(\alpha^{-cL} \log A) \to \gamma_{\text{ntk}}.
\end{aligned}
$$

Therefore, we have $\min_{i\in[n]} \left\langle \nabla q_i(\bar{\boldsymbol{\theta}}_{\text{init}}), \frac{\boldsymbol{h}(T_{\text{c}}^{-}(\alpha); \alpha\bar{\boldsymbol{\theta}}_{\text{init}})}{\|\boldsymbol{h}(T_{\text{c}}^{-}(\alpha); \alpha\bar{\boldsymbol{\theta}}_{\text{init}})\|_2} \right\rangle \to \gamma_{\text{ntk}}$. By the uniqueness of the max-margin solution of (K1), it must hold that $\frac{\boldsymbol{h}(T_{\text{c}}^{-}(\alpha); \alpha\bar{\boldsymbol{\theta}}_{\text{init}})}{\|\boldsymbol{h}(T_{\text{c}}^{-}(\alpha); \alpha\bar{\boldsymbol{\theta}}_{\text{init}})\|_2} \to \boldsymbol{h}_{\text{ntk}}^{*}$ as $\alpha \to +\infty$.

Then, we can approximate the norm of $h$ as

$$
\|\boldsymbol{h}\|_2 = \frac{\log \frac{1}{\mathcal{L}}}{\gamma_{\text{ntk}} + o(1)} = \frac{\log \frac{\alpha^c}{\lambda} + O(\log \log A)}{\gamma_{\text{ntk}} + o(1)} = \frac{1}{\gamma_{\text{ntk}}} \log \frac{\alpha^c}{\lambda} (1 + o(1)).
$$

So $h$ can be approximated by

$$
\boldsymbol{h} = \|\boldsymbol{h}\|_2 \frac{\boldsymbol{h}}{\|\boldsymbol{h}\|_2} = (1 + o(1)) \frac{1}{\gamma_{\text{ntk}}} \log \frac{\alpha^c}{\lambda} \boldsymbol{h}_{\text{ntk}}^{*},
$$

which completes the proof. $\qquad\square$

Now we are ready to prove Theorem 3.4.

**Proof:** For any input $x$, by a simple extension of Lemma C.6 to $f(\cdot; x)$ we have

$$f(\theta; x) = \langle \nabla f(\bar{\theta}_{\text{init}}; x), h \rangle + \mathcal{O}(\alpha^{-L} \log A) \cdot e^{L\lambda t} \|h\|_2.$$

Normalizing by $Z(\alpha)$ and taking limits, we have

$$\frac{f(\theta; x)}{Z(\alpha)} = \left\langle \nabla f(\bar{\theta}_{\text{init}}; x), \frac{h}{Z(\alpha)} \right\rangle + \mathcal{O}(\alpha^{-L} \log A) \cdot e^{L\lambda t} \cdot \frac{\|h\|_2}{Z(\alpha)} \to \langle \nabla f(\bar{\theta}_{\text{init}}; x), h_{\text{ntk}}^* \rangle,$$

which follows from Theorem C.9. $\qquad\square$

### C.3 RICH REGIME

Now we proceed to prove Theorem 3.5 for the rich regime. First, we derive the following bound for the training loss at the end of the kernel regime.

**Lemma C.10** *At $t = \frac{1}{\lambda}(\log \alpha - \frac{1}{L} \log \log A + \Delta T)$, we have $\mathcal{L}_\lambda(\theta(t)) = \mathcal{O}(\lambda(\log \frac{1}{\lambda})^{2/L})$.*

**Proof:** By Lemma C.4, we can easily obtain the bound

$$\mathcal{L}(\theta(t)) = \mathcal{O}(\tfrac{\log A}{A} \exp(2(L-1)(\log \alpha - \tfrac{1}{L} \log \log A))) = \mathcal{O}(\lambda(\log A)^{-1+2/L}).$$

Also we have the norm bound $\|\theta(t)\|_2 = \mathcal{O}(\alpha e^{-\lambda t}) = \mathcal{O}(\log A)^{1/L}$ since it is in the kernel regime. We can conclude the proof by noting that $\mathcal{L}_\lambda = \mathcal{L} + \|\theta\|_2^2$ and $\log A = O(\log \frac{1}{\lambda})$. $\qquad\square$

The loss bound then implies the following norm bound.

**Lemma C.11** *If $\mathcal{L}_\lambda(\theta(0)) = \mathcal{O}(\lambda(\log \frac{1}{\lambda})^{2/L})$, then for all $t \geq 0$,*

$$\|\theta(t)\|_2 = \Theta((\log \tfrac{1}{\lambda})^{1/L}).$$

**Proof:** For all $t \geq 0$, the following relationship can be deduced from the monotonicity of $\mathcal{L}_\lambda$, $r_i(\theta(t)) \leq n\mathcal{L}_\lambda(\theta(t))$, and the Lipschitzness of $f_i$:

$$\Omega(\log(1/\lambda)) \leq \log \frac{1}{n\mathcal{L}_\lambda(\theta(0))} \leq \log \frac{1}{n\mathcal{L}_\lambda(\theta(t))} \leq q_i(\theta(t)) \leq B_0 \|\theta(t)\|_2^L.$$

So $\|\theta(t)\|_2 = \Omega((\log \frac{1}{\lambda})^{1/L})$. Also note $\|\theta(t)\|_2^2 \leq \frac{2}{\lambda}\mathcal{L}_\lambda(\theta(t)) \leq \frac{2}{\lambda}\mathcal{L}_\lambda(\theta(0)) = \mathcal{O}((\log \frac{1}{\lambda})^{1/L})$. So we have the tight norm bound $\|\theta(t)\|_2 = \Theta((\log \frac{1}{\lambda})^{1/L})$. $\qquad\square$

To prove Theorem 3.5, it suffices to prove the following.

**Theorem C.12** *For any starting point $\theta_0(\alpha)$ satisfying $\mathcal{L}_\lambda(\theta_0(\alpha)) = O(\lambda(\log \frac{1}{\lambda})^{2/L})$ and a time function $T(\alpha) = \Omega(\frac{1}{\lambda} \log \frac{1}{\lambda})$, given any sequence $\{\alpha_k\}$ with $\alpha_k \to +\infty$, there exists a sequence $\{t_k\}_{k \geq 1}$ such that $t_k \leq T(\alpha)$, and every limit point of $\{\frac{\theta(t_k; \theta_0(\alpha_k))}{\|\theta(t_k; \theta_0(\alpha_k))\|_2} : k \geq 0\}$ is along the direction of a KKT point of (R1).*

**Proof:** For all $\alpha > 0$, $0 \leq \mathcal{L}_\lambda(\theta(T(\alpha))) = \mathcal{L}_\lambda(\theta_0) - \int_0^{T(\alpha)} \|\nabla \mathcal{L}_\lambda(\theta(t))\|_2^2 \, dt$. Since $\mathcal{L}_\lambda(\theta_0(\alpha)) = O(\lambda(\log A)^{2/L})$, we have the bound $\int_0^{T(\alpha)} \|\nabla \mathcal{L}_\lambda(\theta(t))\|_2^2 \, dt \leq \mathcal{O}(\lambda(\log \frac{1}{\lambda})^{2/L})$, which implies that there exists a time $\tau_\alpha$ such that $\|\nabla \mathcal{L}_\lambda(\theta(\tau_\alpha))\|_2^2 \leq \mathcal{O}(\lambda^2(\log \frac{1}{\lambda})^{-(1-2/L)})$.

Now let $t_k = \tau_{\alpha_k}$. It suffices to prove in the case where $\frac{\theta(t_k; \theta_0(\alpha_k))}{\|\theta(t_k; \theta_0(\alpha_k))\|_2} \in \mathcal{S}^{d-1}$ converges; otherwise we can do the same prove for any convergent subsequence. Denote $\theta(t_k; \theta_0(\alpha_k))$ as $\theta^{(k)}$ and $\frac{\theta(t_k; \theta_0(\alpha_k))}{\|\theta(t_k; \theta_0(\alpha_k))\|_2}$ as $\bar{\theta}^{(k)}$ for short. We claim that the limit $\bar{\theta} := \lim_{k \to +\infty} \bar{\theta}^{(k)}$ is along a KKT-margin direction of (R1).

First, according to our choice of $t_k$, we have

$$\left\| \frac{1}{n} \sum_{i=1}^n r_i(\theta^{(k)}) \nabla q_i(\theta^{(k)}) - \lambda \theta^{(k)} \right\|_2 \leq \mathcal{O}(\lambda(\log A)^{-(1/2-1/L)}). \qquad (21)$$

By Lemma C.11, $\lambda\|\boldsymbol{\theta}^{(k)}\|_2 = \mathcal{O}(\lambda(\log\frac{1}{\lambda})^{1/L})$. Now we divide $\lambda\|\boldsymbol{\theta}^{(k)}\|_2$ on both sides of (21):

$$\left\|\sum_{i=1}^{n}\frac{\|\boldsymbol{\theta}\|_2^{L-2}}{n\lambda}r_i(\boldsymbol{\theta}^{(k)})\nabla q_i(\bar{\boldsymbol{\theta}}^{(k)}) - \bar{\boldsymbol{\theta}}^{(k)}\right\|_2 \leq \mathcal{O}\left(\frac{(\log A)^{-(1/2-1/L)}}{(\log\frac{1}{\lambda})^{1/L}}\right). \tag{22}$$

Let $\boldsymbol{\mu}_k$ be a vector with coordinates $\mu_{k,i} := \frac{\|\boldsymbol{\theta}\|_2^{L-2}}{n\lambda}r_i(\boldsymbol{\theta}^{(k)})$. We can show that $\mu_{k,i} = \mathcal{O}(1)$ by noticing $f_i(\bar{\boldsymbol{\theta}}^{(k)}) = \frac{1}{\|\boldsymbol{\theta}\|_2^L}f_i(\boldsymbol{\theta}^{(k)}) = \Omega(1)$ and

$$\sum_{i=1}^{n}\mu_{k,i}y_if_i(\bar{\boldsymbol{\theta}}^{(k)}) \leq 1 + \left|\sum_{i=1}^{n}\mu_{k,i}y_if_i(\bar{\boldsymbol{\theta}}^{(k)}) - 1\right|$$

$$= 1 + \left|\left\langle\sum_{i=1}^{n}\mu_{k,i}y_i\nabla f_i(\bar{\boldsymbol{\theta}}^{(k)}) - \bar{\boldsymbol{\theta}}^{(k)}, \bar{\boldsymbol{\theta}}^{(k)}\right\rangle\right|$$

$$= 1 + \mathcal{O}\left(\frac{1}{\log\frac{1}{\lambda}}(\log A)^{-(1/2-1/L)}\right) = \mathcal{O}(1).$$

Then, let $\{(\alpha_{k_p}, t_{k_p})\}_{p\geq 1}$ be a subsequence of $\{(\alpha_k, t_k)\}_{k\geq 1}$ so that $\boldsymbol{\mu}_{k_p} \in \mathbb{R}^n$ converges. Let $\bar{\boldsymbol{\mu}} := \lim_{p\to+\infty}\boldsymbol{\mu}_{k_p}$ be the corresponding limit. We can take limit $k_p \to +\infty$ on both sides of (22) and obtain $\sum_{i=1}^{n}\bar{\mu}_iy_i\nabla f_i(\bar{\boldsymbol{\theta}}) = \bar{\boldsymbol{\theta}}$.

Let $i_* \in [n]$ be an index such that $q_{i_*}(\bar{\boldsymbol{\theta}}) = q_{\min}(\bar{\boldsymbol{\theta}})$. We can verify that $\bar{\mu}_i = 0$ for all $i \in [n]$ with $f_i(\bar{\boldsymbol{\theta}}) > q_{\min}(\bar{\boldsymbol{\theta}})$ by noting $\frac{\mu_{k,i}}{\mu_{k,i_*}} = \exp(q_{i_*}(\boldsymbol{\theta}^{(k)}) - f_i(\boldsymbol{\theta}^{(k)})) = \exp(\|\boldsymbol{\theta}\|_2^L(q_{i_*}(\bar{\boldsymbol{\theta}}) - q_i(\bar{\boldsymbol{\theta}}))) \to 0$ and $\bar{\boldsymbol{\mu}} \neq \mathbf{0}$. $\qquad\square$

Finally, Theorem 3.5 can be proved by simply combining Lemma C.10 with Theorem C.12.

# D  PROOFS FOR REGRESSION WITH L2 REGULARIZATION

The analyses for the regression task are similar to those for the classification task, but the use of squared loss simplifies the analysis.

We define $s_i(\boldsymbol{\theta}) := f_i(\boldsymbol{\theta}) - y_i$. In the regression setting, the gradients of the vanilla and regularized square loss can be written as

$$\nabla\mathcal{L}(\boldsymbol{\theta}) = 2\sum_{i=1}^{n} s_i \nabla f_i(\boldsymbol{\theta}), \tag{23}$$

and

$$\nabla\mathcal{L}_\lambda(\boldsymbol{\theta}) = 2\sum_{i=1}^{n} s_i \nabla f_i(\boldsymbol{\theta}) + \lambda\boldsymbol{\theta}. \tag{24}$$

## D.1  KERNEL REGIME

Now we present the proof of Theorem 3.9 for the kernel regime. Following Section 3.3.1, we define $\boldsymbol{h}_{\mathrm{ntk}}^*$ as the solution to (K2). Additionally, let $\boldsymbol{\Phi}(\boldsymbol{\theta}) \in \mathbb{R}^{D\times n}$ be the matrix where the $i$-th column is $\nabla f_i(\boldsymbol{\theta})$. Let $\boldsymbol{K}(\boldsymbol{\theta}) := \boldsymbol{\Phi}(\boldsymbol{\theta})^\top\boldsymbol{\Phi}(\boldsymbol{\theta})$ be the NTK matrix at $\boldsymbol{\theta}$. Let $\nu_{\mathrm{ntk}} > 0$ be the minimum eigenvalue of $\boldsymbol{K}(\bar{\boldsymbol{\theta}}_{\mathrm{init}})$. Let $\epsilon_{\max}$ be a small constant so that $\lambda_{\min}(\boldsymbol{K}(\boldsymbol{\theta})) > \frac{\nu_{\mathrm{ntk}}}{2}$ holds for all $\|\boldsymbol{\theta} - \bar{\boldsymbol{\theta}}_{\mathrm{init}}\|_2 < \epsilon_{\max}$. Let $T_{\max} := \inf\left\{t \geq 0 : \left\|\frac{e^{\lambda t}}{\alpha}\boldsymbol{\theta}(t) - \bar{\boldsymbol{\theta}}_{\mathrm{init}}\right\|_2 > \epsilon_{\max}\right\}$, which is the time that $\boldsymbol{\theta}(t)$ moves far enough to change the NTK features significantly.

First, we derive an upper bound for the loss convergence.

**Lemma D.1** *For all* $0 \leq t \leq T_{\max}$,

$$\frac{\mathrm{d}\mathcal{L}}{\mathrm{d}t} \leq -\frac{\nu_{\mathrm{ntk}}}{2n}\alpha^{2(L-1)}e^{-2(L-1)\lambda t}\mathcal{L} + \frac{\lambda L}{\sqrt{n}}\|\boldsymbol{y}\|_2\sqrt{\mathcal{L}}, \tag{25}$$

$$\mathcal{L}(\boldsymbol{\theta}(t)) \leq \max\left\{\frac{1}{n}\|\boldsymbol{y}\|_2^2\exp\left(-\frac{\nu_{\mathrm{ntk}}A}{8n(L-1)}\left(1 - e^{-2(L-1)\lambda t}\right)\right), \frac{16nL^2\|\boldsymbol{y}\|_2^2\lambda^2}{\nu_{\mathrm{ntk}}^2\alpha^{4(L-1)}}e^{4(L-1)\lambda t}.\right\}. \tag{26}$$

**Proof:**  By the update rule of $\boldsymbol{\theta}$, the following holds for all $t > 0$:

$$\frac{\mathrm{d}\mathcal{L}}{\mathrm{d}t} = \left\langle\nabla\mathcal{L}(\boldsymbol{\theta}), \frac{\mathrm{d}\boldsymbol{\theta}}{\mathrm{d}t}\right\rangle = \underbrace{-\left\|\frac{1}{n}\sum_{i=1}^{n}s_i\nabla f_i\right\|_2^2}_{=:V_1} \underbrace{-\frac{\lambda L}{n}\sum_{i=1}^{n}s_i f_i}_{=:V_2}.$$

For $V_1$, we have

$$-V_1 = \left\|\frac{1}{n}\sum_{i=1}^{n}r_i\nabla f_i(\boldsymbol{\theta})\right\|_2^2 = \frac{1}{n^2}\boldsymbol{s}(\boldsymbol{\theta})^\top\boldsymbol{K}(\boldsymbol{\theta})\boldsymbol{s}(\boldsymbol{\theta})$$

$$= \frac{1}{n^2}\alpha^{2(L-1)}e^{-2(L-1)\lambda t}\boldsymbol{s}(\boldsymbol{\theta})^\top\boldsymbol{K}(\tfrac{e^{\lambda t}}{\alpha}\boldsymbol{\theta})\boldsymbol{s}(\boldsymbol{\theta})$$

$$\geq \frac{\nu_{\mathrm{ntk}}}{2n^2}\alpha^{2(L-1)}e^{-2(L-1)\lambda t}\|\boldsymbol{s}(\boldsymbol{\theta})\|_2^2$$

$$= \frac{\nu_{\mathrm{ntk}}}{2n}\alpha^{2(L-1)}e^{-2(L-1)\lambda t}\mathcal{L}.$$

For $V_2$, we have $\frac{1}{n}\sum_{i=1}^{n}s_i f_i = \frac{1}{n}\sum_{i=1}^{n}(s_i^2 + s_i y_i) \geq \frac{1}{n}\sum_{i=1}^{n}s_i y_i \geq -\frac{1}{\sqrt{n}}\|\boldsymbol{y}\|_2\sqrt{\mathcal{L}}$. So $V_2 \leq \frac{\lambda L}{\sqrt{n}}\|\boldsymbol{y}\|_2\sqrt{\mathcal{L}}$. Combining the inequalities for $V_1$ and $V_2$ together proves (25).

For proving (26), we first divide by $\mathcal{L}$ on both sides of (25) to get

$$\frac{\mathrm{d}}{\mathrm{d}t}\log\mathcal{L} \leq -\frac{\nu_{\mathrm{ntk}}}{2n}\alpha^{2(L-1)}e^{-2(L-1)\lambda t} + \frac{\lambda L}{\sqrt{n}}\|\boldsymbol{y}\|_2 \cdot \frac{1}{\sqrt{\mathcal{L}}}.$$

Let $\mathcal{E} := \{t \in [0, T_{\max}] : \frac{\lambda L}{\sqrt{n}} \|\boldsymbol{y}\|_2 \cdot \frac{1}{\sqrt{\mathcal{L}}} \geq \frac{\nu_{\text{ntk}}}{4n} \alpha^{2(L-1)} e^{-2(L-1)\lambda t}\}$. For all $t \in [0, T_{\max}]$, if $t \in \mathcal{E}$, then

$$\mathcal{L} \leq \frac{\lambda^2 L^2}{n} \|\boldsymbol{y}\|_2^2 \cdot \frac{1}{\left(\frac{\nu_{\text{ntk}}}{4n} \alpha^{2(L-1)} e^{-2(L-1)\lambda t}\right)^2}$$

$$= \frac{16nL^2 \|\boldsymbol{y}\|_2^2}{\nu_{\text{ntk}}^2} \cdot \frac{\lambda^2}{\alpha^{4(L-1)}} e^{4(L-1)\lambda t}.$$

Otherwise, let $t'$ be the largest number in $\mathcal{E} \cup \{0\}$ that is smaller than $t$. Then

$$\log \mathcal{L}(\boldsymbol{\theta}(t)) = \log \mathcal{L}(\boldsymbol{\theta}(t')) + \int_{t'}^{t} \frac{\mathrm{d}}{\mathrm{d}\tau} \log \mathcal{L}(\boldsymbol{\theta}(\tau)) \, \mathrm{d}\tau$$

$$\leq \log \mathcal{L}(\boldsymbol{\theta}(t')) - \int_{t'}^{t} \left(\frac{\nu_{\text{ntk}}}{4n} \alpha^{2(L-1)} e^{-2(L-1)\lambda\tau}\right) \mathrm{d}\tau$$

$$\leq \log \mathcal{L}(\boldsymbol{\theta}(t')).$$

If $t' \in \mathcal{E}$, then $\mathcal{L}(\boldsymbol{\theta}(t)) \leq \mathcal{L}(\boldsymbol{\theta}(t')) \leq \frac{16nL^2 \|\boldsymbol{y}\|_2^2}{\nu_{\text{ntk}}^2} \cdot \frac{\lambda^2}{\alpha^{4(L-1)}} e^{4(L-1)\lambda t'} \leq \frac{16nL^2 \|\boldsymbol{y}\|_2^2}{\nu_{\text{ntk}}^2} \cdot \frac{\lambda^2}{\alpha^{4(L-1)}} e^{4(L-1)\lambda t}$. Otherwise, $t' = 0$ and we have

$$\mathcal{L}(\boldsymbol{\theta}(t)) \leq \mathcal{L}(\boldsymbol{\theta}(0)) \exp\left(-\int_0^t \left(\frac{\nu_{\text{ntk}}}{4n} \alpha^{2(L-1)} e^{-2(L-1)\lambda\tau}\right) \mathrm{d}\tau\right)$$

$$= \frac{1}{n} \|\boldsymbol{y}\|_2^2 \exp\left(-\frac{\nu_{\text{ntk}} A}{8n(L-1)} \left(1 - e^{-2(L-1)\lambda t}\right)\right)$$

which concludes the proof. $\qquad\square$

Now we derive a lower bound for the time that the dynamics stay in the kernel regime.

**Lemma D.2** *There exists a constant $\Delta T$ such that $T_{\max} \geq \frac{1}{\lambda}(\log\alpha + \Delta T)$ when $\alpha$ is sufficiently large. Furthermore, for all $t \leq \frac{1}{\lambda}(\log\alpha + \Delta T)$,*

$$\left\|\frac{1}{\alpha} e^{\lambda t} \boldsymbol{\theta}(t) - \bar{\boldsymbol{\theta}}_{\text{init}}\right\|_2 = \mathcal{O}(\alpha^{-L}) \cdot e^{L\lambda t}.$$

**Proof:** By definition of $T_{\max}$, to prove the first claim, it suffices to show that there exists a constant $\Delta T$ such that $\left\|\frac{1}{\alpha} e^{\lambda t} \boldsymbol{\theta}(t) - \bar{\boldsymbol{\theta}}_{\text{init}}\right\|_2 \leq \epsilon_{\max}$ for all $t \leq \min\{\frac{1}{\lambda}(\log\alpha + \Delta T), T_{\max}\}$.

When $t \leq T_{\max}$, we have the following gradient upper bound:

$$\|\nabla\mathcal{L}(\boldsymbol{\theta}(t))\|_2 = \left\|\frac{1}{n} \sum_{i=1}^{n} s_i \nabla f_i(\boldsymbol{\theta}(t))\right\|_2$$

$$\leq \frac{1}{n} \|\boldsymbol{s}(\boldsymbol{\theta})\|_2 \|\Phi(\boldsymbol{\theta})\|_2$$

$$= \frac{1}{\sqrt{n}} \alpha^{L-1} e^{-(L-1)\lambda t} \|\Phi(\tfrac{1}{\alpha} e^{-\lambda t}\boldsymbol{\theta})\|_2 \cdot \sqrt{\frac{1}{n}\|\boldsymbol{s}(\boldsymbol{\theta})\|_2^2}$$

$$\leq G_{\text{ntk}} \alpha^{L-1} e^{-(L-1)\lambda t} \sqrt{\mathcal{L}}.$$

By update rule, $e^{\lambda t}\boldsymbol{\theta}(t) - \alpha\bar{\boldsymbol{\theta}}_{\text{init}} = \int_0^t e^{\lambda\tau} \nabla\mathcal{L}(\boldsymbol{\theta}(\tau))\mathrm{d}\tau$. Applying the gradient upper bound gives

$$\frac{1}{\alpha}\left\|e^{\lambda t}\boldsymbol{\theta}(t) - \alpha\bar{\boldsymbol{\theta}}_{\text{init}}\right\|_2 \leq \frac{1}{\alpha}\int_0^t \left\|e^{\lambda\tau}\nabla\mathcal{L}(\boldsymbol{\theta}(\tau))\right\|_2 \mathrm{d}\tau$$

$$\leq \frac{1}{\alpha}\int_0^t e^{\lambda\tau} G_{\text{ntk}}\alpha^{L-1} e^{-(L-1)\lambda\tau} \sqrt{\mathcal{L}(\boldsymbol{\theta}(\tau))} \, \mathrm{d}\tau$$

$$= G_{\text{ntk}}\alpha^{L-2}\int_0^t e^{-(L-2)\lambda\tau}\sqrt{\mathcal{L}(\boldsymbol{\theta}(\tau))}\,\mathrm{d}\tau.$$

Now we substitute $\mathcal{L}$ in $\int_0^t e^{-(L-2)\lambda\tau}\sqrt{\mathcal{L}(\boldsymbol{\theta}(\tau))}\,\mathrm{d}\tau$ with the loss upper bound in Lemma D.1. Then for all $t \leq T_{\max}$, we have $\int_0^t e^{-(L-2)\lambda\tau}\sqrt{\mathcal{L}(\boldsymbol{\theta}(\tau))}\,\mathrm{d}\tau \leq \max\{I_1, I_2\}$, where

$$I_1 := \int_0^t \frac{1}{\sqrt{n}}\|\boldsymbol{y}\|_2 \exp\left(-\frac{\nu_{\mathrm{ntk}}A}{16n(L-1)}\left(1 - e^{-2(L-1)\lambda\tau}\right) - (L-2)\lambda\tau\right),$$

$$I_2 := \int_0^t \frac{4\sqrt{n}L\|\boldsymbol{y}\|_2}{\nu_{\mathrm{ntk}}} \cdot \frac{\lambda}{\alpha^{2(L-1)}}e^{L\lambda\tau}\mathrm{d}\tau.$$

We now bound $I_1$ and $I_2$. For $I_1$, we divide the integral into two parts. Let $t_0 = \frac{1}{20(L-1)\lambda}$, for $\tau \in [0, t_0]$, we have $1 - e^{-2(L-1)\lambda\tau} \geq \frac{1}{2} \cdot 2(L-1)\lambda\tau = (L-1)\lambda\tau$, so that

$$\int_0^{t_0} \exp\left(-\frac{\nu_{\mathrm{ntk}}A}{16n(L-1)}\left(1 - e^{-2(L-1)\lambda\tau}\right) - (L-2)\lambda\tau\right)$$

$$\leq \int_0^{t_0} \exp\left(-\frac{\nu_{\mathrm{ntk}}A\lambda\tau}{16n}\right)\mathrm{d}\tau = \mathcal{O}\left(\frac{1}{A\lambda}\right) = \mathcal{O}\left(\frac{1}{\alpha^{2(L-1)}}\right).$$

On the other hand, we have

$$\int_{t_0}^t \exp\left(-\frac{\nu_{\mathrm{ntk}}A}{16n(L-1)}\left(1 - e^{-2(L-1)\lambda\tau}\right) - (L-2)\lambda\tau\right)$$

$$\leq \exp\left(-\Omega(A)\right) \cdot \int_{t_0}^t \exp\left(-(L-2)\lambda\tau\right)\mathrm{d}\tau \leq \exp\left(-\Omega(A)\right) \cdot \mathcal{O}\left(\lambda^{-1}\right) = \mathcal{O}\left(\frac{1}{\alpha^{2(L-1)}}\right).$$

Thus, we have

$$I_1 = \mathcal{O}\left(\frac{1}{\alpha^{2(L-1)}}\right).$$

For $I_2$, it holds for all $t \leq T_{\max}$ that

$$I_2 = \mathcal{O}\left(\frac{1}{\alpha^{2(L-1)}}\right) \cdot e^{L\lambda t}.$$

Putting all these together, we have

$$\frac{1}{\alpha}\left\|e^{\lambda t}\boldsymbol{\theta}(t) - \alpha\bar{\boldsymbol{\theta}}_{\mathrm{init}}\right\|_2 = \mathcal{O}\left(\alpha^{L-2}\right) \cdot \mathcal{O}\left(\frac{1}{\alpha^{2(L-1)}}\right) \cdot e^{L\lambda t} \tag{27}$$

$$= \mathcal{O}\left(\alpha^{-L}\right) \cdot e^{L\lambda t}.$$

Therefore, there exists a constant $C > 0$ such that when $\alpha$ is sufficiently large, $\left\|\frac{1}{\alpha}e^{\lambda t}\boldsymbol{\theta}(t) - \bar{\boldsymbol{\theta}}_{\mathrm{init}}\right\|_2 \leq C\alpha^{-L}e^{L\lambda t}$ for all $t \leq \min\{T_{\max}, \mathcal{O}(\frac{1}{\lambda}\log\alpha)\}$. Setting $\Delta T := \frac{1}{L}\log\frac{\epsilon_{\max}}{C}$, we obtain the following bound for all $t \leq \min\{T_{\max}, \frac{1}{\lambda}(\log\alpha + \Delta T)\}$:

$$\left\|\frac{1}{\alpha}e^{\lambda t}\boldsymbol{\theta}(t) - \bar{\boldsymbol{\theta}}_{\mathrm{init}}\right\|_2 \leq C\alpha^{-L}\exp\left(L\log\alpha + \log\frac{\epsilon_{\max}}{C}\right) = \epsilon_{\max},$$

which implies $T_{\max} \geq \frac{1}{\lambda}(\log\alpha + \Delta T)$. Combining this with (13) prove the second claim. $\qquad\square$

Let $\boldsymbol{\delta}(t; \alpha\bar{\boldsymbol{\theta}}_{\mathrm{init}}) := \left(\frac{1}{\alpha}e^{\lambda t}\boldsymbol{\theta}(t; \alpha\bar{\boldsymbol{\theta}}_{\mathrm{init}}) - \bar{\boldsymbol{\theta}}_{\mathrm{init}}\right)$ and $\boldsymbol{h}(t; \alpha\bar{\boldsymbol{\theta}}_{\mathrm{init}}) := \alpha^L e^{-L\lambda t}\boldsymbol{\delta}(t; \alpha\bar{\boldsymbol{\theta}}_{\mathrm{init}})$. In the following, we show via a series of lemmas that for any $\boldsymbol{v}$ that is in the orthogonal complement of $\Phi$, $\frac{\langle\boldsymbol{\delta}, \boldsymbol{v}\rangle}{\|\boldsymbol{\delta}\|_2} \to 0$, which is a crucial property for deriving the implicit bias.

**Lemma D.3** *For any vector $\boldsymbol{v}$ that lies in the orthogonal space of the column space of $\Phi$, we have* $\left\langle\boldsymbol{\delta}(T_c^-(\alpha); \alpha\bar{\boldsymbol{\theta}}_{\mathrm{init}}), \boldsymbol{v}\right\rangle = \mathcal{O}\left(\alpha^{-2cL}\right)$ *where* $T_c^-(\alpha) := \frac{1-c}{\lambda}\log\alpha$.

**Proof:** By update rule,

$$\frac{\mathrm{d}\boldsymbol{\delta}}{\mathrm{d}t} = \frac{1}{\alpha}e^{\lambda t}\left(-2\sum_{i=1}^n s_i \nabla f_i(\boldsymbol{\theta}(t; \alpha\bar{\boldsymbol{\theta}}_{\mathrm{init}}))\right).$$

Then we have

$$
\begin{aligned}
\frac{\mathrm{d}\langle \boldsymbol{\delta}, \boldsymbol{v}\rangle}{\mathrm{d}t} &= -2\frac{1}{\alpha}e^{\lambda t}\left\langle \sum_{i=1}^{n} s_i \nabla f_i(\boldsymbol{\theta}(t; \alpha\bar{\boldsymbol{\theta}}_{\text{init}})), \boldsymbol{v}\right\rangle \\
&= -2\frac{1}{\alpha}e^{\lambda t}\left\langle \sum_{i=1}^{n} s_i \left(\nabla f_i(\boldsymbol{\theta}(t; \alpha\bar{\boldsymbol{\theta}}_{\text{init}})) - \nabla f_i(\alpha e^{-\lambda t}\bar{\boldsymbol{\theta}}_{\text{init}})\right), \boldsymbol{v}\right\rangle \\
&= -2\alpha^{L-2}e^{-(L-2)\lambda t}\left\langle \sum_{i=1}^{n} s_i \left(\nabla f_i(\alpha^{-1}e^{\lambda t}\boldsymbol{\theta}(t; \alpha\bar{\boldsymbol{\theta}}_{\text{init}})) - \nabla f_i(\bar{\boldsymbol{\theta}}_{\text{init}})\right), \boldsymbol{v}\right\rangle
\end{aligned}
$$

where the second equation is due to our choice of $\boldsymbol{v}$ and the last equation follows from the homogeneity of $f_i$. Now, since $f_i$ is locally smooth in a neighbourhood of $\bar{\boldsymbol{\theta}}_{\text{init}}$ and $\left\|\frac{1}{\alpha}e^{\lambda t}\boldsymbol{\theta}(t) - \bar{\boldsymbol{\theta}}_{\text{init}}\right\|_2 = \mathcal{O}(\alpha^{-L}) \cdot e^{L\lambda t}$ by Lemma D.2, we have

$$
\|\nabla f_i(\alpha^{-1}e^{\lambda t}\boldsymbol{\theta}(t; \alpha\bar{\boldsymbol{\theta}}_{\text{init}})) - \nabla f_i(\bar{\boldsymbol{\theta}}_{\text{init}})\|_2 = \mathcal{O}(\alpha^{-L}e^{L\lambda t}).
$$

Let $t_0 = \frac{1}{20(L-1)\lambda}$, then for $t \in [0, t_0]$, we have $1 - e^{2(L-1)\lambda\tau} \geq (L-1)\lambda\tau$, so that

$$
\begin{aligned}
\frac{\mathrm{d}\langle \boldsymbol{\delta}, \boldsymbol{v}\rangle}{\mathrm{d}t} &= \mathcal{O}\left(\alpha^{-2}e^{2\lambda t}\sqrt{\mathcal{L}\left(\boldsymbol{\theta}(t; \alpha\bar{\boldsymbol{\theta}}_{\text{init}})\right)}\right) \\
&= \mathcal{O}\left(\alpha^{-2}e^{2\lambda t}\max\left\{\exp\left(-\frac{\nu_{\text{ntk}}A\lambda t}{16n} + \lambda t\right), \alpha^{-2(L-1)}\lambda e^{2(L-1)\lambda t}\right\}\right) \\
&= \mathcal{O}\left(\max\left\{\alpha^{-2}\exp\left(-\frac{\nu_{\text{ntk}}A\lambda t}{32n}\right), \lambda\alpha^{-2L}e^{2L\lambda t}\right\}\right)
\end{aligned}
$$

for sufficiently large $\alpha$, since $A = \frac{\alpha^{2(L-1)}}{\lambda}$. On the other hand, for $t \in [t_0, T_c^-(\alpha)]$, the second term in (26) dominates the first term, so that

$$
\begin{aligned}
\frac{\mathrm{d}\langle \boldsymbol{\delta}, \boldsymbol{v}\rangle}{\mathrm{d}t} &= \mathcal{O}\left(\alpha^{-2}e^{2\lambda t}\sqrt{\mathcal{L}\left(\boldsymbol{\theta}(t; \alpha\bar{\boldsymbol{\theta}}_{\text{init}})\right)}\right) \\
&= \mathcal{O}\left(\alpha^{-2}e^{2\lambda t} \cdot \alpha^{-2(L-1)}\lambda e^{2(L-1)\lambda t}\right) = \mathcal{O}(\lambda\alpha^{-2L}e^{2L\lambda t}).
\end{aligned}
$$

Hence,

$$
\begin{aligned}
\langle \boldsymbol{\delta}(T_c^-(\alpha); \alpha\bar{\boldsymbol{\theta}}_{\text{init}}), \boldsymbol{v}\rangle &= \mathcal{O}\left(\max\left\{\alpha^{-2}\int_0^{T_c^-(\alpha)} \exp\left(-\frac{\nu_{\text{ntk}}A\lambda t}{32n}\right)\mathrm{d}t, \lambda\alpha^{-2L}\int_0^{T_c^-(\alpha)} e^{2L\lambda t}\mathrm{d}t\right\}\right) \\
&= \mathcal{O}\left(\max\left\{\alpha^{-2}\frac{1}{A\lambda}, \alpha^{-2L}\alpha^{2L(1-c)}\right\}\right) = \mathcal{O}(\alpha^{-2cL}),
\end{aligned}
$$

which proves the claim. $\qquad\square$

**Lemma D.4** *For $\boldsymbol{\theta} = \boldsymbol{\theta}(t)$ with $t \leq \frac{1}{\lambda}(\log \alpha + \Delta T)$ as defined in* Lemma D.2, *we have*

$$
f(\tfrac{1}{\alpha}e^{\lambda t}\boldsymbol{\theta}; \boldsymbol{x}) = \langle \nabla f(\bar{\boldsymbol{\theta}}_{\text{init}}; \boldsymbol{x}), \boldsymbol{\delta}\rangle + \mathcal{O}(\alpha^{-L}) \cdot e^{L\lambda t}\|\boldsymbol{\delta}\|_2, \tag{28}
$$
$$
f(\boldsymbol{\theta}; \boldsymbol{x}) = \langle \nabla f(\bar{\boldsymbol{\theta}}_{\text{init}}; \boldsymbol{x}), \boldsymbol{h}\rangle + \mathcal{O}(\alpha^{-L}) \cdot e^{L\lambda t}\|\boldsymbol{h}\|_2. \tag{29}
$$

**Proof:** By Lemma D.2 and Taylor expansion, we have

$$
\begin{aligned}
f(\tfrac{1}{\alpha}e^{\lambda t}\boldsymbol{\theta}; \boldsymbol{x}) &= f(\bar{\boldsymbol{\theta}}_{\text{init}}; \boldsymbol{x}) + \langle \nabla f(\bar{\boldsymbol{\theta}}_{\text{init}}; \boldsymbol{x}), \boldsymbol{\delta}\rangle + \mathcal{O}(\|\boldsymbol{\delta}\|_2^2) \\
&= f(\bar{\boldsymbol{\theta}}_{\text{init}}; \boldsymbol{x}) + \langle \nabla f(\bar{\boldsymbol{\theta}}_{\text{init}}; \boldsymbol{x}), \boldsymbol{\delta}\rangle + \mathcal{O}(\alpha^{-L}) \cdot e^{L\lambda t}\|\boldsymbol{\delta}\|_2 \\
&= \langle \nabla f(\bar{\boldsymbol{\theta}}_{\text{init}}; \boldsymbol{x}), \boldsymbol{\delta}\rangle + \mathcal{O}(\alpha^{-L}) \cdot e^{L\lambda t}\|\boldsymbol{\delta}\|_2,
\end{aligned}
$$

which proves (28). Combining this with the $L$-homogeneity of $q_i$ proves (29). $\qquad\square$

**Lemma D.5** $\|\boldsymbol{h}(T_c^-(\alpha); \alpha\bar{\boldsymbol{\theta}}_{\text{init}})\|_2 = \Theta(1)$.

**Proof:** By Lemma D.2 we have $\|\boldsymbol{\delta}(T_c^-(\alpha); \alpha\bar{\boldsymbol{\theta}}_{\text{init}})\|_2 = \mathcal{O}(e^{-cL})$, so that $\|\boldsymbol{h}(T_c^-(\alpha); \alpha\bar{\boldsymbol{\theta}}_{\text{init}})\|_2 = \alpha^L e^{-L\lambda T_c^-}\|\boldsymbol{\delta}(T_c^-(\alpha); \alpha\bar{\boldsymbol{\theta}}_{\text{init}})\|_2 = \mathcal{O}(1)$.

On the other hand, recall that $T_c^-(\alpha) = \frac{1-c}{\lambda}\log\alpha$, so by (26) we have

$$|s_i| \leq \sqrt{n\mathcal{L}(\boldsymbol{\theta}(T_c^-(\alpha); \alpha\bar{\boldsymbol{\theta}}_{\text{init}}))} = \mathcal{O}\left(\max\left\{e^{-\Omega(A)}, \lambda\alpha^{-2(L-1)c}\right\}\right) = \mathcal{O}\left(\alpha^{-2(L-1)c}\right).$$

Thus, for sufficiently large $\alpha$, we have $\left|f_i(\boldsymbol{\theta}(T_c^-(\alpha); \alpha\bar{\boldsymbol{\theta}}_{\text{init}}))\right| \geq \frac{1}{2}y_i$.

By Lemma D.4, we have that

$$\frac{1}{2}y_i \leq \|\nabla f_i(\bar{\boldsymbol{\theta}}_{\text{init}})\|_2\|\boldsymbol{h}\|_2 + \mathcal{O}(\alpha^{-cL})\|\boldsymbol{h}\|_2 \Rightarrow \|\boldsymbol{h}\|_2 = \Omega(1)$$

as desired. $\qquad\square$

**Corollary D.6** $\lim_{\alpha\to 0}\left\langle\frac{\boldsymbol{\delta}}{\|\boldsymbol{\delta}\|_2}(T_c^-(\alpha); \alpha\bar{\boldsymbol{\theta}}_{\text{init}}), \boldsymbol{v}\right\rangle = 0.$

**Proof:** By the previous lemma, we have $\|\boldsymbol{\delta}(T_c^-(\alpha); \alpha\bar{\boldsymbol{\theta}}_{\text{init}})\|_2 = \Omega\left(\alpha^{-L}e^{L\lambda T_c^-(\alpha)}\right) = \Omega(\alpha^{-cL})$. On the other hand, $\left\langle\boldsymbol{\delta}(T_c^-(\alpha); \alpha\bar{\boldsymbol{\theta}}_{\text{init}}), \boldsymbol{v}\right\rangle = \mathcal{O}\left(\alpha^{-2cL}\right)$ by Lemma D.3. The conclusion immediately follows. $\qquad\square$

**Theorem D.7** *For any constant $c \in (0, 1)$, letting $T_c^-(\alpha) := \frac{1-c}{\lambda}\log\alpha$, it holds that*

$$\forall\boldsymbol{x} \in \mathbb{R}^d: \qquad \lim_{\alpha\to+\infty} f(\boldsymbol{\theta}(T_c^-(\alpha); \alpha\bar{\boldsymbol{\theta}}_{\text{init}}); \boldsymbol{x}) = \left\langle\nabla f(\bar{\boldsymbol{\theta}}_{\text{init}}; \boldsymbol{x}), \boldsymbol{h}_{\text{ntk}}^*\right\rangle.$$

**Proof:** By Lemmas D.4 and D.5 we have

$$\begin{aligned}
&f(\boldsymbol{\theta}(T_c^-(\alpha); \alpha\bar{\boldsymbol{\theta}}_{\text{init}}); \boldsymbol{x})\\
&= \left\langle\nabla f(\bar{\boldsymbol{\theta}}_{\text{init}}; \boldsymbol{x}), \boldsymbol{h}(\boldsymbol{\theta}(T_c^-(\alpha); \alpha\bar{\boldsymbol{\theta}}_{\text{init}}))\right\rangle + \mathcal{O}(\alpha^{-L})\cdot e^{L\lambda T_c^-(\alpha)}\|\boldsymbol{h}(T_c^-(\alpha); \alpha\bar{\boldsymbol{\theta}}_{\text{init}})\|_2 \qquad (30)\\
&= \left\langle\nabla f(\bar{\boldsymbol{\theta}}_{\text{init}}; \boldsymbol{x}), \boldsymbol{h}(\boldsymbol{\theta}(T_c^-(\alpha); \alpha\bar{\boldsymbol{\theta}}_{\text{init}}))\right\rangle + \mathcal{O}(\alpha^{-cL})
\end{aligned}$$

By Lemma D.1 we have for $\forall i \in [n]$, $\left|f_i(\boldsymbol{\theta}(T_c^-(\alpha); \alpha\bar{\boldsymbol{\theta}}_{\text{init}})) - y_i\right| = \mathcal{O}(\alpha^{-2(L-1)})$. Since $y_i = \left\langle\nabla f_i(\bar{\boldsymbol{\theta}}_{\text{init}}), \boldsymbol{h}_{\text{ntk}}^*\right\rangle$, we have

$$\lim_{\alpha\to+\infty}\left\langle\nabla f_i(\bar{\boldsymbol{\theta}}_{\text{init}}), \boldsymbol{h}(\boldsymbol{\theta}(T_c^-(\alpha); \alpha\bar{\boldsymbol{\theta}}_{\text{init}})) - \boldsymbol{h}_{\text{ntk}}^*\right\rangle = 0. \qquad (31)$$

By Corollary D.6 and Lemma D.5 we have that for any direction $\boldsymbol{v}$ orthogonal to the column space of $\Phi$, we have $\lim_{\alpha\to+\infty}\left\langle\boldsymbol{h}(\boldsymbol{\theta}(T_c^-(\alpha); \alpha\bar{\boldsymbol{\theta}}_{\text{init}})), \boldsymbol{v}\right\rangle = 0$. Let $\boldsymbol{P}_\Phi$ be the projection operator onto the column space of $\Phi$, then

$$\lim_{\alpha\to+\infty}(\boldsymbol{I} - \boldsymbol{P}_\Phi)\boldsymbol{h}(\boldsymbol{\theta}(T_c^-(\alpha); \alpha\bar{\boldsymbol{\theta}}_{\text{init}})) = 0. \qquad (32)$$

Combined with (31), we deduce that

$$\lim_{\alpha\to+\infty}\left\langle\nabla f_i(\bar{\boldsymbol{\theta}}_{\text{init}}), \boldsymbol{P}_\Phi\boldsymbol{h}(\boldsymbol{\theta}(T_c^-(\alpha); \alpha\bar{\boldsymbol{\theta}}_{\text{init}})) - \boldsymbol{h}_{\text{ntk}}^*\right\rangle = 0.$$

The above holds for all $\nabla f_i(\bar{\boldsymbol{\theta}}_{\text{init}}), i \in [d]$, which are exactly the columns of $\Phi$. Note also that $\boldsymbol{h}_{\text{ntk}}^*$ also lies in the column space of $\Phi$, so we actually have $\lim_{\alpha\to+\infty}\|\boldsymbol{P}_\Phi\boldsymbol{h}(\boldsymbol{\theta}(T_c^-(\alpha); \alpha\bar{\boldsymbol{\theta}}_{\text{init}})) - \boldsymbol{h}_{\text{ntk}}^*\|_2 = 0$. Hence, we can use (32) to deduce that $\lim_{\alpha\to+\infty}\|\boldsymbol{h}(\boldsymbol{\theta}(T_c^-(\alpha); \alpha\bar{\boldsymbol{\theta}}_{\text{init}})) - \boldsymbol{h}_{\text{ntk}}^*\|_2 = 0$. Finally, plugging this into (30) gives the desired result. $\qquad\square$

### D.2 RICH REGIME

Now we proceed to prove Theorem 3.10 for the rich regime. First, we derive a norm bound.

**Lemma D.8** *For any constant $c > 0$, let $T_c^+(\alpha) := \frac{1+c}{\lambda}\log\alpha$, then we have that*

$$\max_{\frac{1}{\lambda}\log\alpha \leq t \leq T_c^+(\alpha)}\|\boldsymbol{\theta}\left(t; \alpha\bar{\boldsymbol{\theta}}_{\text{init}}\right)\|_2 = \mathcal{O}(1).$$

**Proof:** We know from Lemmas D.1 and D.2 that

$$\mathcal{L}\left(\boldsymbol{\theta}\left(\frac{1}{\lambda}\log\alpha;\alpha\bar{\boldsymbol{\theta}}_{\text{init}}\right)\right) \le \max\left\{\frac{1}{n}\|y\|_2^2\exp\left(-\frac{\nu_{\text{ntk}}A}{16n(L-1)}\right),\frac{16nL^2\|\boldsymbol{y}\|_2^2\lambda^2}{\nu_{\text{ntk}}^2}\right\} \le \mathcal{O}(\lambda^2).$$

Moreover, by Lemma D.2 we know that $\|\boldsymbol{\theta}\left(\frac{1}{\lambda}\log\alpha;\alpha\bar{\boldsymbol{\theta}}_{\text{init}}\right)\|_2 = \mathcal{O}(1)$, so that

$$\mathcal{L}_\lambda\left(\boldsymbol{\theta}\left(\frac{1}{\lambda}\log\alpha;\alpha\bar{\boldsymbol{\theta}}_{\text{init}}\right)\right) = \mathcal{L}\left(\boldsymbol{\theta}\left(\frac{1}{\lambda}\log\alpha;\alpha\bar{\boldsymbol{\theta}}_{\text{init}}\right)\right) + \frac{\lambda}{2}\left\|\boldsymbol{\theta}\left(\frac{1}{\lambda}\log\alpha;\alpha\bar{\boldsymbol{\theta}}_{\text{init}}\right)\right\|_2^2 = \mathcal{O}(\lambda).$$

Since $\left\{\boldsymbol{\theta}\left(t;\alpha\bar{\boldsymbol{\theta}}_{\text{init}}\right)\right\}_{t\ge0}$ is the trajectory of GF on $\mathcal{L}_\lambda$, we know that for $\forall t \ge \frac{1}{\lambda}\log\alpha$, we have $\mathcal{L}_\lambda\left(\boldsymbol{\theta}\left(t;\alpha\bar{\boldsymbol{\theta}}_{\text{init}}\right)\right) = \mathcal{O}(\lambda)$ as well. This implies that $\|\boldsymbol{\theta}\left(t;\alpha\bar{\boldsymbol{\theta}}_{\text{init}}\right)\|_2 = \mathcal{O}(1)$ as desired. $\qquad\square$

Now we prove Theorem 3.10.

**Proof:** By Lemma C.5, we know that for $t = \frac{1}{\lambda}\log\alpha$, we have $\|\boldsymbol{\theta}(t;\alpha\bar{\boldsymbol{\theta}}_{\text{init}}) - \bar{\boldsymbol{\theta}}_{\text{init}}\|_2 = \mathcal{O}(1) \Rightarrow \|\boldsymbol{\theta}(t;\alpha\bar{\boldsymbol{\theta}}_{\text{init}})\|_2 = \mathcal{O}(1)$. Since

$$\mathcal{L}_\lambda\left(\boldsymbol{\theta}\left(\frac{1}{\lambda}\log\alpha;\alpha\bar{\boldsymbol{\theta}}_{\text{init}}\right)\right) - \mathcal{L}_\lambda\left(\boldsymbol{\theta}\left(T_c^+(\alpha);\alpha\bar{\boldsymbol{\theta}}_{\text{init}}\right)\right) = \int_{\frac{1}{\lambda}\log\alpha}^{T_c^+(\alpha)}\|\nabla\mathcal{L}_\lambda(\boldsymbol{\theta}(t;\alpha\bar{\boldsymbol{\theta}}_{\text{init}}))\|_2^2\mathrm{d}t$$

and $0 \le \mathcal{L}_\lambda\left(\boldsymbol{\theta}\left(T_c^+(\alpha);\alpha\bar{\boldsymbol{\theta}}_{\text{init}}\right)\right), \mathcal{L}_\lambda\left(\boldsymbol{\theta}\left(\frac{1}{\lambda}\log\alpha;\alpha\bar{\boldsymbol{\theta}}_{\text{init}}\right)\right) \le \mathcal{O}(\lambda)$ by Lemma D.1, we deduce that there exists $t_\alpha \in \left[\frac{1}{\lambda}\log\alpha, T_c^+(\alpha)\right]$ such that $\left\|\nabla\mathcal{L}_\lambda(\boldsymbol{\theta}(t;\alpha\bar{\boldsymbol{\theta}}_{\text{init}}))\right\|_2 \le \mathcal{O}\left(\sqrt{\lambda\frac{c}{\lambda}\log\alpha}\right) = \mathcal{O}(\lambda(\log\frac{1}{\lambda})^{-1/2})$.

For any sequence $\{\alpha_k\}_{k\ge1}$ with $\alpha_k \to +\infty$, we choose $t_k = t_{\alpha_k}$. Let $\left\{\boldsymbol{\theta}(t_{i_k};\alpha_{i_k}\bar{\boldsymbol{\theta}}_{\text{init}}) : k \ge 1\right\}$ be any convergent subsequence of $\left\{\boldsymbol{\theta}(t_k;\alpha_k\bar{\boldsymbol{\theta}}_{\text{init}}) : k \ge 1\right\}$, then we have that $\lim_{k\to+\infty}\|\nabla\mathcal{L}_\lambda\left(\boldsymbol{\theta}(t_{i_k};\alpha\bar{\boldsymbol{\theta}}_{\text{init}})\right)\|_2 = 0$, *i.e.*,

$$\lim_{k\to+\infty}\left\|2\sum_{i=1}^n s_i\nabla f_i(\boldsymbol{\theta}(t_{i_k};\alpha_{i_k}\bar{\boldsymbol{\theta}}_{\text{init}})) + \lambda\boldsymbol{\theta}(t_{i_k};\alpha_{i_k}\bar{\boldsymbol{\theta}}_{\text{init}})\right\|_2 = 0. \tag{33}$$

By the choice of $\{i_k\}_{k\ge1}$ we know that $\left\{\boldsymbol{\theta}(t_{i_k};\alpha_{i_k}\bar{\boldsymbol{\theta}}_{\text{init}})\right\}_{k\ge1}$ converges to a point $\boldsymbol{\theta}^* \in \mathbb{R}^d$, so the above implies that

$$2\sum_{i=1}^n s_i\nabla f_i(\boldsymbol{\theta}^*) + \lambda\boldsymbol{\theta}^* = 0,$$

thus $\boldsymbol{\theta}^*$ is a KKT point of (R2), as desired. $\qquad\square$

# E   GENERALIZATION ANALYSES FOR LINEAR CLASSIFICATION

In this section, we provide supplementary generalization analyses for the linear classification experiments in Section 3.2.3. We follow the notations in Section 3.1. Let the input distribution $\mathcal{D}_X$, and the ground-truth label $y^*(\boldsymbol{x})$ be $\text{sign}(\boldsymbol{x}^\top \boldsymbol{w}^*)$ for an unknown $\boldsymbol{w}^* \in \mathbb{R}^d$. For a training set $S = (\boldsymbol{x}_1, \ldots, \boldsymbol{x}_n)$, we define $y_i := y^*(\boldsymbol{x}_i)$ for all $i \in [n]$.

## E.1   PRELIMINARIES

We base our generalization analyses on Rademacher complexity.

**Definition E.1** *Given a family of functions $\mathcal{F}$ mapping from $\mathbb{R}^d$ to $\mathbb{R}$, the empirical Rademacher complexity of $\mathcal{F}$ for a set of samples $S = (\boldsymbol{x}_1, \ldots, \boldsymbol{x}_n)$ is defined as*

$$\hat{\mathfrak{R}}_S(\mathcal{F}) = \mathbb{E}_{\sigma_1, \ldots, \sigma_n \sim \text{unif}\{\pm 1\}} \left[ \sup_{f \in \mathcal{F}} \frac{1}{n} \sum_{i=1}^n \sigma_i f(\boldsymbol{x}_i) \right]. \tag{34}$$

Rademacher complexity is useful in deriving margin-based generalization bounds for classification. The following bound is standard in the literature Koltchinskii and Panchenko (2002); Mohri et al. (2018).

**Theorem E.2 (Theorem 5.8, Mohri et al. (2018))** *Let $\mathcal{F}$ be a family of functions mapping from $\mathbb{R}^d \to \mathbb{R}$. $q > 0$. For any $\delta > 0$, with probability at least $1 - \delta$ over the random draw of $S = (\boldsymbol{x}_1, \ldots, \boldsymbol{x}_n) \sim \mathcal{D}_X^n$, the following bound for the test error holds for all $f \in \mathcal{F}$:*

$$\mathbb{E}_{x \sim \mathcal{D}_X}[\mathbb{1}_{[y^*(\boldsymbol{x})f(\boldsymbol{x}) \leq 0]}] \leq \frac{1}{n} \sum_{i=1}^n \mathbb{1}_{[y_i f(\boldsymbol{x}_i) \leq q]} + \frac{2}{q} \hat{\mathfrak{R}}_S(\mathcal{F}) + 3\sqrt{\frac{\log(2/\delta)}{2n}}. \tag{35}$$

Let $\mathcal{F}_p = \{\boldsymbol{x} \mapsto \langle \boldsymbol{w}, \boldsymbol{x} \rangle : \|\boldsymbol{w}\|_p \leq 1\}$ be a family of linear functions on $\mathbb{R}^d$ with bounded weight in $L^p$-norm. The following theorem from Awasthi et al. (2020) bounds the empirical Rademacher complexity of $\mathcal{F}_p$.

**Theorem E.3 (Direct Corollary of Corollary 3 in Awasthi et al. (2020))** *For a set of samples $S = (\boldsymbol{x}_1, \ldots, \boldsymbol{x}_n)$, the empirical Rademacher complexity of $\mathcal{F}_1$ and $\mathcal{F}_2$ is bounded as*

$$\hat{\mathfrak{R}}_S(\mathcal{F}_1) \leq \frac{1}{n} \sqrt{2\log(2d)} \cdot \left( \sum_{i=1}^n \|\boldsymbol{x}_i\|_\infty^2 \right)^{1/2}, \tag{36}$$

$$\hat{\mathfrak{R}}_S(\mathcal{F}_2) \leq \frac{1}{n} \left( \sum_{i=1}^n \|\boldsymbol{x}_i\|_2^2 \right)^{1/2}. \tag{37}$$

## E.2   GENERALIZATION BOUNDS FOR SPARSE LINEAR CLASSIFICATION

Let $\mathcal{D}_X$ be the uniform distribution on $\text{unif}\{\pm 1\}^d$. Let $k = \mathcal{O}(1)$ be a positive odd number. We draw $\boldsymbol{w}^*$ as follows: randomly draw the first $k$ coordinates from $\text{unif}\{\pm 1\}^k$, and set all the other coordinates to 0.

**Theorem E.4** *With probability at least $1 - \delta$ over the random draw of the training set $S = (\boldsymbol{x}_1, \ldots, \boldsymbol{x}_n)$, for any linear classifier $f : \mathbb{R}^d \to \mathbb{R}, \boldsymbol{x} \mapsto \langle \boldsymbol{w}, \boldsymbol{x} \rangle$ that maximizes the $L^1$-margin $\gamma_1(\boldsymbol{w}) := \min_{i \in [n]} \frac{y_i \langle \boldsymbol{w}, \boldsymbol{x}_i \rangle}{\|\boldsymbol{w}\|_1}$ on $S$, the following bound for the test error holds:*

$$\mathbb{E}_{x \sim \mathcal{D}_X}[\mathbb{1}_{[y^*(\boldsymbol{x})f(\boldsymbol{x}) \leq 0]}] \leq 4k \cdot \sqrt{\frac{2\log(2d)}{n}} + 3\sqrt{\frac{\log(2/\delta)}{2n}}. \tag{38}$$

**Proof:** It suffices to consider the case where $\|\boldsymbol{w}\|_1 = 1$ because rescaling $\boldsymbol{w}$ does not change the test error. Since $\boldsymbol{w} \in \arg\max \gamma_1(\boldsymbol{w})$, $\gamma_1(\boldsymbol{w}) \geq \gamma_1(\boldsymbol{w}^*) = \frac{1}{k}$. By definition, this implies

$y_i f(\boldsymbol{x}_i) \geq \frac{1}{k}$. Letting $q := \frac{1}{2k}$ and applying Theorem E.2, we have

$$\mathbb{E}_{x \sim \mathcal{D}_{\mathrm{X}}}[\mathbb{1}_{[y^*(\boldsymbol{x})f(\boldsymbol{x}) \leq 0]}] \leq 4k \cdot \hat{\mathfrak{R}}_S(\mathcal{F}_1) + 3\sqrt{\frac{\log(2/\delta)}{2n}}. \tag{39}$$

By Theorem E.3, $\hat{\mathfrak{R}}_S(\mathcal{F}_1) \leq \frac{1}{n}\sqrt{2\log(2d)} \cdot \sqrt{n} = \sqrt{\frac{2\log(2d)}{n}}$. Combining this with (41) completes the proof. □

**Theorem E.5** *With probability at least $1 - \delta$ over the random draw of the training set $S = (\boldsymbol{x}_1, \ldots, \boldsymbol{x}_n)$, for any linear classifier $f : \mathbb{R}^d \to \mathbb{R}, \boldsymbol{x} \mapsto \langle \boldsymbol{w}, \boldsymbol{x} \rangle$ that maximizes the $L^2$-margin $\gamma_2(\boldsymbol{w}) := \min_{i \in [n]} \frac{y_i \langle \boldsymbol{w}, \boldsymbol{x}_i \rangle}{\|\boldsymbol{w}\|_2}$ on $S$, the following bound for the test error holds:*

$$\mathbb{E}_{x \sim \mathcal{D}_{\mathrm{X}}}[\mathbb{1}_{[y^*(\boldsymbol{x})f(\boldsymbol{x}) \leq 0]}] \leq 4\sqrt{\frac{kd}{n}} + 3\sqrt{\frac{\log(2/\delta)}{2n}}. \tag{40}$$

**Proof:** It suffices to consider the case where $\|\boldsymbol{w}\|_2 = 1$ because rescaling $\boldsymbol{w}$ does not change the test error. Since $\boldsymbol{w} \in \arg\max \gamma_2(\boldsymbol{w})$, $\gamma_2(\boldsymbol{w}) \geq \gamma_2(\boldsymbol{w}^*) = \frac{1}{\sqrt{k}}$. By definition, this implies $y_i f(\boldsymbol{x}_i) \geq \frac{1}{\sqrt{k}}$. Letting $q := \frac{1}{2\sqrt{k}}$ and applying Theorem E.2, we have

$$\mathbb{E}_{x \sim \mathcal{D}_{\mathrm{X}}}[\mathbb{1}_{[y^*(\boldsymbol{x})f(\boldsymbol{x}) \leq 0]}] \leq 4\sqrt{k} \cdot \hat{\mathfrak{R}}_S(\mathcal{F}_2) + 3\sqrt{\frac{\log(2/\delta)}{2n}}. \tag{41}$$

By Theorem E.3, $\hat{\mathfrak{R}}_S(\mathcal{F}_2) \leq \frac{1}{n} \cdot \sqrt{nd} = \sqrt{\frac{d}{n}}$. Combining this with (41) completes the proof. □

# F    Proofs for Matrix Completion

In this section, we give the detailed proof of results in Section 3.3.3.

Let $\Omega = \{ \boldsymbol{x}_k = (i_k, j_k) : 1 \le k \le n \}$ and we have $n$ observations $\boldsymbol{P}_{\boldsymbol{x}_k} = \boldsymbol{e}_{i_k} \boldsymbol{e}_{j_k}^\top, 1 \le k \le n$. For convenience, we use $\boldsymbol{P}_k$ to denote $\boldsymbol{P}_{\boldsymbol{x}_k}$. Define $\boldsymbol{\theta}_t = (\boldsymbol{U}_t, \boldsymbol{V}_t)$, $\boldsymbol{W}_t = \boldsymbol{U}_t \boldsymbol{U}_t^\top - \boldsymbol{V}_t \boldsymbol{V}_t^\top$ and $f_i(\boldsymbol{\theta}) = \langle \boldsymbol{W}_t, \boldsymbol{P}_i \rangle$, and our goal is to minimize the function (5). Since $\boldsymbol{W}_t$ is symmetric, it is more convenient to replace $\boldsymbol{P}_i$ with $\frac{1}{2} \left( \boldsymbol{P}_i + \boldsymbol{P}_i^\top \right)$. From now on, we let $\boldsymbol{P}_i = \frac{1}{2} \left( \boldsymbol{e}_{i_i} \boldsymbol{e}_{j_i}^\top + \boldsymbol{e}_{j_i} \boldsymbol{e}_{i_i}^\top \right)$.

The key step is to show that the loss $\mathcal{L}$ defined in (5) satisfies the following two properties:

**Property F.1** *All local minima of $\mathcal{L}$ are global.*

**Property F.2** *At any saddle point $(\boldsymbol{U}_s, \boldsymbol{V}_s)$ of $\mathcal{L}$, there is a direction $(\boldsymbol{\mathcal{E}}_{\boldsymbol{U}}, \boldsymbol{\mathcal{E}}_{\boldsymbol{V}})$ such that*

$$\boldsymbol{vec}\, (\boldsymbol{\mathcal{E}}_{\boldsymbol{U}}, \boldsymbol{\mathcal{E}}_{\boldsymbol{V}})^\top \nabla^2 \mathcal{L}(\boldsymbol{U}_s, \boldsymbol{V}_s) \boldsymbol{vec}\, (\boldsymbol{\mathcal{E}}_{\boldsymbol{U}}, \boldsymbol{\mathcal{E}}_{\boldsymbol{V}}) < 0.$$

Given these two properties, we can deduce that GF with random initialization can converge to global minimizers almost surely.

**Theorem F.3** *The function $\mathcal{L}$ satisfies Properties F.1 and F.2.*

**Proof:**    Recall that

$$\mathcal{L}(\boldsymbol{U}, \boldsymbol{V}) = \frac{1}{n} \sum_{i=1}^n \left( \langle \boldsymbol{P}_i, \boldsymbol{U}\boldsymbol{U}^\top - \boldsymbol{V}\boldsymbol{V}^\top \rangle - y_i^* \right)^2 + \frac{\lambda}{2} \mathrm{tr}(\boldsymbol{U}\boldsymbol{U}^\top + \boldsymbol{V}\boldsymbol{V}^\top).$$

Define

$$\hat{\mathcal{L}}(\boldsymbol{W}) = \frac{1}{n} \sum_{i=1}^n (\langle \boldsymbol{P}_i, \boldsymbol{W} \rangle - y_i^*)^2 + \frac{\lambda}{2} \|\boldsymbol{W}\|_*, \tag{42}$$

then we have $\mathcal{L}(\boldsymbol{U}, \boldsymbol{V}) \ge \hat{\mathcal{L}} \left( \boldsymbol{U}\boldsymbol{U}^\top - \boldsymbol{V}\boldsymbol{V}^\top \right)$. Since $\hat{L}$ is convex, $\boldsymbol{W}$ is a global minimizer of $\hat{\mathcal{L}}$ if and only if

$$0 \in \partial \nabla \hat{\mathcal{L}}(\boldsymbol{W}) = \frac{2}{n} \sum_{i=1}^n (\langle \boldsymbol{P}_i, \boldsymbol{W} \rangle - y_i^*) \, \boldsymbol{P}_i + \frac{\lambda}{2} \partial \|\boldsymbol{W}\|_*. \tag{43}$$

Moreover, for any $\boldsymbol{W} \in \mathbb{R}^{d \times d}$ it holds that

$$\min_{\boldsymbol{U}\boldsymbol{U}^\top - \boldsymbol{V}\boldsymbol{V}^\top = \boldsymbol{W}} \mathcal{L}(\boldsymbol{U}, \boldsymbol{V}) = \hat{L}(\boldsymbol{W}). \tag{44}$$

now consider some $(\boldsymbol{U}_*, \boldsymbol{V}_*)$ such that

$$\nabla \mathcal{L}(\boldsymbol{U}_*, \boldsymbol{V}_*) = 0 \quad \text{and} \quad \nabla^2 \mathcal{L}(\boldsymbol{U}_*, \boldsymbol{V}_*) \succeq \mathbf{0}.$$

To prove the theorem's statement, we only need to show that $(\boldsymbol{U}_*, \boldsymbol{V}_*)$ is a global minimizer of $\mathcal{L}$. The subsequent proof is organized into two parts:

**Claim 1.** $\boldsymbol{W}^* = \boldsymbol{U}_* \boldsymbol{U}_*^\top - \boldsymbol{V}_* \boldsymbol{V}_*^\top$ is a global minimizer of $\hat{\mathcal{L}}$.

*Proof of Claim 1:* We check that the condition (43) holds at $\boldsymbol{W}^*$. First the first order condition imply that

$$\nabla_{\boldsymbol{U}} \mathcal{L}(\boldsymbol{U}_*, \boldsymbol{V}_*) = \left( \frac{4}{n} \sum_{i=1}^n (\langle \boldsymbol{P}_i, \boldsymbol{W}^* \rangle - y_i^*) \, \boldsymbol{P}_i + \lambda \boldsymbol{I} \right) \boldsymbol{U}_* = 0. \tag{45}$$

Similarly,

$$\left( -\frac{4}{n} \sum_{i=1}^n (\langle \boldsymbol{P}_i, \boldsymbol{W}^* \rangle - y_i^*) \, \boldsymbol{P}_i + \lambda \boldsymbol{I} \right) \boldsymbol{V}_* = 0. \tag{46}$$

Second, for any $\boldsymbol{\mathcal{E}} \in \mathbb{R}^{d \times d}$ we have

$$
\begin{aligned}
\left(\nabla_{\boldsymbol{U}}^2 \mathcal{L}(\boldsymbol{U}_*, \boldsymbol{V}_*)\right) \boldsymbol{\mathcal{E}} &= \lim_{t \to 0} \frac{1}{t} \left(\nabla_{\boldsymbol{U}} \mathcal{L}(\boldsymbol{U}_* + t\boldsymbol{\mathcal{E}}, \boldsymbol{V}_*) - \nabla_{\boldsymbol{U}} \mathcal{L}(\boldsymbol{U}_*, \boldsymbol{V}_*)\right) \\
&= \lim_{t \to 0} \frac{1}{t} \frac{4}{n} \sum_{i=1}^n \left[ \left( \left\langle (\boldsymbol{U}_* + t\boldsymbol{\mathcal{E}}) (\boldsymbol{U}_* + t\boldsymbol{\mathcal{E}})^\top - \boldsymbol{V}_* \boldsymbol{V}_*^\top, \boldsymbol{P}_i \right\rangle - y_i^* \right) \boldsymbol{P}_i (\boldsymbol{U}_* + t\boldsymbol{\mathcal{E}}) \right. \\
&\qquad \left. - \left( \langle \boldsymbol{P}_i, \boldsymbol{W}^* \rangle - y_i^* \right) \boldsymbol{P}_i \boldsymbol{U}_* \right] + \lambda \boldsymbol{\mathcal{E}} \\
&= \frac{4}{n} \sum_{i=1}^n \left[ \left( \langle \boldsymbol{P}_i, \boldsymbol{W}^* \rangle - y_i^* \right) \boldsymbol{P}_i \boldsymbol{\mathcal{E}} + 2 \left\langle \boldsymbol{U}_* \boldsymbol{\mathcal{E}}^\top, \boldsymbol{P}_i \right\rangle \boldsymbol{P}_i \boldsymbol{U}_* \right] + \lambda \boldsymbol{\mathcal{E}}.
\end{aligned}
\tag{47}
$$

where we use $\left\langle \boldsymbol{U}_* \boldsymbol{\mathcal{E}}^\top + \boldsymbol{\mathcal{E}} \boldsymbol{U}_*^\top, \boldsymbol{P}_i \right\rangle = 2 \left\langle \boldsymbol{U}_* \boldsymbol{\mathcal{E}}^\top, \boldsymbol{P}_i \right\rangle$ since $\boldsymbol{P}_i$ is symmetric. Similarly,

$$
\left(\nabla_{\boldsymbol{V}}^2 \mathcal{L}(\boldsymbol{U}_*, \boldsymbol{V}_*)\right) \boldsymbol{\mathcal{E}} = \frac{4}{n} \sum_{i=1}^n \left[ - \left( \langle \boldsymbol{P}_i, \boldsymbol{W}^* \rangle - y_i^* \right) \boldsymbol{P}_i \boldsymbol{\mathcal{E}} + 2 \left\langle \boldsymbol{V}_* \boldsymbol{\mathcal{E}}^\top, \boldsymbol{P}_i \right\rangle \boldsymbol{P}_i \boldsymbol{V}_* \right] + \lambda \boldsymbol{\mathcal{E}}
\tag{48}
$$

and

$$
\nabla_{\boldsymbol{V}} \nabla_{\boldsymbol{U}} \mathcal{L}(\boldsymbol{U}_*, \boldsymbol{V}_*) \boldsymbol{\mathcal{E}} = -\frac{8}{n} \sum_{i=1}^n \left\langle \boldsymbol{V}_* \boldsymbol{\mathcal{E}}^\top, \boldsymbol{P}_i \right\rangle \boldsymbol{P}_i \boldsymbol{U}_*
\tag{49}
$$

Let $\boldsymbol{M} = \frac{4}{n} \sum_{i=1}^n \left( \langle \boldsymbol{P}_i, \boldsymbol{W}^* \rangle - y_i^* \right) \boldsymbol{P}_i$. Since $\nabla^2 \mathcal{L}(\boldsymbol{U}_*, \boldsymbol{V}_*) \succeq \boldsymbol{0}$, for any $\boldsymbol{\mathcal{E}}_{\boldsymbol{U}}, \boldsymbol{\mathcal{E}}_{\boldsymbol{V}} \in \mathbb{R}^{d \times d}$ we have

$$
\begin{aligned}
0 &\leq \left\langle \boldsymbol{\mathcal{E}}_{\boldsymbol{U}}, \left(\nabla_{\boldsymbol{U}}^2 \mathcal{L}(\boldsymbol{U}_*, \boldsymbol{V}_*)\right) \boldsymbol{\mathcal{E}}_{\boldsymbol{U}} \right\rangle + \left\langle \boldsymbol{\mathcal{E}}_{\boldsymbol{V}}, \left(\nabla_{\boldsymbol{V}}^2 \mathcal{L}(\boldsymbol{U}_*, \boldsymbol{V}_*)\right) \boldsymbol{\mathcal{E}}_{\boldsymbol{V}} \right\rangle + 2 \left\langle \boldsymbol{\mathcal{E}}_{\boldsymbol{U}}, \nabla_{\boldsymbol{V}} \nabla_{\boldsymbol{U}} \mathcal{L}(\boldsymbol{U}_*, \boldsymbol{V}_*) \boldsymbol{\mathcal{E}}_{\boldsymbol{V}} \right\rangle \\
&= \left\langle \boldsymbol{\mathcal{E}}_{\boldsymbol{U}}, (\boldsymbol{M} + \lambda \boldsymbol{I}) \boldsymbol{\mathcal{E}}_{\boldsymbol{U}} \right\rangle + \left\langle \boldsymbol{\mathcal{E}}_{\boldsymbol{V}}, (-\boldsymbol{M} + \lambda \boldsymbol{I}) \boldsymbol{\mathcal{E}}_{\boldsymbol{V}} \right\rangle - \frac{16}{n} \sum_{i=1}^n \left\langle \boldsymbol{U}_* \boldsymbol{\mathcal{E}}_{\boldsymbol{U}}^\top, \boldsymbol{P}_i \right\rangle \left\langle \boldsymbol{V}_* \boldsymbol{\mathcal{E}}_{\boldsymbol{V}}^\top, \boldsymbol{P}_i \right\rangle \\
&\quad + \frac{8}{n} \sum_{i=1}^n \left( \left\langle \boldsymbol{V}_* \boldsymbol{\mathcal{E}}_{\boldsymbol{V}}^\top, \boldsymbol{P}_i \right\rangle^2 + \left\langle \boldsymbol{U}_* \boldsymbol{\mathcal{E}}_{\boldsymbol{U}}^\top, \boldsymbol{P}_i \right\rangle^2 \right).
\end{aligned}
\tag{50}
$$

We now consider two cases:

- $\max \{\text{rank} \boldsymbol{U}_*, \text{rank} \boldsymbol{V}_*\} = d$. We assume WLOG that $\text{rank} \boldsymbol{U}_* = d$, then (45) immediately implies that $\boldsymbol{M} + \lambda \boldsymbol{I} = 0$. Then $-\boldsymbol{M} + \lambda \boldsymbol{I} = 2\lambda \boldsymbol{I}$ and by (46) we have $\boldsymbol{V}_* = 0$. In this case, $\boldsymbol{W}^*$ is positive semi-definite, and (43) holds since $\boldsymbol{I} \in \partial \|\boldsymbol{W}^*\|_*$ by Lemma F.4.

- $\max \{\text{rank} \boldsymbol{U}_*, \text{rank} \boldsymbol{V}_*\} < d$. In this case, there exists non-zero vectors $\boldsymbol{a}, \boldsymbol{c} \in \mathbb{R}^d$ such that $\boldsymbol{U}_* \boldsymbol{a} = \boldsymbol{V}_* \boldsymbol{c} = 0$. For arbitrary vectors $\boldsymbol{b}, \boldsymbol{d} \in \mathbb{R}^d$, we set $\boldsymbol{\mathcal{E}}_{\boldsymbol{U}} = \boldsymbol{b} \boldsymbol{a}^\top$ and $\boldsymbol{\mathcal{E}}_{\boldsymbol{V}} = \boldsymbol{d} \boldsymbol{c}^\top$ in (50), so that

$$
\|\boldsymbol{a}\|^2 \boldsymbol{b}^\top (\boldsymbol{M} + \lambda \boldsymbol{I}) \boldsymbol{b} + \|\boldsymbol{c}\|^2 \boldsymbol{d}^\top (\lambda \boldsymbol{I} - \boldsymbol{M}) \boldsymbol{d} \geq 0.
$$

Since $\boldsymbol{b}, \boldsymbol{d}$ are arbitrarily chosen, the above implies that

$$
\boldsymbol{M} + \lambda \boldsymbol{I} \succeq 0, \quad \text{and} \quad \lambda \boldsymbol{I} - \boldsymbol{M} \succeq 0.
\tag{51}
$$

Since $\boldsymbol{M}$ is symmetric, there exists an orthogonal basis $\{\boldsymbol{f}_i : i \in [d]\}$ of $\mathbb{R}^d$ and corresponding eigenvalues $\boldsymbol{m}_i, i \in [d]$ such that $\boldsymbol{M} \boldsymbol{f}_i = \boldsymbol{m}_i \boldsymbol{f}_i$. We partition the set $[d]$ into three subsets:

$$
\mathcal{S}_- = \{i \in [d] : \boldsymbol{m}_i = -\lambda\}, \quad \mathcal{S}_+ = \{i \in [d] : \boldsymbol{m}_i = \lambda\}, \quad \mathcal{S}_0 = [d] - \mathcal{S}_- - \mathcal{S}_+.
$$

Since $(\boldsymbol{M} + \lambda \boldsymbol{I}) \boldsymbol{U}_* = 0$, we have $\boldsymbol{U}_*^\top \boldsymbol{f}_i = 0$ for all $i \in \mathcal{S}_+ \cup \mathcal{S}_0$. As a result, we can write

$$
\boldsymbol{U}_* \boldsymbol{U}_*^\top = \sum_{i,j \in \mathcal{S}_-} a_{ij} \boldsymbol{f}_i \boldsymbol{f}_j^\top, \quad a_{ij} \in \mathbb{R}.
$$

Similarly, we can write

$$
\boldsymbol{V}_* \boldsymbol{V}_*^\top = \sum_{i,j \in \mathcal{S}_+} a_{ij} \boldsymbol{f}_i \boldsymbol{f}_j^\top, \quad a_{ij} \in \mathbb{R}.
$$

Assume WLOG that $\mathcal{S}_- = [t]$ and $\mathcal{S}_+ = \{t+1, t+2, \cdots, s\}$, then Hence $\boldsymbol{W}^* = \sum_{i,j=1}^{t} a_{ij} \boldsymbol{f}_i \boldsymbol{f}_j^\top - \sum_{i,j=t+1}^{s} a_{ij} \boldsymbol{f}_i \boldsymbol{f}_j^\top$. note that the matrices $(a_{ij})_{i,j\in\mathcal{S}_-}$ and $(a_{ij})_{i,j\in\mathcal{S}_+}$ are both positive semi-definite, so there exists $b_i \geq 0, i \in [d]$ and an orthogonal basis $\{\boldsymbol{g}_i : i \in [d]\}$ such that

$$\sum_{i,j=1}^{t} a_{ij} \boldsymbol{f}_i \boldsymbol{f}_j^\top = \sum_{i=1}^{t} b_i \boldsymbol{g}_i \boldsymbol{g}_i^\top \quad \text{and} \quad \sum_{i,j=t+1}^{s} a_{ij} \boldsymbol{f}_i \boldsymbol{f}_j^\top = \sum_{i=t+1}^{s} b_i \boldsymbol{g}_i \boldsymbol{g}_i^\top \tag{52}$$

and $b_i = 0$ and $\boldsymbol{g}_i = \boldsymbol{f}_i$ for $i > s$. now let

$$\boldsymbol{G}_1 = (\boldsymbol{g}_i : i \in [d]), \quad \boldsymbol{G}_2 = \left((-1)^{\mathbb{I}\{i>t\}} \boldsymbol{g}_i : i \in [d]\right), \quad \text{and} \quad \boldsymbol{B} = \mathrm{diag}|b_i| : i \in [d],$$

then both $\boldsymbol{G}_1$ and $\boldsymbol{G}_2$ are orthonormal matrices and we have $\boldsymbol{W}^* = \boldsymbol{G}_1 \boldsymbol{B} \boldsymbol{G}_2^\top$. This gives a singular value decomposition of $\boldsymbol{W}^*$. By Lemma F.4 we can write down the explicit form of $\partial \|\boldsymbol{W}^*\|_*$ as follows:

$$\partial \|\boldsymbol{W}^*\|_* = \left\{ \sum_{i\in\mathcal{T}} (-1)^{\mathbb{I}\{i>t\}} \boldsymbol{g}_i \boldsymbol{g}_i^\top + \boldsymbol{E} : \|\boldsymbol{E}\| \leq 1 \text{ and } \boldsymbol{E}\boldsymbol{g}_i = 0, \forall i \in \mathcal{T} \right\}.$$

Here $\mathcal{T} = \{i \in [d] : b_i \neq 0\} \subset [s]$. We choose

$$\begin{aligned}
\boldsymbol{E} &= -\left( \lambda^{-1} \boldsymbol{M} + \sum_{i\in\mathcal{T}} (-1)^{\mathbb{I}\{i>t\}} \boldsymbol{g}_i \boldsymbol{g}_i^\top \right) \\
&= -\lambda^{-1} \left( -\lambda \sum_{i=1}^{t} \boldsymbol{f}_i \boldsymbol{f}_i^\top + \lambda \sum_{i=t+1}^{s} \boldsymbol{f}_i \boldsymbol{f}_i^\top + \sum_{i>s} m_i \boldsymbol{f}_i \boldsymbol{f}_i^\top \right) - \sum_{i\in\mathcal{T}} (-1)^{\mathbb{I}\{i>t\}} \boldsymbol{g}_i \boldsymbol{g}_i^\top \\
&= -\lambda^{-1} \left( -\lambda \sum_{i=1}^{t} \boldsymbol{g}_i \boldsymbol{g}_i^\top + \lambda \sum_{i=t+1}^{s} \boldsymbol{g}_i \boldsymbol{g}_i^\top + \sum_{i>s} m_i \boldsymbol{g}_i \boldsymbol{g}_i^\top \right) - \sum_{i\in\mathcal{T}} (-1)^{\mathbb{I}\{i>t\}} \boldsymbol{g}_i \boldsymbol{g}_i^\top \\
&= \sum_{i\in[s]-\mathcal{T}} (-1)^{\mathbb{I}\{i>t\}} \boldsymbol{g}_i \boldsymbol{g}_i^\top - \lambda^{-1} \sum_{i>s} m_i \boldsymbol{g}_i \boldsymbol{g}_i^\top
\end{aligned}$$

where the third equation holds because of the definition (52) and $\boldsymbol{g}_i = \boldsymbol{f}_i$ when $i > s$. Since by (51) we have $m_i \in [-\lambda, \lambda]$ for all $i \in [d]$, the above expression of $\boldsymbol{E}$ immediately implies that $\|\boldsymbol{E}\| \leq 1$. Moreover, we obviously have $\boldsymbol{E}\boldsymbol{g}_i = 0$ when $i \in \mathcal{T}$ (since $\{\boldsymbol{g}_i\}$ is orthogonal). Hence, we have $0 \in \boldsymbol{M} + \lambda \partial \|\boldsymbol{W}^*\|$.

To summarize, we have shown that (43) always holds at $\boldsymbol{W}^*$. Therefore, $\boldsymbol{W}^*$ is a global minimizer of $\hat{\mathcal{L}}$ which concludes the proof of **Claim 1**.

**Claim 2.** $(\boldsymbol{U}_*, \boldsymbol{V}_*)$ solves the problem

$$\text{minimize} \quad \|\boldsymbol{U}\|_F^2 + \|\boldsymbol{V}\|_F^2 \quad s.t. \quad \boldsymbol{U}\boldsymbol{U}^\top - \boldsymbol{V}\boldsymbol{V}^\top = \boldsymbol{W}^*. \tag{53}$$

*Proof of Claim 2.* We will use the notations in the proof of Claim 1 for convenience. note that

$$\|\boldsymbol{U}_*\|_F^2 = \mathrm{tr}\boldsymbol{U}_*\boldsymbol{U}_*^\top = \sum_{i=1}^{t} b_i$$

and similarly

$$\|\boldsymbol{V}_*\|_F^2 = \sum_{i=t+1}^{s} b_i.$$

Thus $\|\boldsymbol{U}_*\|_F^2 + \|\boldsymbol{V}_*\|_F^2 = \sum_{i=1}^{s} b_i = \|\boldsymbol{W}\|_*$. On the other hand, for any $(\boldsymbol{U}, \boldsymbol{V})$ satisfying $\boldsymbol{U}\boldsymbol{U}^\top - \boldsymbol{V}\boldsymbol{V}^\top = \boldsymbol{W}^*$, we have

$$\|\boldsymbol{W}^*\|_* = \left\|\boldsymbol{U}\boldsymbol{U}^\top - \boldsymbol{V}\boldsymbol{V}^\top\right\|_* \leq \left\|\boldsymbol{U}\boldsymbol{U}^\top\right\|_* + \left\|\boldsymbol{V}\boldsymbol{V}^\top\right\|_* = \|\boldsymbol{U}\|_F^2 + \|\boldsymbol{V}\|_F^2.$$

Hence $(\boldsymbol{U}_*, \boldsymbol{V}_*)$ is a minimizer of (53), as desired.

We can now prove that $(\boldsymbol{U}_*, \boldsymbol{V}_*)$ is a global minimizer of $\mathcal{L}$, which completes the proof of Theorem F.3. Indeed, for any $(\boldsymbol{U}, \boldsymbol{V})$ we have

$$\mathcal{L}(\boldsymbol{U}, \boldsymbol{V}) \geq \hat{\mathcal{L}}(\boldsymbol{U}\boldsymbol{U}^\top - \boldsymbol{V}\boldsymbol{V}^\top) \geq \hat{\mathcal{L}}(\boldsymbol{W}^*) = \mathcal{L}(\boldsymbol{U}_*, \boldsymbol{V}_*).$$

The last step follows from **Claim 2** and (44). $\qquad\square$

**Lemma F.4** *(Watson, 1992, Example 2) Let $\boldsymbol{A} = \boldsymbol{L}\boldsymbol{\Sigma}\boldsymbol{R}^\top$ be its singular value decomposition, then we have*

$$\partial\|\boldsymbol{A}\|_* = \left\{ \boldsymbol{L}\boldsymbol{R}^\top + \boldsymbol{E} : \|\boldsymbol{E}\| \leq 1, \boldsymbol{L}^\top\boldsymbol{E} = 0 \text{ and } \boldsymbol{E}\boldsymbol{R} = 0 \right\}.$$

It is well-known that under cerntain regularity conditions on the ground-truth matrix $\boldsymbol{X}^*$, the global minimizer of the convex problem (42) is close to the ground-truth $\boldsymbol{X}^*$. Here we present a version of this result adapted from (Candes and Plan, 2010, Theorem 7).

**Theorem F.5** *Suppose that* $\mathrm{rank}(\boldsymbol{X}^*) = r = \mathcal{O}(1)$, $\boldsymbol{X}^* = \boldsymbol{V}_{\boldsymbol{X}^*}\boldsymbol{\Sigma}_{\boldsymbol{X}^*}\boldsymbol{V}_{\boldsymbol{X}^*}^\top$ *is (a version of) its SVD, and each row of* $\boldsymbol{V}_{\boldsymbol{X}^*}$ *has $\ell_\infty$-norm bounded by* $\sqrt{\frac{\mu}{d}}$, *then if the number of observed entries satisfies* $n \gtrsim \mu^4 d \log^2 d$, *then we have that* $\|\boldsymbol{W}^* - \boldsymbol{X}^*\|_F \lesssim \sqrt{\lambda\|\boldsymbol{X}^*\|_*}\mu^2 \log d$ *with probability* $\geq 1 - d^{-3}$.

