# OpenReview forum: "Dichotomy of Early and Late Phase Implicit Biases Can Provably Induce Grokking"
_ICLR.cc/2024/Conference — ICLR 2024 poster_

### Official Review · Reviewer_vAjC · 2023-10-16

**Soundness:** 3 good
**Presentation:** 3 good
**Contribution:** 3 good
**Rating:** 6
**Confidence:** 3

**Summary:**

The authors seek to explain the phenomenon of grokking highlighted by Power, Burda, Edwards, Babuschkin, and Misra in arXiv 2201.02177, focusing on a theoretical analysis, and paying special attention to sharpness of the transition from memorisation to generalisation.  The central thesis in this paper, which is summarised in its title, is that grokking occurs due to different implicit biases in early and late training.  Several theoretical results in support of this are proved for training by gradient flow homogeneous neural networks with large initialisations and small weight decay, for classification as well as regression tasks.  The pattern is that general theorems are proved first, and then corollaries are derived that explain grokking in specific settings, such as classification using diagonal linear networks, or low-rank symmetric matrix completion.  In the final section, the authors consider other scenarios where grokking occurs; they provide numerical experiments without weight decay for learning modular addition using two-layer ReLU networks, and they show how theoretical results from Li, Wang, and Arora in ICLR 2021 confirm grokking when training two-layer diagonal linear networks by label noise SGD with a sufficiently small learning rate.

**Strengths:**

The paper seems to be a first one to succeed in proving rigorously that grokking occurs in relatively general settings, and in showing interesting upper bounds on the length of the transition from memorisation to generalisation.

The explanation of grokking in terms of different implicit biases in early and late phases of training is elegant, and the restrictions to homogeneous neural networks, large initialisation, small weight decay, and gradient flow are reasonable and motivated by the previous empirical work of Liu, Michaud, and Tegmark in ICLR 2023.  Moreover, the versatility of the explanation in terms of the implicit biases is demonstrated by the identification of "misgrokking" in this paper, a dual phenomenon in which the test accuracy transitions from very high to very low late in the training.

Grokking and misgrokking as indicated by the theory are verified by numerical experiments for classification using two-layer diagonal linear networks.

Full proofs of the theoretical results are provided in the appendix.  The biggest and most novel are those of Theorems 3.3 and 3.7, which show that the small weight decay does not spoil the kernel regime and its implicit bias early in the training for long enough so that a relatively short transitional phase follows.

**Weaknesses:**

The paper comes across as dense and can be difficult to read.  One reason is that a number of settings for theory and experiments are considered.  Another is that the statements of many of the theoretical results are relatively complex, with several quantifications and subtle details, and often without much explanation.  A minor comment is that I think that writing "for all" or "for every" is clearer than "for any".

The theoretical results about grokking for regression tasks, and in particular for overparameterised matrix completion, are not illustrated experimentally.

The conclusion of the paper (Section 5) does not seem to indicate any directions for future work.

**Questions:**

In the first sentence in "Our Contributions", it is claimed that homogeneous networks allow the ReLU activation, however homogeneity is then assumed together with $\mathcal{C}^2$-smoothness (in Assumption 3.1), which I believe disallows the ReLU activation due to its non-differentiability at 0?

Code was not submitted with the paper, only the appendix with proofs.  Are you able to supplement the paper with code, so that readers are able to reproduce your numerical experiments?

What is the purpose of the section on sharpness reduction with label noise (Section 4.2), since it mostly consists of statements of results from the paper of Li, Wang, and Arora in ICLR 2021, i.e. it does not seem to contain much new material?

---

> ### Author Response · Authors · 2023-11-23
> **Response to Reviewer vAjC**
>
> **Q1:** The paper comes across as dense and can be difficult to read. One reason is that a number of settings for theory and experiments are considered. Another is that the statements of many of the theoretical results are relatively complex, with several quantifications and subtle details, and often without much explanation.
>
> **A:** Thanks for pointing out this issue! We have polished the writing and changed the organization in the revised version. Now we have included a lot of explanations for our assumptions and theorems.
>
> **Q2:** The theoretical results about grokking for regression tasks, and in particular for overparameterised matrix completion, are not illustrated experimentally.
>
> **A:** Thanks for the suggestion. In the revised version, we have added an experiment on matrix completion for the multiplication table; please see the “Empirical validation” paragraph of Sec. 3.3.3 for more details.
>
> **Q3:** The conclusion of the paper (Section 5) does not seem to indicate any directions for future work.
>
> **A:** Thanks for pointing this out. We have now added two possible future directions in the last section. First, our work only studies the training dynamics with large initialization and small weight decay, but these may not be the only source of the dichotomy of the implicit biases. Also, our work focuses on understanding the cause of grokking but does not study how to make neural nets generalize without so much delay in time. We leave it to future work to explore the other sources of grokking and practical methods to eliminate grokking.
>
> **Q4:** In the first sentence in "Our Contributions", it is claimed that homogeneous networks allow the ReLU activation, but Assumption 3.1 assumes smoothness
>
> **A:** Thanks for pointing this out. We have clarified this issue in the paragraph following Assumption 3.1. Indeed, the specific examples on sparse linear classification and matrix completion that we consider in this paper satisfy this assumption, but for non-smooth functions, we believe that our analysis can be generalized by invoking clarke differential, similar to previous works on homogeneous nets (Lyu and Li, 2020; Ji and Telgarsky, 2020).
>
> Kaifeng Lyu and Jian Li. Gradient descent maximizes the margin of homogeneous neural networks.In International Conference on Learning Representations, 2020.
>
> Ziwei Ji and Matus Telgarsky. Directional convergence and alignment in deep learning.
> Advances in Neural Information Processing Systems, 33:17176–17186, 2020a.
>
> **Q5:** Code was not submitted with the paper.
>
> **A:** Thanks for pointing it out. We have included our code in the supplementary material.
>
> **Q6:** What is the purpose of the section on sharpness reduction with label noise (Section 4.2), since it mostly consists of statements of results from Li et al. (2021)?
>
> **A:** The main purpose of this section is to point out that sharpness reduction is another form of implicit bias that provably causes grokking. Although this section does not include original theoretical contributions, we still think it valuable to notice the connection between the grokking phenomenon and previous works.

---

> > ### Comment · Reviewer_vAjC · 2023-11-23
> >
> > Thank you very much for these responses.

---

### Official Review · Reviewer_oyWf · 2023-10-26

**Soundness:** 2 fair
**Presentation:** 3 good
**Contribution:** 3 good
**Rating:** 6
**Confidence:** 3

**Summary:**

The paper studies an empirical phenomenon called "Grokking" where-in neural networks can display delayed generalization under certain conditions. The paper puts forth the hypothesis that the observed memorization and eventual transition to generalization can be attributed to the optimization occurring two different inductive biases that are separated by a transition at a certain time (step). The paper studies training of neural network under large initialization and weight decay and suggests that for the first block of time the training occurs in a "kernel regime" before transitioning to "rich regime". The paper studies behavior under classification and regression settings using diagonal linear networks as a concrete example

**Strengths:**

- The paper attempts to explain "Grokking" using analytical tools. As an empirical researcher, I found this to be a good reminder that theory can shed light on even perplexing phenomena ``Grokking''
- The paper implicitly summarizes the various conditions under which ``Grokking'' has been observed and reported in practice
- The paper's hypothesis of two separate indicative biases separated by transition at a certain time makes intuitive sense
- The paper empirically constructs examples that empirically  demonstrate both Grokking and mis-Grokking by manipulating the inductive biases while creating datasets

**Weaknesses:**

The biggest concern that I have is that I do not quite see the core claim os "sharp transition in test accuracy" being supported by any of the analysis made in the paper. The paper states theorems in the main paper that show that two different inductive biases are at work and provide corollaries for diagonal linear networks. However, I do not see any statements, analysis or proofs in the main paper for generalization risks (or make generalization guarantees).

Without the above, the best one can conclude is that the models can fit the training data and reach perfect training accuracy eventually but this line of work has already been done by Moroshko et. al (Moroshko). I may have missed something obvious so I look forward to the rebuttal to improve my understanding of the paper

However, at this point, I am afraid I can't really support accepting this paper.

[Moroshko] https://arxiv.org/abs/2007.06738

**Questions:**

- In Section 3.1.3 on page 5 of the draft, the paper claims that implicit biases imply transition in test accuracy. As noted in the weakness, this is not supported by any analysis (bounds etc).
- No empirical results are provided that helps the reader crystallize their understanding of the work in terms of when the transition might occur in practice. Can the authors add more examples in text with datasets commonly used in ``Grokking'' literature?
- One point that the paper makes is about identifying a transition time from memorization to generalization and attributes it to different inductive biases. This is not true when weight decay = 0 which is a case handled that the paper talks about as well. This suggestion weakens about the importance of the first phase as it may not even be necessary to understand ``Grokking''

---

> ### Author Response · Authors · 2023-11-23
> **Response to Reviewer oyWf**
>
> ### Main Concerns
>
> **Q1:** The core claim of "sharp transition in test accuracy" is not supported by any of the generalization analyses. Without such an analysis, the best one can conclude is that the models can fit the training data and reach perfect training accuracy eventually.
>
> **A:** We thank the reviewer for raising the concern.
> 1. In our paper, we have instantiated our theorems on regression tasks on the matrix completion example in Sec. 3.3.3. We can prove that initially gradient flow can fit the observed entries but make no progress on the remaining ones, but eventually it can generalize to unobserved entries and recover the ground truth.
> 2. For classification tasks, we added generalization bounds in the updated version of our paper. Concretely, in the “empirical validation: grokking” part of Sec. 3.2.3, where we consider the sparse linear classification task, we added generalization bounds (proved in Section E), showing that the sample complexity required for achieving good generalization in the kernel regime is much higher than that of the rich regime.
> 3. In the general cases, our main results for both classification (Thm. 3.4 and 3.5) and regression (Thm. 3.9 and 3.10) settings highlight a sharp kernel-to-rich transition from $T=\frac{1-c}{\lambda}\log\alpha$ to $T=\frac{1+c}{\lambda}\log\alpha$. As the reviewer noticed, our results do not directly lead to generalization guarantees. However, for general deep learning, it is known to be very hard to provide any tight generalization bounds. In the literature, existing generalization bounds for DL are usually vacuous in real-world settings, and no generalization measures are known to be fully predictive of the actual generalization error in practice. Hence, the line of works on implicit bias usually uses the following strategy: analyze the training dynamics and show how the training method shapes a neural net’s properties that are known to be correlated with generalization, then exemplify the result in simple settings with generalization analysis. This is indeed the strategy we take.
>
> **Q2:** Comparison with Moroshko et al. (2020)
>
> **A:** While Moroshko et al. (2020) also characterize the implicit bias in the kernel and rich regimes, our results differ from theirs in two crucial aspects:
> 1. We consider running gradient flow with weight decay, while they do not use weight decay.
> 2. Our results highlight a sharp transition from kernel to rich regimes, while their paper does not imply such transition bounds.
>
> ### Other Questions
>
> **Q3:** Can the authors add more examples in text with datasets commonly used in ``Grokking'' literature?
>
> **A:**
> 1. The grokking phenomenon was originally observed on modular arithmetic datasets. In the revised version of the paper, we conducted experiments on modular addition and showed that enlarging the initialization scale or reducing the weight decay indeed delays the sharp transition in test accuracy. This is consistent with our theory, which predicts the grokking time to be scaled with $\frac{1}{\lambda}$ and $\log(\text{init scale})$,
> 2. Beyond algorithmic datasets, Liu et al. (2023) observed that grokking can also be induced by using a large initialization and a small but nonzero weight decay. This is true for many tasks, including image classification on MNIST and sentiment analysis on IMDB. This can be seen as an important justification of our theory.
>
> Ziming Liu, Eric J Michaud, and Max Tegmark. Omnigrok: Grokking beyond algorithmic data. In The Eleventh International Conference on Learning Representations, 2023.
>
> **Q4:** One point that the paper makes is about identifying a transition time from memorization to generalization and attributes it to different inductive biases. This is not true when weight decay = 0, which weakens the importance of the first phase as it may not even be necessary to understand ``Grokking''.
>
> **A:** In Sec 4 (now Sec B.1), we have included a discussion on the training regime without weight decay, where the perfect generalization is delayed. However, the transition in test accuracy is extremely slow: in Fig 2 (now Fig 4), we have shown that the transition lasts from $10^{10^2}$ to $10^{10^6}$ in continuous time. (The experiments are done via some simulation with unnaturally growing LR.) This means this grokking phenomenon is not relevant to the practice. It also does not quite satisfy the most strict definition of grokking, where the transition is required to be “sharp”.
>
> In contrast, in our main setting, we study training with large initialization and small weight decay, where the transition in the implicit bias is very sharp: $\frac{1-c}{\lambda} \log \alpha$ leads to the kernel predictor, but increasing it slightly to $\frac{1+c}{\lambda} \log \alpha$ leads to a KKT solution of min-norm/max-margin problems associated with the neural net.

---

### Official Review · Reviewer_XQUe · 2023-11-01

**Soundness:** 3 good
**Presentation:** 3 good
**Contribution:** 3 good
**Rating:** 6
**Confidence:** 3

**Summary:**

This paper studies grokking theoretically in homogeneous neural networks with large initialization scale and small weight decay. It shows that in both classification and regression settings, there are early and late implicit biases. Particularly, it shows that in the early phase the model corresponds to a kernel model and later it can escape it. It suggests that grokking happens when the solution given by the early implicit bias generalizes poorly while the solution given by the late bias generalizes well. It is also shown that the converse phenomenon can happen which they call misgrokking.

As an example, the paper focuses on a simple diagonal linear neural net for classification and shows that the early and late bias resp. correspond to minimization of L2 and L1 margin. They also identify these biases for a matrix completion model.

The paper further provides experiments showing that grokking can happen even in the absence of weight decay.

**Strengths:**

- The paper is quite rigorous and provides a provable instance of grokking for simple models. Furthermore, it attributes grokking to a transition from the kernel to the rich regime.
- The paper does the analyses for both classification and regression settings, further it shows the possibility of misgrokking.
- Empirically, it's shown that grokking can happen without weight decay but more slowly.

**Weaknesses:**

- There is very limited discussion on the possible extensions and limitations of this model. Moreover, results are in the case that the initialization scale goes to infinity which doesn't show (at least immediately) if these observations can be observed in the more common settings.
- Additional experiments can clarify other potential limitations of the analysis (e.g., gradient flow).

See the questions for more details.

**Questions:**

- Q1. What are the limitations of the analysis done in this paper? Ignoring the proof difficulties, do you think the change of implicit biases can be observed and explain grokking in more common settings?
- Q2. What would be the effects of stochasticity in GD (SGD) and large step size (distancing from gradient flow)?
- Q3. Can you further explain Remark 3.11?
- Q4. Can you further explain the learning rate and time scaling for Figure 2? (E.g., the $\log(t)$ has often very large values.)
- Q5. Is there any proof provided for Theorem 4.1?
- Q6. Can you further clarify why proving $ \|\frac 1\alpha
 \exp(\lambda t) \theta(t) - \overline{\theta}_{\mathrm{init}} \| \leq \epsilon$   for all $t \leq \min \{\frac{1}{\lambda} \log \alpha + \cdots + \Delta T, T_m \} $ would imply $T_m \geq \frac 1 \lambda \log \alpha + \cdots + \Delta T$ in proofs of Lemma A.5 and B.2?
- Q7. What is $q(\theta)$ in Appendix A.2? Also $r_i$'s?

## Minor Questions/Remarks
- Q8. In Figure 1, what is the norm presented? (Whether it's L1/L2 and if it's for $w$ or $u,v$?)
- Q9. I think the analyses require the data to be linearly separable for the classification setting? (Also, the model should be able to interpolate in the regression setting.) Although these assumptions are reasonable for overparametrized model, it would be beneficial if they were stated more clearly in the main text.
- Q10. In the paragraph before Corollary 3.5, isn't the kernel feature $(2x, -2x)$?
- R1. In the paragraph of Empirical Validation: Grokking, it would be nice if some references for $O(d)$ and $O(k\log d)$ sample complexities were provided. Further, if some references/theoretical results for poor generalization of L2 margin are provided, it would complete the theoretical part of grokking for classification.

---

> ### Author Response · Authors · 2023-11-23
> **Response to Reviewer XQUe (1/2)**
>
> ### Main Questions
>
> **Q1:** What are the limitations of the analysis done in this paper? Ignoring the proof difficulties, do you think the change of implicit biases can be observed and explain grokking in more common settings?
>
> **A:**
> 1. We added new experiments on modular addition in Sec. 2 to confirm that enlarging the initialization scale or reducing the weight decay indeed delays the sharp transition in test accuracy. Beyond algorithmic dataset, Liu et al. (2023) observed that large initialization and small weight can induce grokking on many popular tasks, including image classification, sentiment analysis and molecules. Thus, we believe that our theoretical insights on implicit bias can have broad implications in these settings as well.
> 2. One limitation is that our analysis can only handle the training dynamics with large initialization and small weight decay, but these may not be the only source of the dichotomy of the implicit biases. We further discuss in Sec B (previously Sec 4) that the late phase implicit bias can also be induced by implicit margin maximization and sharpness reduction, though the transition may not be as sharp as the weight decay case.
>
> **Q2:** What would be the effects of stochasticity in GD (SGD) and large step size (distancing from gradient flow)?
>
> **A:** As the training pipeline of deep learning can induce many confounding factors (e.g., stochastic noise, finite step size, weight decay, etc.), in this paper we focus on identifying simple yet insightful theoretical setups where grokking with sharp transitions can be rigorously proved and its mechanism can be intuitively understood. This is the reason why we focus on gradient flow in our analysis, and we leave it as future work to explore the effects of noise and large LR.
>
>
> **Q3:** Can you further explain Remark 3.11?
>
> **A:** Remark 3.11 (now Remark 3.13) states that if we use random initialization as stated in Theorem 3.10 (now Theorem 3.12) with small variance $\sigma^2$, then gradient flow on the matrix completion problem exhibits a transition from the kernel regime to rich regime. We need to add noise to initialization here because gradient flow may get stuck at saddle points for some initial points that form a zero-measure set. The result of the rich regime is already stated in Theorem 3.10. For the kernel regime, Corollary 3.9 (now Corollary 3.11) only applies to deterministic initialization, but if we add sufficiently small noise to the initialization, the trajectory of gradient flow would remain close to the original one for arbitrarily long time period, so gradient flow would still learn a solution that is close to the one in Corollary 3.9 (now Corollary 3.11).
>
> **Q4:** Can you further explain the learning rate and time scaling for Figure 2?
>
> **A:** The time scaling in Figure 2 (now Figure 4) is in log scale or double log scale. The main point of this figure is to show that implicit margin maximization can take an extremely long time to improve generalization. To simulate gradient flow for a very long time, we set the learning rate as $\frac{\eta}{\mathcal{L}(\theta(t))}$, where $\eta$ is a constant. The main intuition behind this is that the Hessian of $\mathcal{L}(\theta)$ can be shown to be bounded by $O(\mathcal{L}(\theta) \cdot \mathrm{poly}(||\theta||))$, and gradient descent stays close to gradient flow when the learning rate is much lower than the reciprocal of the smoothness. Since we view gradient descent as a simulation of gradient flow, the variable “t” in the figure stands for the continuous time, calculated by summing the learning rates up to this point. We have included these details in the revision.
>
> **Q5:** Is there any proof provided for Theorem 4.1?
>
> **A:** We do not view Theorem 4.1 (now Theorem B.1) as a part of our novel theoretical contributions because it directly follows from the previous paper (Li et al., 2021). Still, we present this theorem since it is worth pointing out that (Li et al., 2021) implicitly demonstrated one possible cause of grokking, which does not seem to be known in the literature.
>
> **Q6:** Can you clarify … [about a proof step in Lemma A.5 and B.2]
>
> **A:** At the beginning of Sec. A.2 (now Sec. C.2), we define $T_{\max} := \inf\\{ t \ge 0 :\|\frac{e^{\lambda t}}{\alpha}\theta(t) - \bar{\theta}\| > \epsilon_{\max}\\}$. In our proof of Lemma A.5, for all $t \le \min\\{T_{\max}, \frac{1}{\lambda}(\log \alpha - \frac{1}{L} \log \log A + \Delta T)\\}$, we can derive the bound $\|\frac{1}{\alpha}e^{\lambda t}\theta(t) - \bar{\theta}\| \le \epsilon_{\max}$, so by definition we must have $T_{\max}\ge \frac{1}{\lambda}(\log \alpha - \frac{1}{L} \log \log A + \Delta T)$.
>
> **Q7:** What are q and r?
>
> **A:** Sorry for the confusion of notations here. The function $q_i(\theta)$ is defined as $y_i f_i(\theta)$ (following Lyu & Li, 2020), and $r_i(\theta) = \ell(f_i(\theta); y_i) = \exp(-q_i(\theta))$ (following Moroshko et al., 2020). We have updated the paper.

---

> ### Author Response · Authors · 2023-11-23
> **Response to Reviewer XQUe (2/2)**
>
> ### Other Questions
>
> **Q8:** What is the norm in Figure 1?
>
> **A:** The norm in figure 1 is the parameter norm, i.e., $\\|\theta\\|_2$, where $\theta$ is rigorously defined in Sec. 3.2.3. We have made this explicit in the revised paper.
>
> **Q9:** The analyses require the data to be linearly separable for the classification setting. Also, the model should be able to interpolate in the regression setting. Although these assumptions are reasonable for overparametrized model, it would be beneficial if they were stated more clearly.
>
> **A:** Thanks for pointing this out. We have made these two assumptions explicit in Assumptions 3.3 and 3.8 in the revised version.
>
> **Q10.** In the paragraph before Corollary 3.5, isn't the kernel feature (2x, -2x)?
>
> **A:** Thanks for pointing out this typo. We have updated the paper.
>
> **R1.** In the paragraph of Empirical Validation: Grokking, it would be nice if some references for and sample complexities were provided. Further, if some references/theoretical results for poor generalization of L2 margin are provided, it would complete the theoretical part of grokking for classification.
>
> **A:** Thanks for the suggestions. In the revised version of the paper, we provide generalization guarantees for both L2 and L1 margin maximization in Sec. E. Our guarantees indicate that L2-max margin classifier requires $O(kd)$ samples to achieve good generalization, while L1-max margin classifier only requires $\tilde{O}(k^2)$ samples. Here since we are considering sparse vectors, $k$ is much smaller than $d$. As a result, L1 max-margin has a much smaller sample complexity.

---

### Author Response · Authors · 2023-11-23
**Paper Updated**

We would like to thank all the reviewers for providing helpful feedback. By taking the reviewers’ constructive suggestions into account, we have revised our paper. Concretely, we make the following important updates:

1. **More experiments:** As suggested by reviewers XQUe and oyWf, we added more experiments to the revised version to support our theoretical result.
    * To motivate the use of large initialization and small weight decay, we conduct experiments on module addition, which is the task where grokking was first observed (Power et al., 2022). We confirm that large initialization and small weight decay are important factors: enlarging the initialization scale or reducing the weight decay delays the sharp transition in test accuracy. The detailed setting of this experiment is discussed in Sec. 2.
    * For the low-rank matrix completion task that we consider in Sec. 3.3.3, we conduct experiments where the low-rank ground-truth matrix is constructed from a multiplication table, and the grokking phenomenon is also observed for this task.
2. **Generalization bounds:** As suggested by reviewers oyWf and XQUe, we add more discussions and results on the generalization guarantee in the rich regime.
    * For matrix completion, the original version already contains analyses on generalization. Now we added more explanations on the theorems (Sec. 3.3.3).
    * For sparse linear classification, we added generalization bounds for max L1-margin and max L2-margin classifiers to support our view that the former generalizes well but the latter does not (Sec. E).
3. **Polishing:** To make the paper easier to read, we added more explanations on our assumptions and results, and also reorganized the paper. Due to the space limit, we have moved Sec. 2 and Sec. 4 in the original version to the appendix.

---

### Meta-Review · Area_Chair_rWSB · 2023-12-11

**Metareview:**

This paper studies grokking theoretically in homogeneous neural networks with large initialization scale and small weight decay. It demonstrates early and late implicit biases. It suggests that grokking happens when the solution given by the early implicit bias generalizes poorly while the solution given by the late bias generalizes well. It is also shown that the converse phenomenon can happen, which here is called misgrokking.

Reviewers generally liked the rigorous analysis and concrete examples given in the paper (provided for two-layer diagonal linear networks.)

We hope the authors will incorporate the several minor points mentioned by the reviewers during the discussions.

**Justification For Why Not Higher Score:**

All reviewers gave a consistent assessment and score of 6, given the current scope of results and novelty.

**Justification For Why Not Lower Score:**

No major negative points were mentioned, which would warrant for reject at this point

---

### Decision · Program_Chairs · 2024-01-16

Accept (poster)